# Broadcast Ephemeris with Centimetric Accuracy: Test Results for GPS, Galileo, Beidou and Glonass

Alessandro Caporali * and Joaquin Zurutuza

Department of Geosciences and CISAS 'G.Colombo', University of Padova, 35131 Padova, Italy; joaquin.zurutuzajuaristi@unipd.it
* Correspondence: alessandro.caporali@unipd.it

**Abstract:** Here we test the capability of the Broadcast Ephemeris Message, in both its GPS-like (Keplerian ellipse with secular and periodic perturbations) and Glonass-like (numerical integration of a 9D state vector) formats, to reproduce a corresponding precise ephemeris. We start from a daily Rinex 3.04 navigation file for multiple GNSS and the corresponding SP3 precise orbits computed by CNES (Centre National d'Etudes Spatiales) for GPS, Glonass, Galileo and CODE (Center for Orbit Determination in Europe) for Beidou, and compute broadcast ECEF coordinates and clocks. The pre-fit discrepancies are converted by least squares to corrections to the broadcast ephemeris parameters in two-hour consecutive arcs (for GPS, Galileo and Beidou) and to a set of seven Helmert parameters for the entire day, to align in origin, orientation and scale to the common GNSS IGS14 Reference Frame. The test cases suggest that the Broadcast Ephemeris Message, complemented with Reference Frame information, can reproduce the precise ephemeris and clocks with centimetric accuracy for intervals at least equal to the respective validity times, typically 2 h. The broadcast ephemeris of Glonass consists of three initial positions and velocities at epoch, three constant Lunisolar accelerations for the satellite position, and of three polynomial coefficients for the satellite clock. The 9D vector of state is numerically integrated to generate position and velocity data within the validity time (0.5 h) of the message. To test the capability of this model to reproduce the corresponding values of a precise ephemeris, the 9D vector of state and clock polynomials are adjusted until the rms (root mean squared spread) of the post-fit residuals relative to a precise orbit (CNES's in our case) is minimum. We show in one example (one satellite for one day) that the Glonass type of message can reproduce a precise ephemeris and clock with a rms spread of 0.025 m over one-hour arcs. Volume computations on one month of data with all available satellites confirm the test results. For GPS, Glonass, Galileo and Beidou, the best fitting clock values predicted by our second order polynomials, based on a 15 min sampling, are shown to fit the corresponding high rate clocks (30 s sampling) of MGEX with zero bias and a rms spread of 0.062 ns (GPS G01), 0.023 ns (Galileo E01), 0.43 ns (Glonass R01), 0.086 ns (Beidou C07) and 0.086 ns (Beidou C12). Modifications to the GPS-like message structure and Glonass algorithm are proposed to increase the validity time by including the effect of the 3rd zonal harmonic of the Earth's gravity field. The potential of the RTCM messages for broadcasting the improved navigation message is reviewed.

**Keywords:** navigation message; precise ephemeris; orbital model; clock model; reference frame; multi GNSS

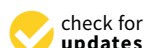



## 1. Introduction

Is the limited accuracy of the broadcast ephemeris caused by inherent limitations of the model to compute the spacecraft coordinates and clock offset, or by the model coefficients in the broadcast message being insufficiently accurate? In this paper we provide arguments based first on detailed test cases and then on volume computations that the broadcast ephemeris can in fact provide spacecraft coordinates and clock offsets as accurate as the corresponding precise ephemeris provided that the coefficients of the broadcast ephemeris are appropriately tuned.

For each GNSS, the Interface Control Document (ICD) and the documentation of the RINEX format of the Navigation Message provide full description of the parameters which enable the calculation of the Earth Centered Earth Fixed (ECEF) coordinates of a Space Vehicle (SV), and its clock offset relative to the common time scale of a specific GNSS. Most GNSSs adopt a model which approximates the orbit of the SV with an arc of a Keplerian ellipse with periodic and secular perturbations in the Keplerian orbital elements [1]. The length of the arc, or validity time of the set of these orbital parameters, is typically two hours, so that a new ephemeris message is normally issued every two hours, with a reference epoch called Toe (Time of ephemeris) and a corresponding orbital phase angle denoted by $M_0$. The clock offset is instead described by a quadratic polynomial with a reference epoch called Toc (Time of clock). There are 15 parameters in the orbital model and 3 parameters of the clock polynomial. Often Toe and Toc coincide. GNSSs falling into this mathematical scheme of the orbit are GPS (USA), Galileo (Europe), Beidou (China), NAVIC/IRNSS (India) and QZSS (Japan). Each GNSS computes the broadcast message using different control stations and reference clocks, which may result in small but significant differences in the realization of the Terrestrial Reference Frame and System Time [2–6].

Regarding Glonass, the instantaneous position of each SV is computed by numerical integration of the equations of motion. The force field consists of a term approximating the gravitational field of the Earth truncated to the $J_2$ zonal. Because the numerical integration is carried out in a rotating frame, the force includes also the centrifugal and Coriolis terms. Third body perturbations, typically caused by the gravity of the Moon and the Sun, are modeled by accelerations kept constant during the validity time of the message, which, for Glonass, is 30 min. Thus, there are in each Ephemeris Message three ECEF positions and three velocities at epoch Toe, and three constant Lunisolar accelerations, for a total of nine parameters. The clock offset is again modeled by a quadratic polynomial with a time Toc as reference. An important feature of the Glonass approach is that the Reference Time scale tracks the Leap Second, so that the computation epoch must be offset by the number of Leap Seconds corresponding to the epoch [7].

Further details can be found in the pertinent ICDs (Interface Control Documents). Noteworthy is the use of different values of the product of the gravitational constant by the mass of the Earth. Galileo, Glonass and Beidou, for example, use $\mu = 3.986004418 \times 10^{14} \, \mathrm{m^3 \, s^{-2}}$, whereas GPS and QZSS use $\mu = 3.9860050 \times 10^{14} \, \mathrm{m^3 \, s^{-2}}$.

Merged files of Broadcast Ephemeris (BRDM) in the standard RINEX 3.04 format can be downloaded for example from the BKG server (*Bundesamt fuer Kartographie und Geodaesie*). As mentioned earlier, these messages inherit systematic differences due to the realization of the Reference Frame, the relative alignment of the GNSS-specific System Time and other peculiarities such as the gravity constant or the nominal Earth rotation rate. All these interoperability issues must be considered by a user who wants to compute its ECEF coordinates by simultaneously processing data from several GNSSs [8]. Likewise, the receiver clock will have to be synchronized to the time scales of different GNSSs, which are not aligned to each other to sufficient accuracy [4].

Recognizing the several issues related to the simultaneous use of data from several GNSSs, the International GNSS Service (IGS) of the International Association of Geodesy (IAG) has promoted the MGEX (Multiple GNSS Experiment [5,9,10]). As part of the MGEX activities, several Analysis Centers make available Precise Ephemeris files where ECEF coordinates and satellite clock offsets are computed using a fiducial network of several hundreds of stations with nearly global coverage and a common time scale. These ephemeris files in the conventional SP3 format contain directly the ECEF coordinates and satellite clock offsets relative to a common time scale at, normally, 15-min intervals. The orbit calculation follows international conventions on the coordinates of the stations, origin orientation and scale of the Terrestrial Reference System (currently ITRF2014 or equivalently IGb14; IGS14 during the dates of the study) [11], antenna models [12], Earth Rotation Parameters, Earth and Ocean Tides, atmospheric loading and gravity field coefficients.

The solution includes modelling the SV attitude for Solar Radiation Pressure [13], the ionospheric delay, the tropospheric refraction and the receiver antenna model, for example. These precise ephemeris files are consequently considered the most accurate and rigorous representation of the instantaneous position of the Center of Mass (CoM) of each SV in an equally well-defined Terrestrial Reference Frame [14,15]. Several Analysis Centers deliver precise multiGNSS precise ephemeris files. The mutual consistency among the different solutions is discussed by [16] who estimated in a few cm to several decimeters, depending on the GNSS, the differences of the orbit/clock products of the Analysis Center. The impact of MGEX products provided by different IGS ACs on post-processing PPP performance in terms of accuracy, availability and consistency is discussed in [17]. According to [18] orbit comparisons among the MGEX Analysis Centers show agreements of about 0.1–0.25 m for Galileo, 0.1–0.2 m for Beidou MEOs (Medium Earth Orbit), 0.2–0.3 m for Beidou IGSOs (Inclined Geosynchronous Orbit) and 0.2–0.4 m for QZSS. Clock comparisons of individual ACs have a consistency of 0.2–0.4 ns for Galileo, 0.2–0.3 ns for Beidou IGSOs, 0.15–0.2 ns for Beidou MEOs, 0.5–0.8 ns for Beidou GEOs (Geostationary Orbit) and 0.4–0.8 ns for QZSS.

The satellite clock corrections sampled at 15 min may lose accuracy when interpolated at a higher rate, due to the stability characteristics of the on-board atomic clocks [19]. The Allan variance of the Cesium and Rubidium frequency standards is known to increase at shorter sampling times due to flicker noise more than Galileo's Passive Hydrogen Maser. Consequently, for those applications (e.g., PPP Precise Point Positioning) which require the satellite clock offset to be computed at a much higher rate (1–10 Hz) than one sample every 15 min, it is necessary to have epoch estimates rather than interpolations. For this purpose, MGEX issues specific clock files with 30 s sampling.

Comparisons between ECEF coordinates and clock offsets computed with merged Broadcast Ephemeris files (BRDM) and SP3 values taken as reference have been the subject of several investigations. The authors of [20,21] proposed the concept of SISRE (Signal in Space Range Error), a statistical characterization of the uncertainty of the modeled pseudorange due to errors in the broadcast orbit and clock. Comparing the Broadcast and precise (IGS) data, they demonstrated SISREs ranging of 0.2 m (Galileo), 0.6 m (GPS), 1 m (Beidou) and 2 m (Glonass) over a 12-month period in 2017. In [22], Galileo's potential for highest accuracy in positioning is confirmed. In this comparison a crucial role is played by the Center of Mass (CoM) to Antenna Phase Center (APC) correction (CoM is the reference point for SP3 orbits and APC is the reference point for Broadcast Ephemeris). These range from 0.04 m (GPS IIR-B/M) to 2.5 m (Glonass M) and can be modeled as a boresight bias or as a small-scale factor (up to $8 \times 10^{-8}$) [23].

As we shall see in detail in the next sections, for each SV, the differences between $XYZT$(BRDM) and $XYZT$(SP3), the ECEF coordinates and satellite time correction computed from the broadcast message and the precise SP3 data file, show clear systematics and discontinuities at the crossover between two contiguous ephemeris blocks. It is then natural to ask if the broadcast parameters can be adjusted, for example in a least square sense, to minimize such discrepancies. As it will be shown in a number of examples, there are indications that the differences between broadcast and SP3 ephemeris and clock can be brought down to the centimeter level by appropriate tuning of the broadcast parameters complemented by additional reference frame parameters, depending on the type of navigation message (GPS-like or Glonass-like).

The approach is then applied to process one month of data using all the available satellites. The average spectra of the Broadcast to SP3 differences in the spatial coordinates clearly indicate that the proposed approach remove the spectral features at low frequencies caused by the inaccuracies of the broadcast model. The spectra of the post-fit residuals are shown to be very nearly flat and with features of at most few centimeters.

This paper is organized as follows. Section 2 addresses the mathematical model of the least squares adjustment and discusses the results for GPS, Galileo and Beidou in IGSO and MEO orbits. We concentrate on specific examples, in order to do a detailed analysis. Hence, we consider one satellite per constellation and one day, both chosen arbitrarily. Section 3

addresses the improvement of the Glonass navigation message using SP3 orbits and clocks. Then, specific issues are addressed: comparison between high rate clock products of IGS/MGEX and the polynomial model (Section 4), and possible improvements in the message content for increased accuracy (Section 5). In Section 6, we apply the approach described above to volume computations involving one month of data and all the satellites of the four constellations. We show that the seven Helmert parameters averaged over the 30 days and all the satellites of each constellations average to zero, within one standard deviation, with the only exceptions of the rotations Rx and Rz of Galileo slightly above the one sigma threshold. This suggests that, on average, the GNSS-specific reference frame is aligned to the ITRF2014, but that for centimetric accuracy the seven Helmert parameters appropriate to the day and satellite should be used. Then we compare spectra of the pre-fit and post-fit residuals of the Broadcast–SP3 position differences averaged over one month and all satellites of a given constellation. We show that the former has significant power (1 m typically) at low frequencies (<1 cycle/6 h), whereas the latter has a spectrum very close to flat. This supports the idea that the proposed approach has effectively accounted for most of the signal, at the centimetric level. We remark in the Conclusions that if the broadcast ephemeris, when appropriately tuned, is capable of centimetric accuracy, then there are some important consequences. One is that the broadcast format, which is optimized in bandwidth requirements, could be used in real time for precision applications, taking advantage of the existing RTCM messages. Another inference is that ephemeris and clocks in the broadcast format could be used in place of the same data in the SP3 format, at least for MEO satellites.

## 2. Mathematical Model and Results for GPS, Galileo, BeiDou

To compute the ECEF coordinates of an SV as a function of time, the GPS navigation message adopts an algorithm based on a Keplerian ellipse with secular and periodic perturbations on selected orbital parameters. The validity time of the model is nominally two hours. Outside the validity time interval, the accuracy of the coordinates tends to degrade progressively. The orbital arc of nominal duration two hours has an orientation defined by three angles: Inclination $I_0$, Longitude of Ascending Node $\Omega_0$ and Longitude of Perigee $\omega$. The size of the ellipse is given by the semimajor axis $a$ and the eccentricity $e$. The angular position of the SV at a Reference Time Toe (Time of Ephemeris) is $M_0$ and is counted from the perigee (for a pictorial view and definitions see https://www.gsc-europa.eu/system-service-status/orbital-and-technical-parameters, accessed on 10 October 2021).

These six orbital parameters, which would be all constant if the total force were that of a point mass, are allowed to vary with time to account for deviations from the point mass force model. Inclination and Ascending Node are allowed to increase with a rate $dI_0/dt$ and $d\Omega_0/dt$ constant during the validity interval; the orbital angular velocity $n$, which by Kepler's third law is inversely proportional to the semimajor axis to the 3/2 power, is perturbed by a constant term $\Delta n$. The radial, along track and cross track position of the satellite are assigned periodic perturbations with a period equal to one-half the orbital period. These perturbations are described by sinusoids with separate amplitudes for the sine and cosine components. There is a total of 15 parameters which refer to the Time of Ephemeris Toe. Three additional clock parameters (offset $a0$, drift $a1$ and drift rate $a2$) refer to the Time of Clock Toc. Hence, every two-hour arc is described by 18 parameters and two reference epochs, Toe and Toc, which often coincide.

The Reference Precise Ephemeris are computed with rigorous constraints on the origin, orientation and scale of the Terrestrial Reference Frame (TRF), which in our case is ITRF2014 or, specifically, IGS14. In the GPS-like algorithm, only the shape and size of the orbital arc are defined. It follows that to generate spatial coordinates as close as possible to those in the SP3 file, it is appropriate to solve for a set of Helmert parameters. This can be done once per day, i.e., about two complete revolutions, because their global character requires sampling over the entire orbital arc.

The mathematical model used for the GNSSs with a GPS-like message to express precise ephemeris and clocks in a broadcast form is described by Equation (1):

$$\sum_{i=1}^{96} [\Delta(XYZT)]^2 =$$

$$f(Tx, Ty, Tz, Rx, Ry, Rz, Sc; a0, a1, a2, Crs, \Delta n, M_0, Cuc, e, Cus, sqrt(a), Cic, \Omega_0, Cis, I_0, Crc, \omega, \frac{d\Omega}{dt}, \frac{dI}{dt}) = min$$

(1)

We seek to minimize the sum, over the 96 epochs (384 equations) in one daily SP3 file, of the squared differences $\Delta(XYZT)$ of $XYZT$(SP3) and $XYZT$(BRDM), that is, the difference between the SP3 and broadcast ECEF coordinates and clock correction of a given Space Vehicle. This difference is parametrized by one set of seven Helmert parameters (three translations $T$, three rotations $R$ and one scale $Sc$) valid for the entire day and 12 sets each of 18 parameters, one for each two-hour arc. The function $f$ can be linearized by series expansion in a neighborhood of the broadcast values of the parameters (the Helmert parameters have zero a priori value). A partial derivative matrix H is computed numerically by approximating the partial derivatives with finite central differences. The best fitting corrections to the a priori values are computed by solving the linear system of 223 normal equations. The matrix $H^T H$ ($^T$ stands for transposed), however, is poorly conditioned, as it may be expected. The rotational Helmert parameters tend to be highly correlated with the angles of orientation of the orbit. A well-conditioned matrix is obtained by increasing the weight of the matrix elements corresponding the parameters in red in Equation (1), which are therefore fixed to the broadcast values. In practice we solve for seven global Helmert parameters (i.e., one set for one day) and 12 sets each of seven parameters (the mean anomaly $M_0$ at the reference epoch Toe and three pairs of amplitudes of cosine and sine components of periodic perturbations along track ($Cuc$, $Cus$), radial ($Crc$, $Crs$) and across track ($Cic$, $Cis$) (Figure 1). This is equivalent to constrain the Helmert rotations and solve for the angles defining the spatial orientation of the orbit.

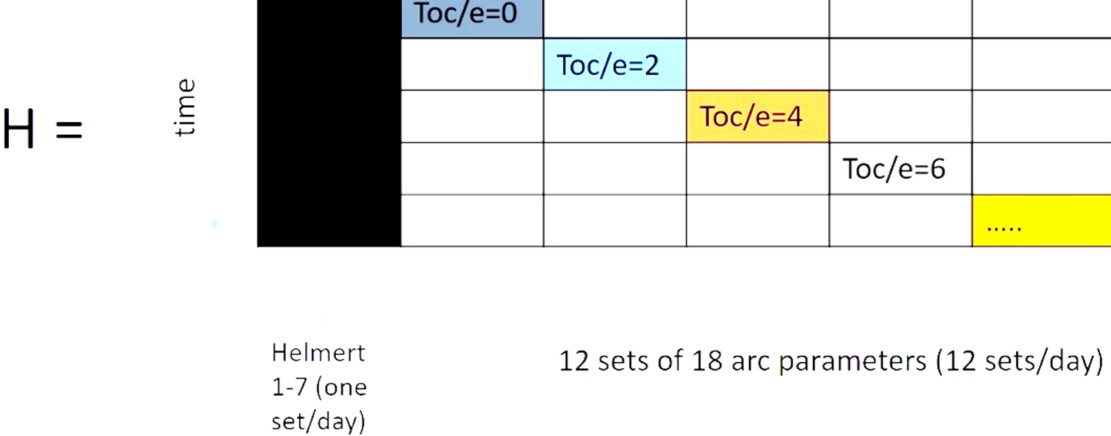

**Figure 1.** Schematic structure of the partial derivative matrix for the GPS-like model. The first vertical block represents the partials of $XYZT$ pre-fit residuals relative to the Helmert parameters, appropriately scaled. The block diagonal part contains 12 blocks each of 4 rows of 18 partial derivatives, using as a priori the broadcast values of the parameters which are indexed with Toe for the coordinates and Toc for the clock.

In the following, we use this model for data referring to 2 January 2020, as an arbitrary choice. We consider G01 for GPS, E01 for Galileo, C07 for Beidou IGSO and C12 for Beidou MEO. The reference SP3 file is available at the IGS/MGEX servers. We have arbitrarily chosen the CNES SP3 file for GPS and Galileo (https://cddis.nasa.gov/archive/gnss/products/mgex/2086/GRG0MGXFIN_20200020000_01D_15M_ORB.SP3.gz, accessed on 10 October 2021), and the CODE SP3 file for Beidou (https://cddis.nasa.gov/archive/gnss/products/mgex/2086/COD0MGXFIN_20200020000_01D_05M_ORB.SP3.gz, accessed on 19 September 2021). The Mixed broadcast ephemeris file was downloaded from the BKG

server (https://igs.bkg.bund.de/root_ftp/MGEX/BRDC_v3/2020/002/brdm0020.20p.Z, accessed on 10 October 2021). As mentioned in the Introduction, our results will be valid for the specific satellites and epochs only. To extend the validity of the conclusions drawn in this paper beyond the specific satellites and epoch requires volume calculations. In this paper we concentrate on the mathematical approach as a prerequisite for further massive processing. Epoch-dependent departures from the results described here could for example be caused by a different angle between the Sun's geocentric vector and the satellite orbital plane (beta angle), which controls the force caused by the solar radiation pressure.

## 2.1. Results for GPS

Figure 2 shows the pre-fit and post-fit residuals for G01 (Block IIF). The pre-fit residuals are computed using the Broadcast Ephemeris blocks of appropriate Toe and Toc, and the post-fit residuals are computed with the tuned parameters computed with Equation (1). In Figure 2, the pre-fit differences are represented with colored dots, one color for each two-hour ephemeris block, with the exception of the first block where the broadcast ephemeris with Toe = 2 h has been used for an interval of 4 h, from 00:00 to 04:00, for test purposes. The vertical lines have a two-hour spacing. The diamonds represent the post-fit residuals, after the adjustment of the seven Helmert parameters and of the 11 sets of seven arc parameters. In Section 5, we will discuss in more detail the oscillatory pattern of the post-fit residuals, in connection to an improvement of the model to include the contribution of the third zonal harmonic of the gravity field. Likewise, it will be pointed out that the higher noise in the first 4 h of the post-fit residuals of the clock corrections suggests that the second order clock polynomial is unable to track the high frequency noise of the satellite clock, if the fit interval is as long as 4 h, as it may be expected. This will be further discussed in Section 4 in relation to the comparison with high rate clocks.

The overall improvement in doing the fit described by Equation (1) is summarized as follows: the mean and rms (root mean square spread) of the pre-fit residuals is $-0.013 \pm 0.917$ m, and $0.000 \pm 0.051$ m for the post-fit residuals. This statistic includes the spatial and temporal coordinates. The adjusted clock polynomials fit the SP3 clock data with 0.007 m rms, whereas for the spatial coordinates the corresponding figure is a factor of seven larger.

Figure 3 describes the epoch-wise corrections to the 11 sets of seven arc parameters, and the corrections to two of the three estimated clock parameters ($a2$ is not shown but computed). The correction to the mean anomaly $M_0$ means that at the epoch Toe the satellite is moved back or forth along the orbit, relative to the orbital phase predicted by the broadcast ephemeris. Table 1 finally provides the seven Helmert parameters for this daily set, and their estimated formal error 1 sigma.

**Table 1.** Helmert parameters relating the origin, orientation and scale of the Broadcast reference frame of G01 relative to the IGS14 frame of the SP3 precise ephemeris, 2 January 2020. 'mas' stands for milliarcsec.

| | Tx (m) | Ty (m) | Tz (m) | Rx (mas) | Ry (mas) | Rz (mas) | Scale |
|---|---|---|---|---|---|---|---|
| Estimated | $-0.09$ | 0.20 | $-0.39$ | 1.2 | 1.2 | $-11.1$ | $0.60 \times 10^{-08}$ |
| 1 sigma formal error | 0.04 | 0.04 | 0.02 | 0.00 | 0.00 | 0.00 | $8.48 \times 10^{-10}$ |

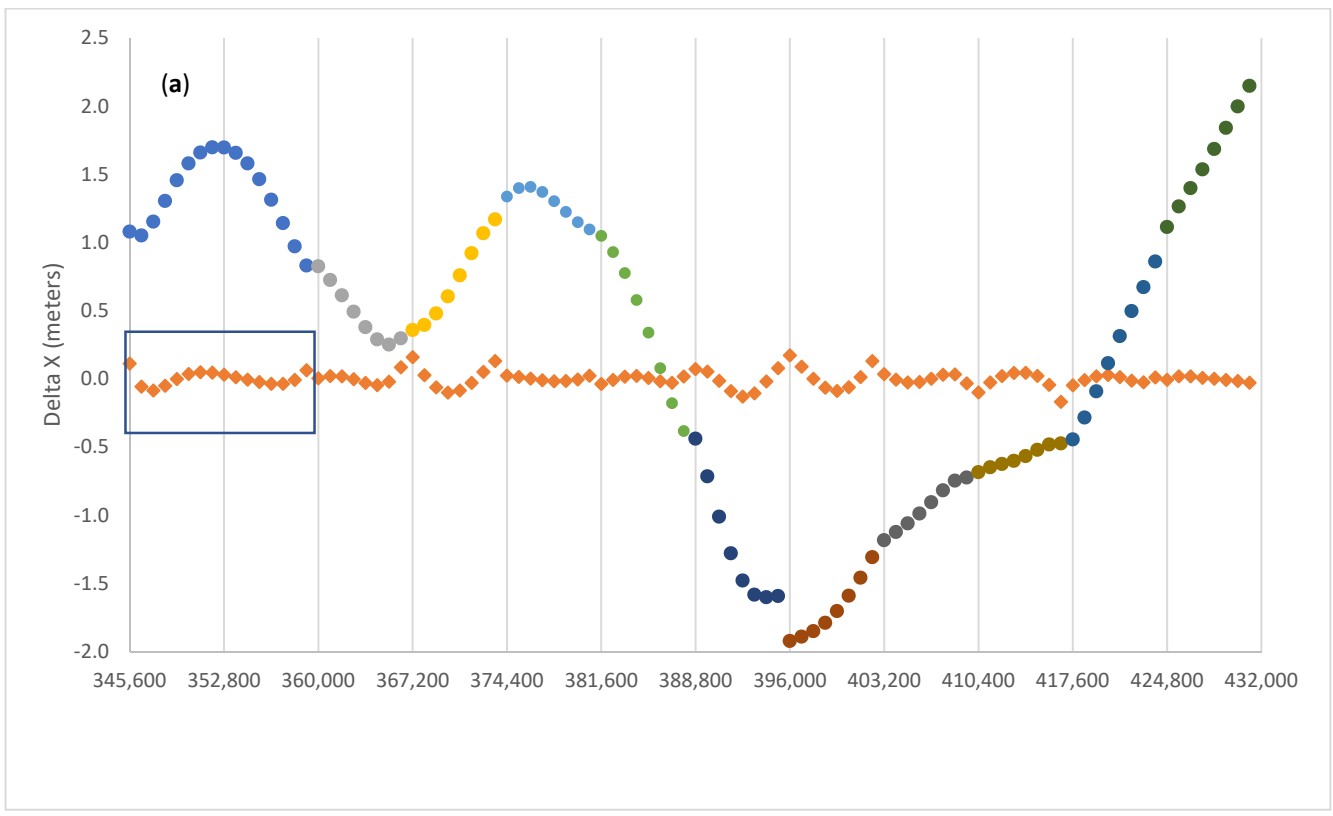

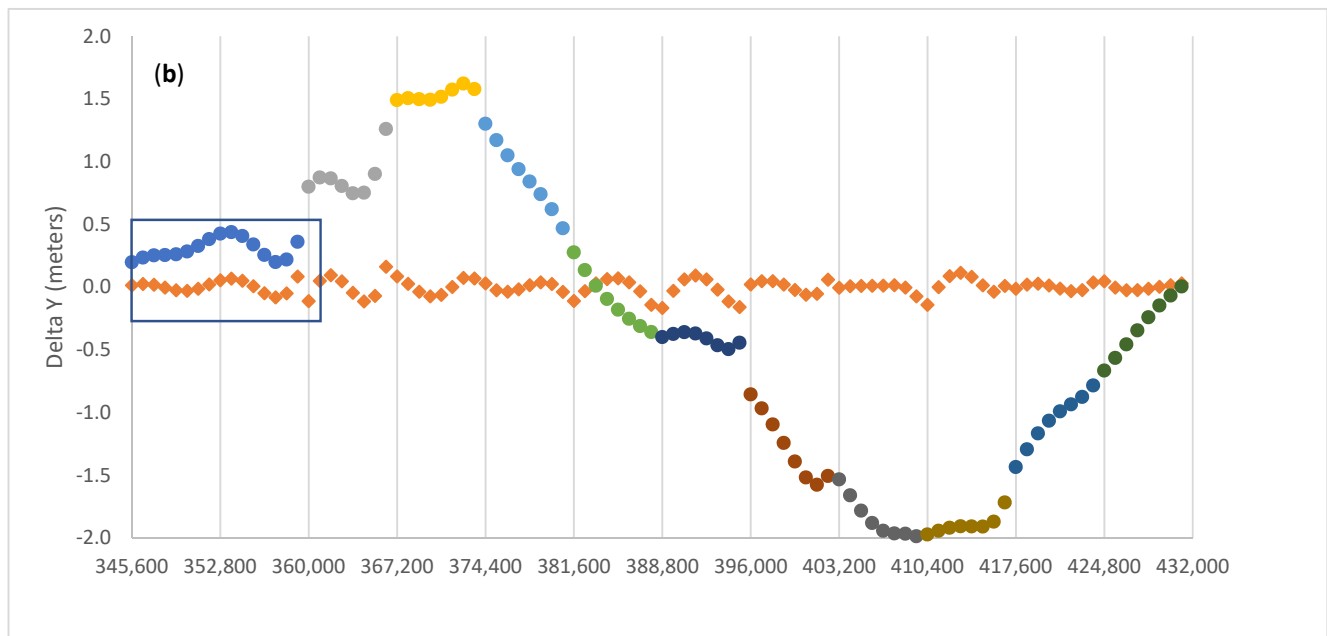

**Figure 2.** *Cont.*

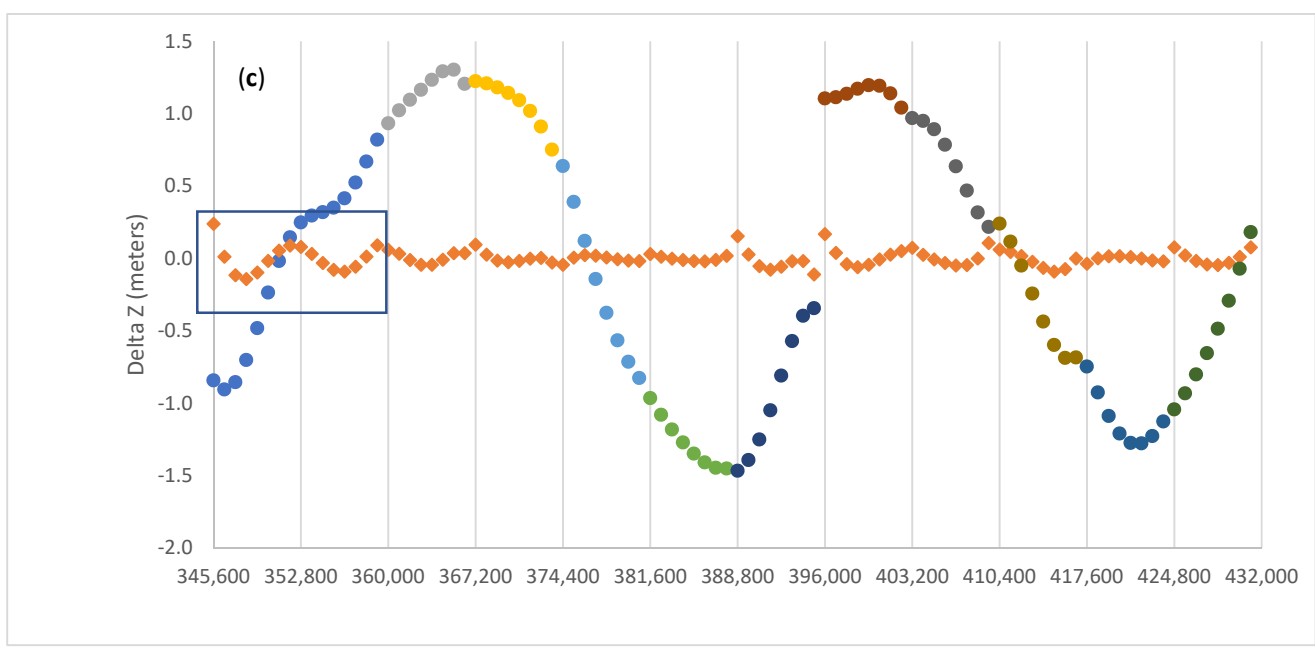

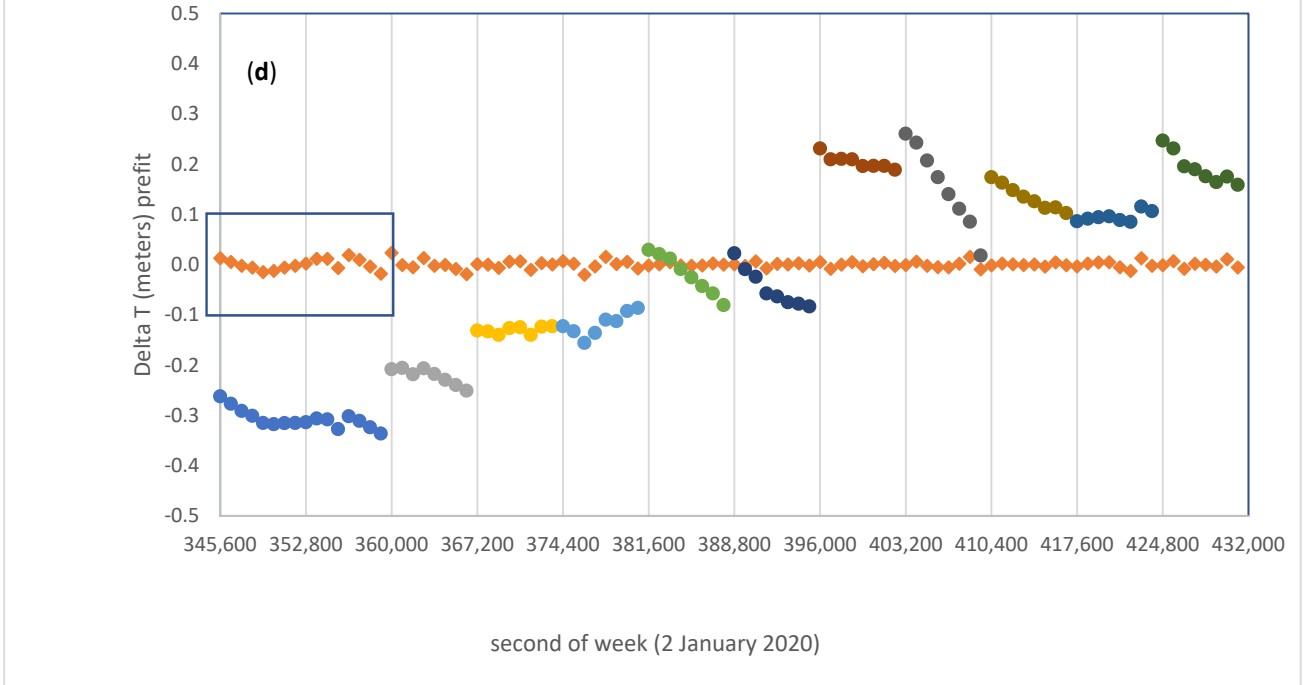

**Figure 2.** Pre-fit (dots: different colours for different ephemeris blocks) and post-fit (diamonds) residuals of (**a**) X, (**b**) Y, (**c**) Z, and (**d**) T for G01. Vertical line spacing is every two hours. The rectangle indicates a 4 h rather than 2 h arc. The residuals will be discussed in Section 4.

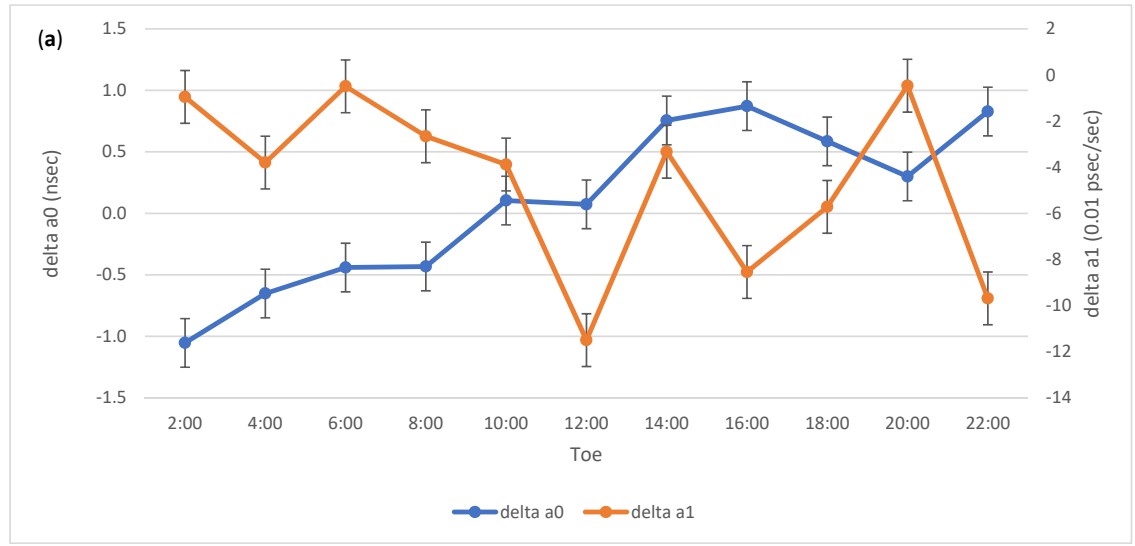

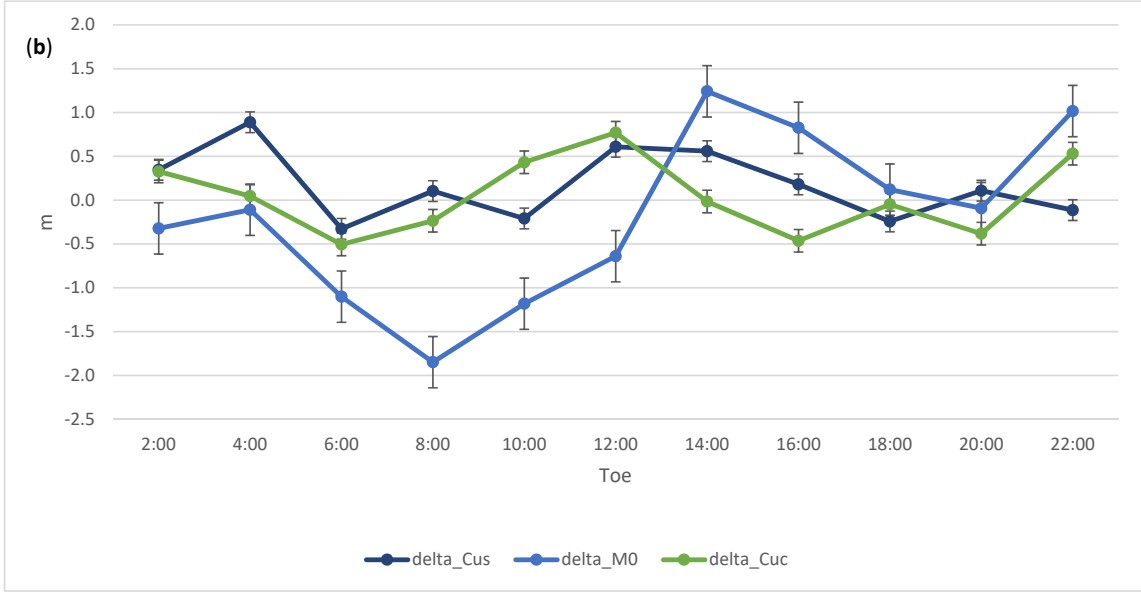

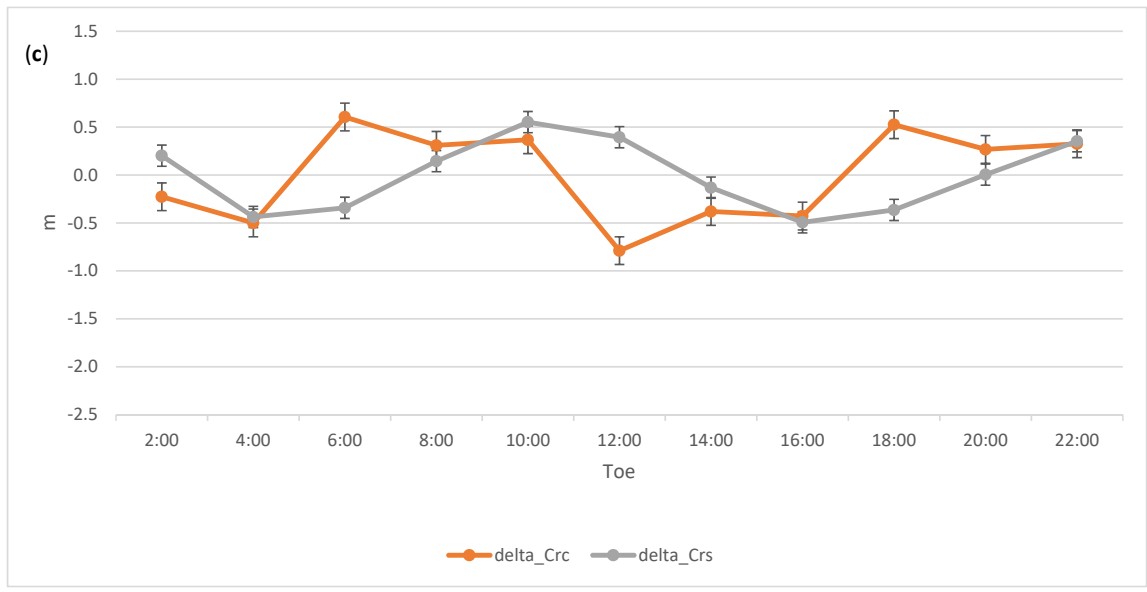

**Figure 3.** *Cont.*

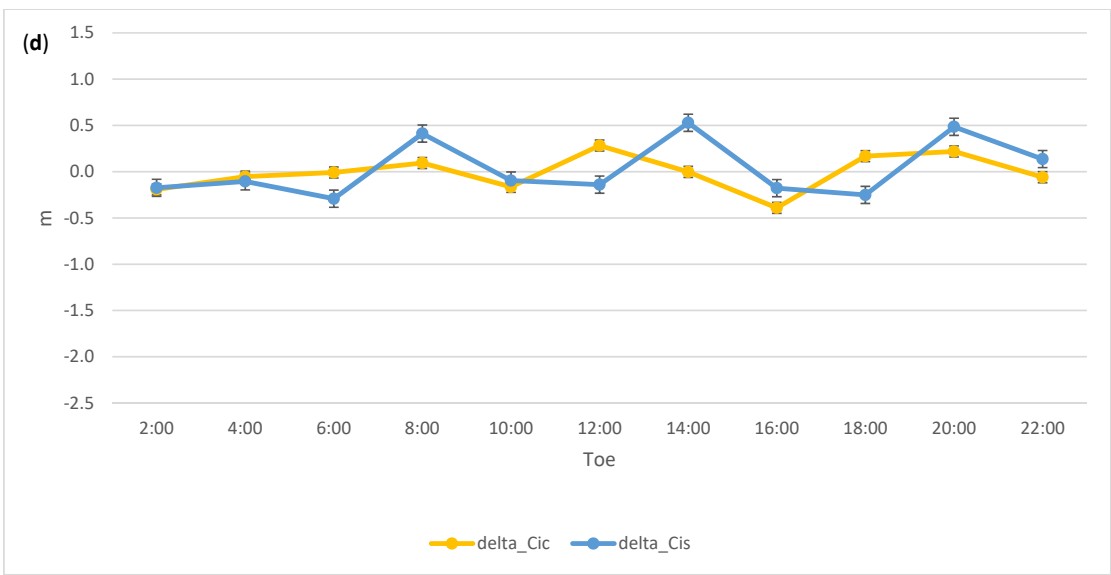

**Figure 3.** Best fitting corrections to (**a**) the clock polynomial parameters; (**b**) the Mean anomaly at Toe and amplitude of the cosine and sine perturbations along track (Cuc, Cus) due to the second zonal harmonic; the amplitude of the cosine and sine perturbations in the (**c**) radial (Crc, Crs) and (**d**) cross track (Cic, Cis) directions. To convert to radians the scale factor $26 \times 10^6$ m should be used. Error bars are 1 sigma formal uncertainties. Satellite G01 for day 2 January 2020.

### 2.2. Results for Galileo

For Galileo we have chosen the E01, an FOC (Full Operational Capability) SV, as an example. The repetition rate of the broadcast ephemeris is for Galileo usually 10 min, but not always at a regular rate. The rate of update of the message is considerably higher than GPS, so that a tailored ephemeris/clock is available at each epoch of observation. We seek broadcast model coefficients which make this high repletion rate unnecessary both for position and clocks. We expect that the validity time of the Galileo navigation message should be in fact comparable with that of GPS, for example. We have chosen those ephemeris blocks with a Toe/Toc closest to the even hours of day 2 January 2020. Figure 4 shows the difference between the raw residuals (original broadcast coordinates and clock minus corresponding SP3 values) described by colored dots (one color for each Toe); the post-fit residuals are shown by diamonds. The clock correction is expressed in meters. By comparison with GPS, it can be noted that the original ephemeris has, relative to the reference precise orbit, a pattern suggesting periodic resets, to keep the broadcast position and time as close as possible to the precise value, in the neighborhood of the respective Toe. In this way the effects of the third zonal harmonic ($J_3$) are probably less visible than with a longer sampling time, as discussed for G01 in Figure 2. Moving away from this reference epoch, the departure from the reference values increases, so that the ephemeris needs to be replaced by a more recent block near the edge of the validity period. The broadcast ephemeris with best-fitting parameters (post-fit residuals, y axis to the right) tracks instead the precise ephemeris very closely and with no divergence at the edges of the validity interval.

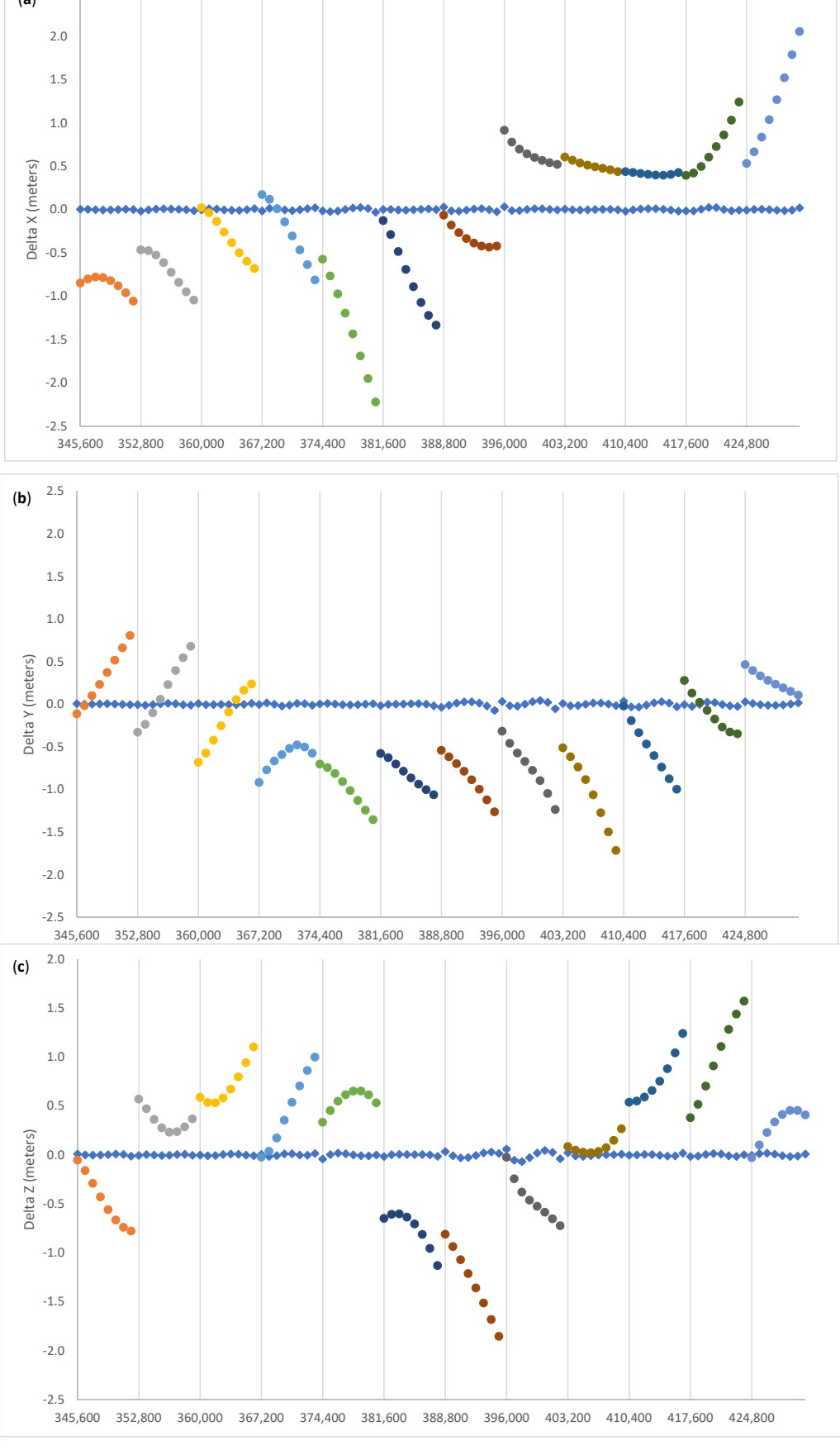

**Figure 4.** *Cont.*

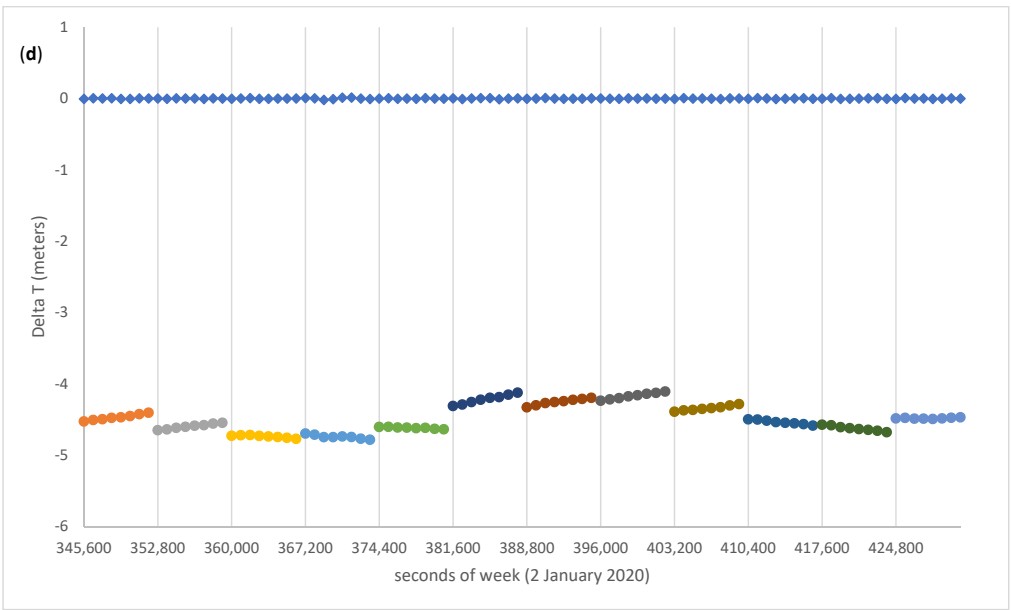

**Figure 4.** Pre-fit (dots, different colors for different ephemeris blocks) and post-fit (diamonds) residuals of (**a**) X, (**b**) Y, (**c**) Z, and (**d**) T for E01.

The overall improvement for Galileo's E01 in doing the fit described by Equation (1) is summarized as follows: the mean and rms of the pre-fit residuals is $-1.224 \pm 1.992$ m, and $0.000 \pm 0.014$ m for the post-fit residuals. This statistic includes the spatial and temporal coordinates. The adjusted clock polynomials fit the SP3 clock data with 0.002 m rms, whereas for the spatial coordinates the corresponding figure is a factor of seven larger.

Figure 5 shows the corrections to two clock parameters (a), to the Mean anomaly at epoch Toe and cosine and sine amplitude of the corrections to second harmonic periodic perturbations along track (b). In the bottom left and bottom right plots, we have the cosine and sine amplitudes of the corrections for the radial (c) and across track (d) components, respectively. Table 2 gives the Helmert parameters of the broadcast reference frame relative to the IGS14 frame of the precise ephemeris. Here it is noteworthy that the broadcast frame of E01 has a negligibly small misalignment to IGS14. To account for the CoM-APC offset the offsets in https://www.gsc-europa.eu/support-to-developers/galileo-satellite-metadata (accessed on 10 October 2021) have been used. This calibration implies that the scale factor must be negligibly small, as it results from Table 2.

**Table 2.** Helmert parameters relating the origin, orientation and scale of the Broadcast reference frame of E01 relative to the IGS14 frame of the SP3 precise ephemeris, 2 January 2020.

|  | Tx (m) | Ty (m) | Tz (m) | Rx (mas) | Ry (mas) | Rz (mas) | Scale |
|---|---|---|---|---|---|---|---|
| Estimated | −0.05 | −0.03 | −0.04 | 0.3 | −0.6 | −0.8 | $0.70 \times 10^{-10}$ |
| 1 sigma formal error | 0.02 | 0.02 | 0.02 | 0.29 | 0.17 | 0.06 | $4.68 \times 10^{-10}$ |

### 2.3. Results for Beidou C07

We address here C07, an example of Beidou GNSS, that is an SV in an IGSO orbit: this is a geosynchronous orbit with an inclination comparable to that of GPS or Galileo. The orbit pattern in ECEF coordinates is such that the X and Y coordinates sample a limited range of values. Consequently, one expects a rank deficiency in the normal equations and that the translational Helmert parameters are poorly constrained by the lack of geometry. We have therefore constrained the translational Helmert parameters to zero and solved for Helmert rotations and scale (Table 3), and the seven arc parameters and three clock parameters at intervals of 2 h. For Beidou, the precise SP3 ephemeris computed by CODE

were used, being those of CNES not yet available. This also provides a way of using data from more than one Center.

One more aspect to consider is that this SV is modeled with the GPS type of ephemeris which is optimized on an orbital radius of ca. 26,000 km, whereas the geosynchronous orbit has a radius of roughly 42,000 km. It follows that the perturbations due to the Earth's gravity field will be attenuated, and those due to the Sun and the Moon will tend to increase.

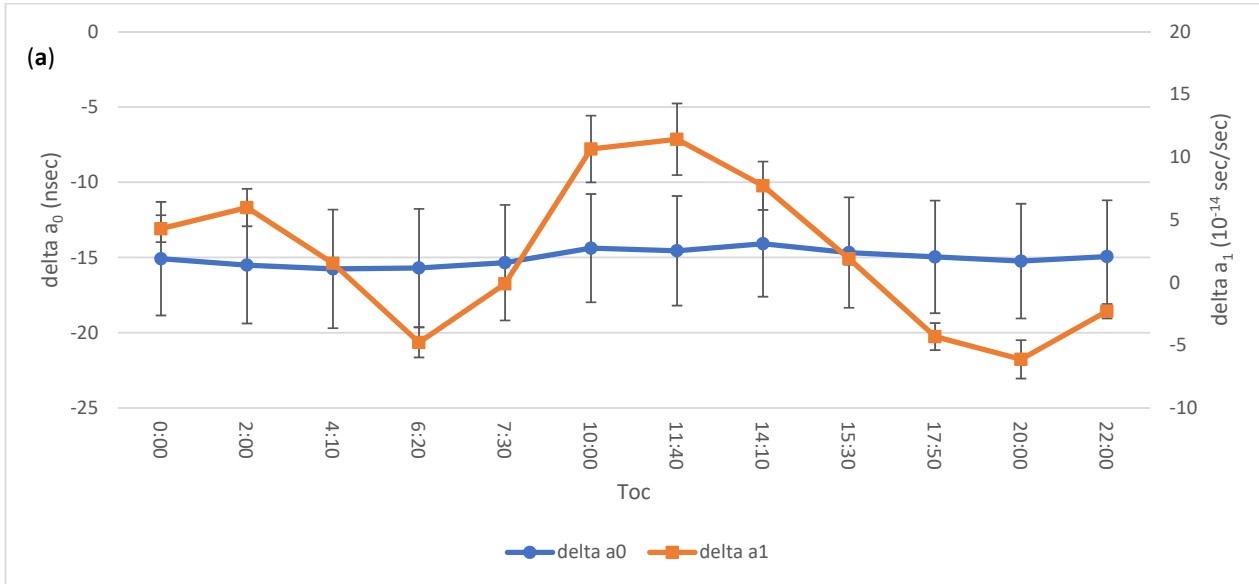

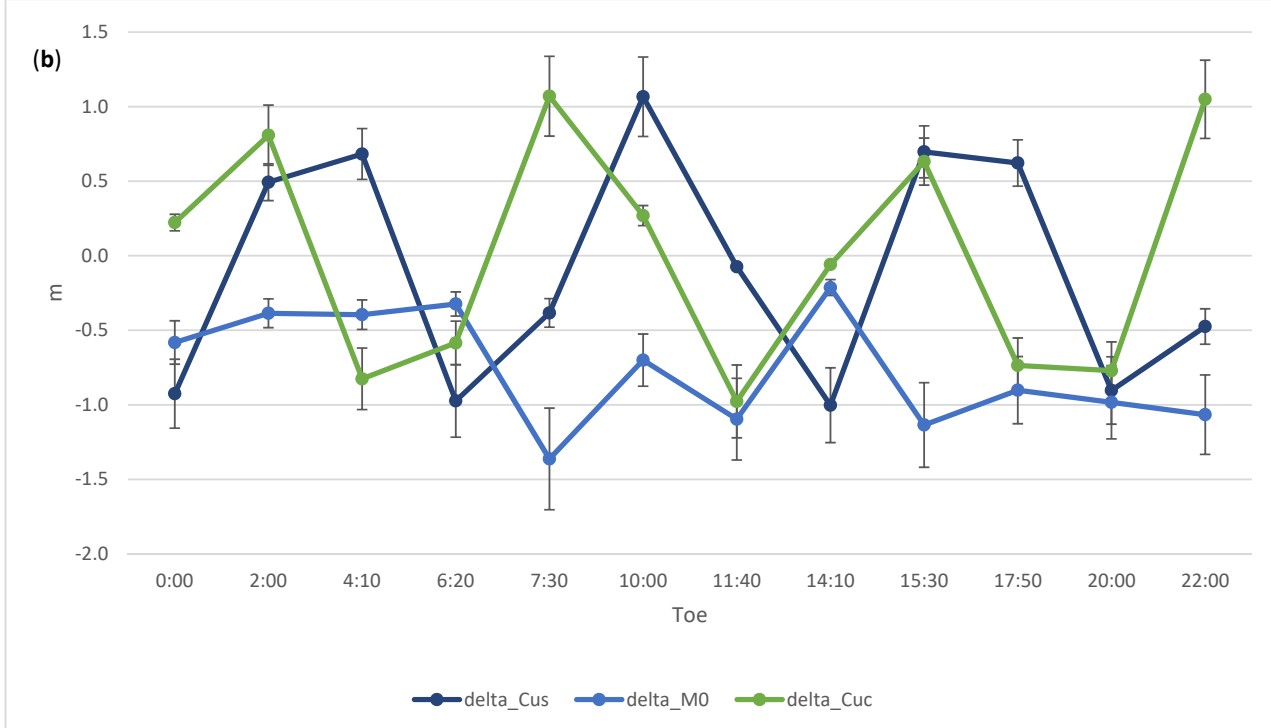

**Figure 5.** *Cont.*

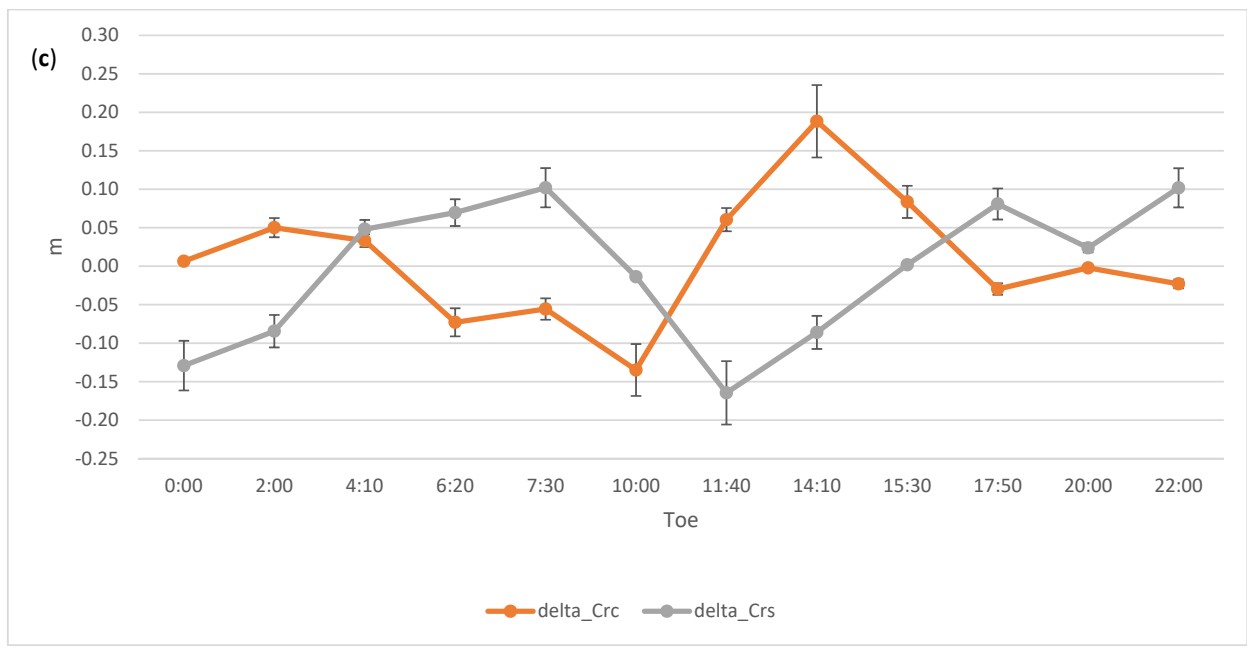

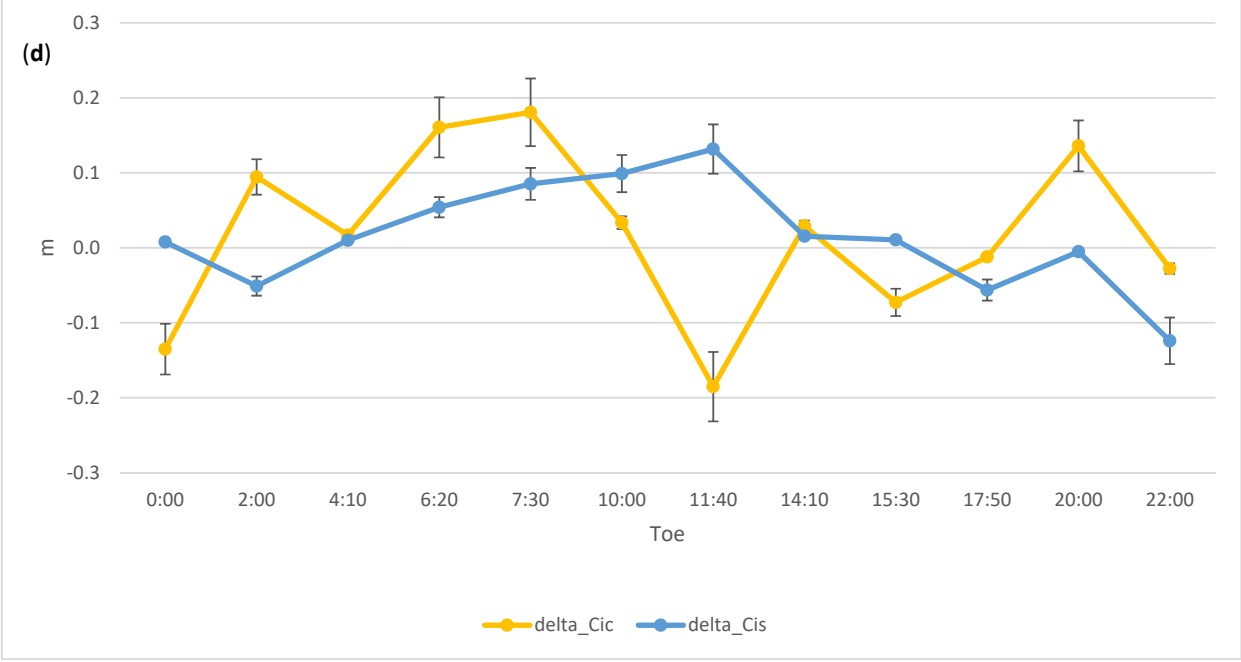

**Figure 5.** Best fitting corrections to (**a**)the clock polynomial parameters, (**b**) Mean anomaly at Toe and amplitude of the cosine and sine perturbations along track due to the second zonal harmonic (Cuc, Cus), and amplitude of the cosine and sine perturbations in the (**c**) radial (Crc, Crs) and (**d**) cross track (Cic, Cis) directions. To convert to radians the scale factor $26 \times 10^6$ m should be used. Error bars are 1 sigma formal uncertainties. Satellite E01 for day 2 January 2020.

**Table 3.** Helmert parameters relating the origin, orientation and scale of the Broadcast reference frame of C07 relative to the IGS14 frame of the SP3 precise ephemeris for SV C07, 2 January 2020. The scale factor accounts for the Center of Mass–Antenna Phase Center correction in the radial direction. The translational parameters were constrained to zero due to lack of geometry in a IGSO orbit.

|                    | Tx (m) | Ty (m) | Tz (m) | Rx (mas) | Ry (mas) | Rz (mas) | Scale |
|--------------------|--------|--------|--------|----------|----------|----------|-------|
| Estimated          | 0.00   | 0.00   | 0.00   | −4.5     | 0.7      | 5.0      | $2.29 \times 10^{-8}$ |
| 1 sigma formal error | 0.00 | 0.00   | 0.00   | 1.70     | 3.20     | 0.06     | $1.73 \times 10^{-9}$ |

Figures 6 and 7 summarize our results for C07. The broadcast ephemeris shows a discrepancy relative to the precise orbit of the order of the meter, and there is a constant bias in the clock corrections (Figure 6). After the fit we observe that the post-fit residuals for both coordinates and clock agree very closely with the corresponding precise ephemeris, suggesting that the correction to the 12 sets of seven parameters each (mean anomaly at epoch Toe; amplitudes and phases of the second zonal components) have successfully accounted for the modeling imperfections of the broadcast message.

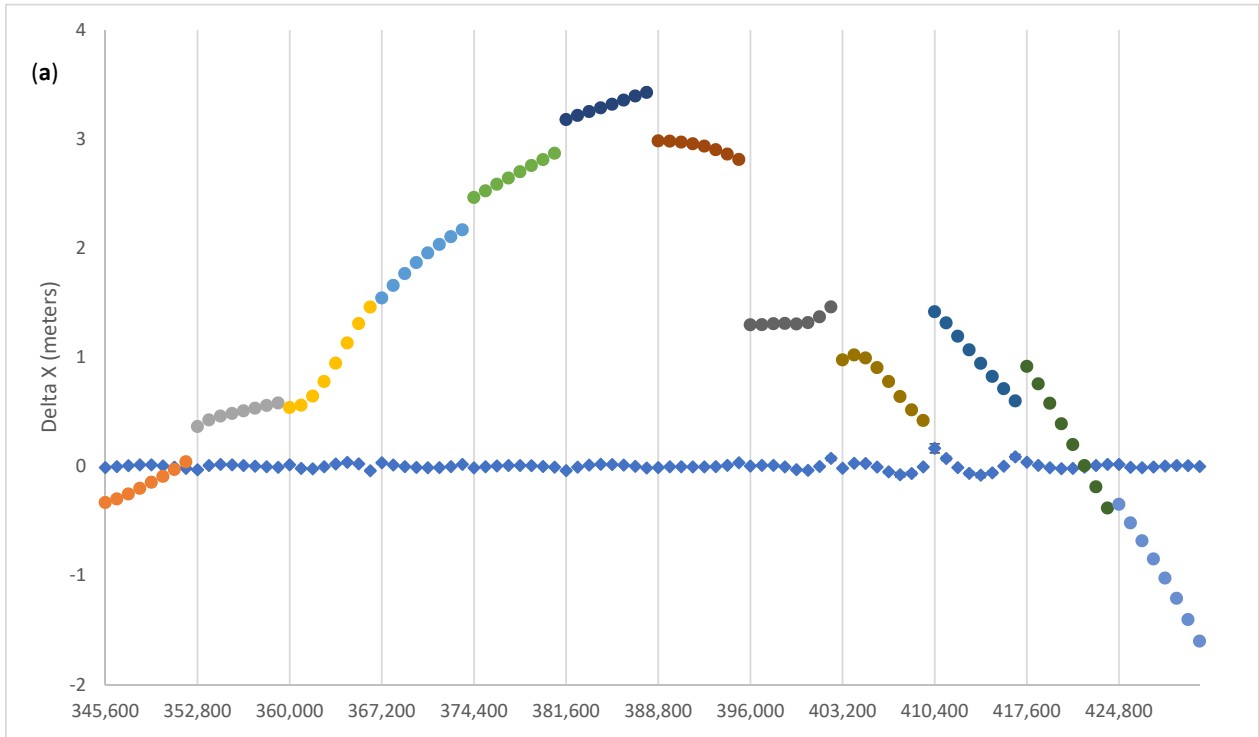

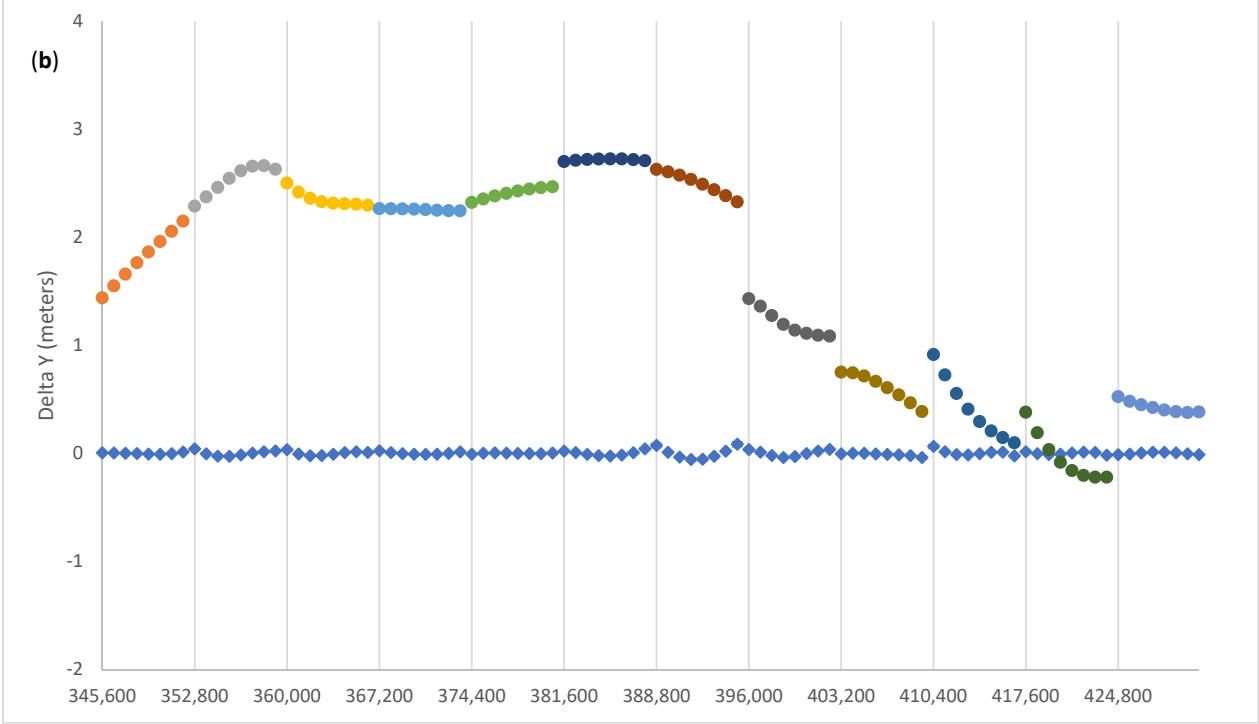

**Figure 6.** *Cont.*

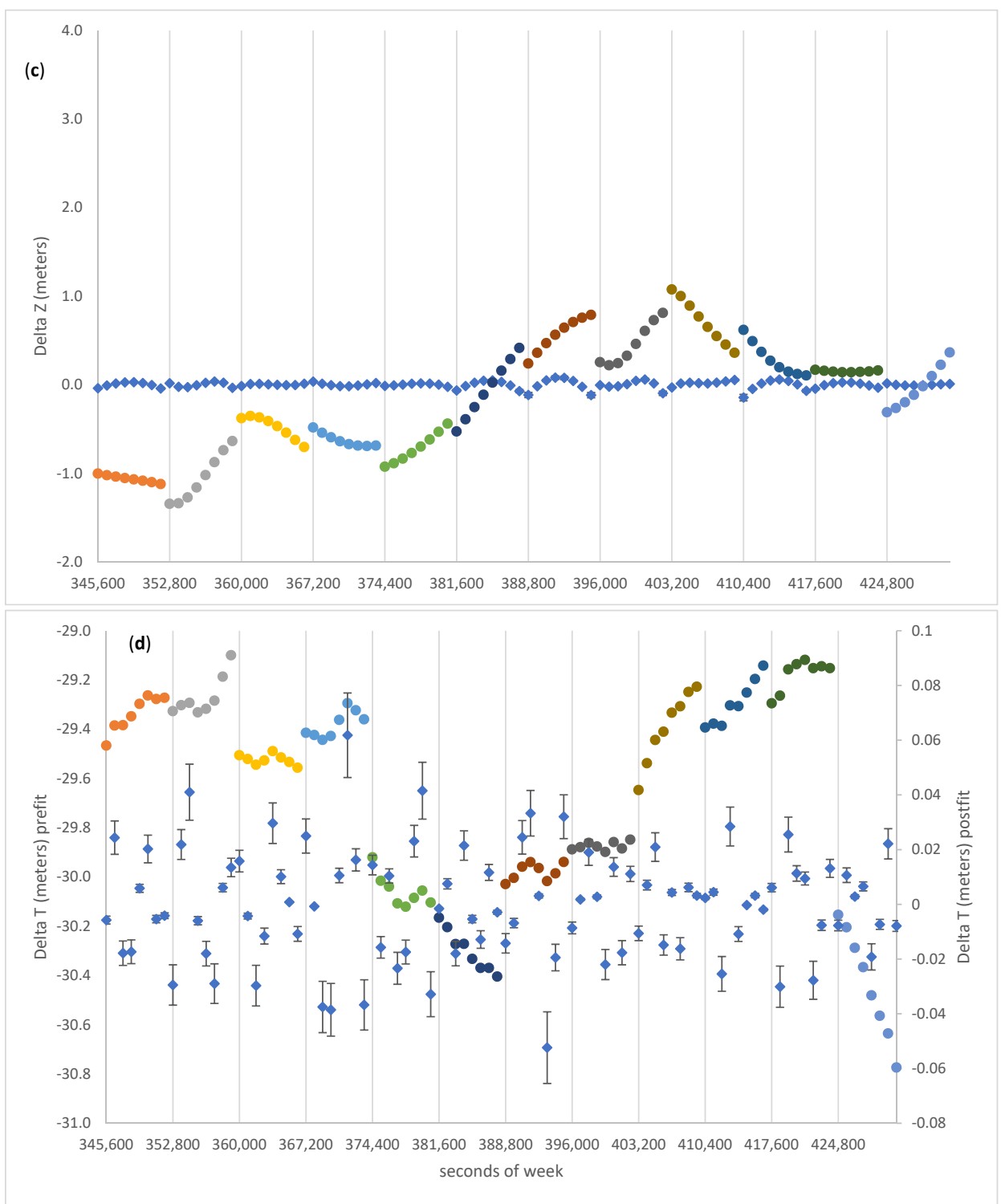

**Figure 6.** Pre-fit (dots, different colours refer to different ephemeris blocks) and post-fit (diamonds) residuals of (**a**) X, (**b**) Y, (**c**) Z for C07. For T (plot (**d**)) pre-fit (dots) refer to the left y-axis and post-fit (diamonds) to the right y-axis.

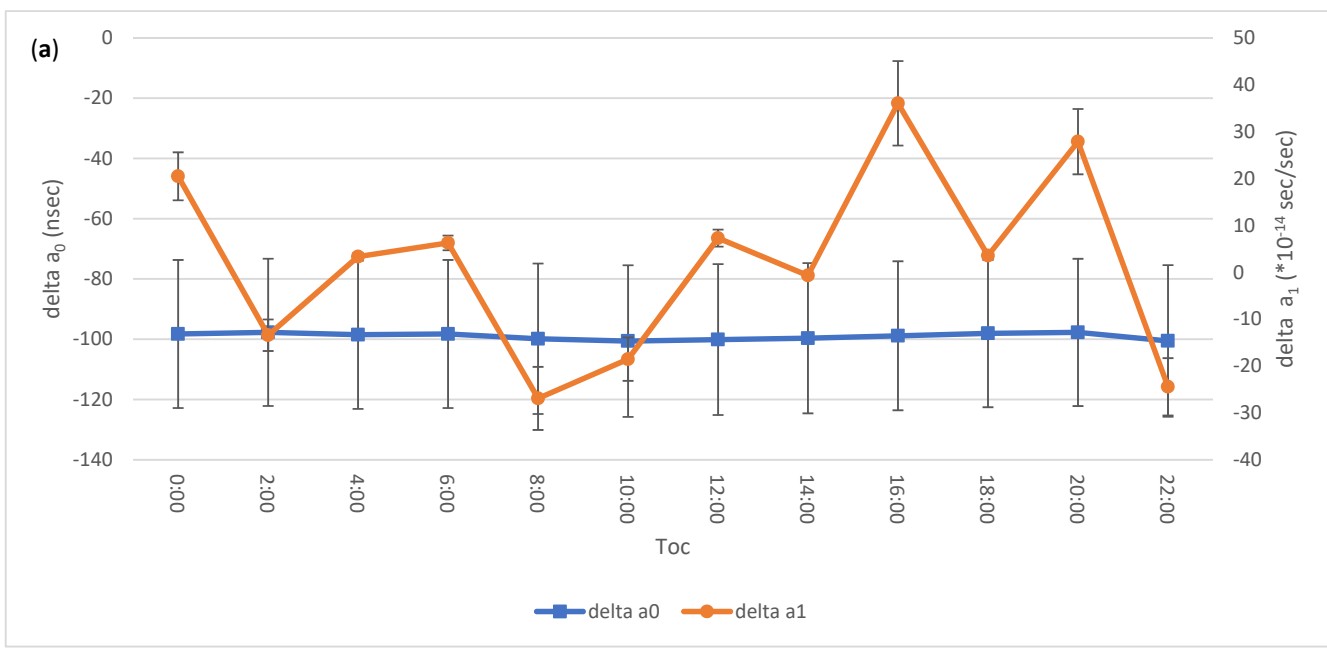

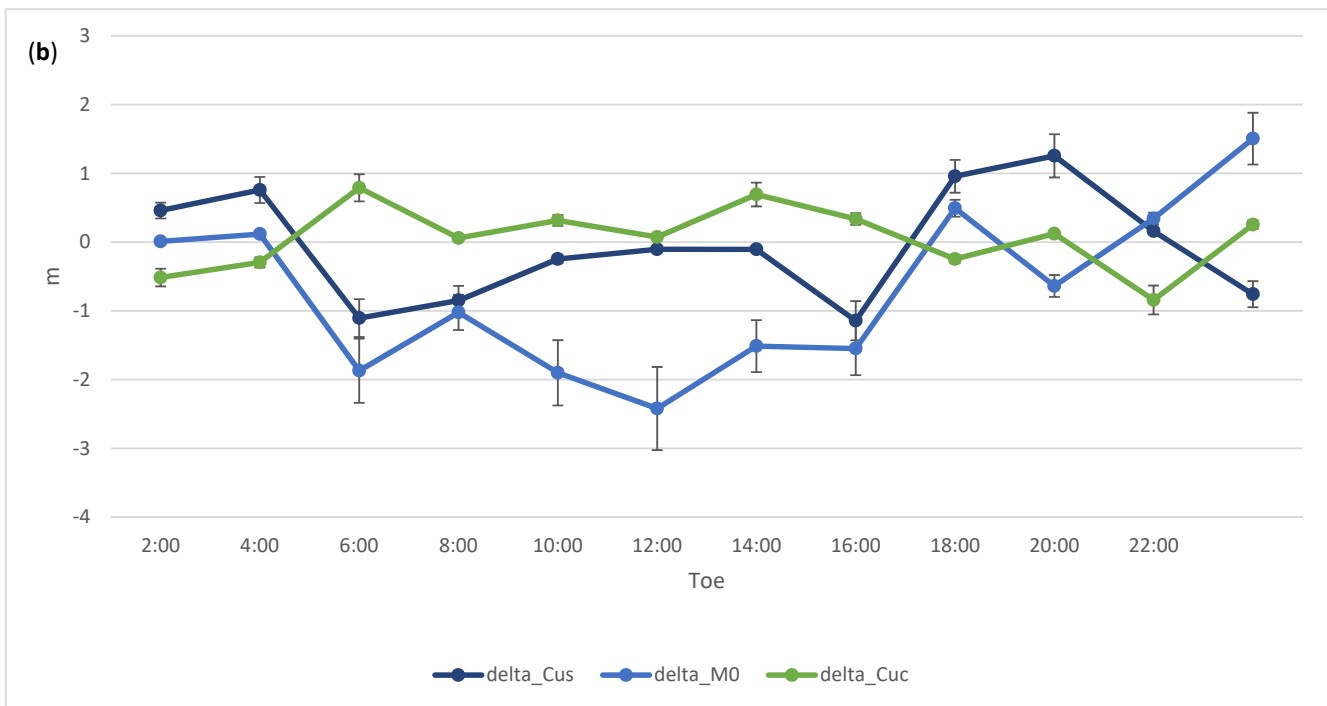

**Figure 7.** *Cont.*

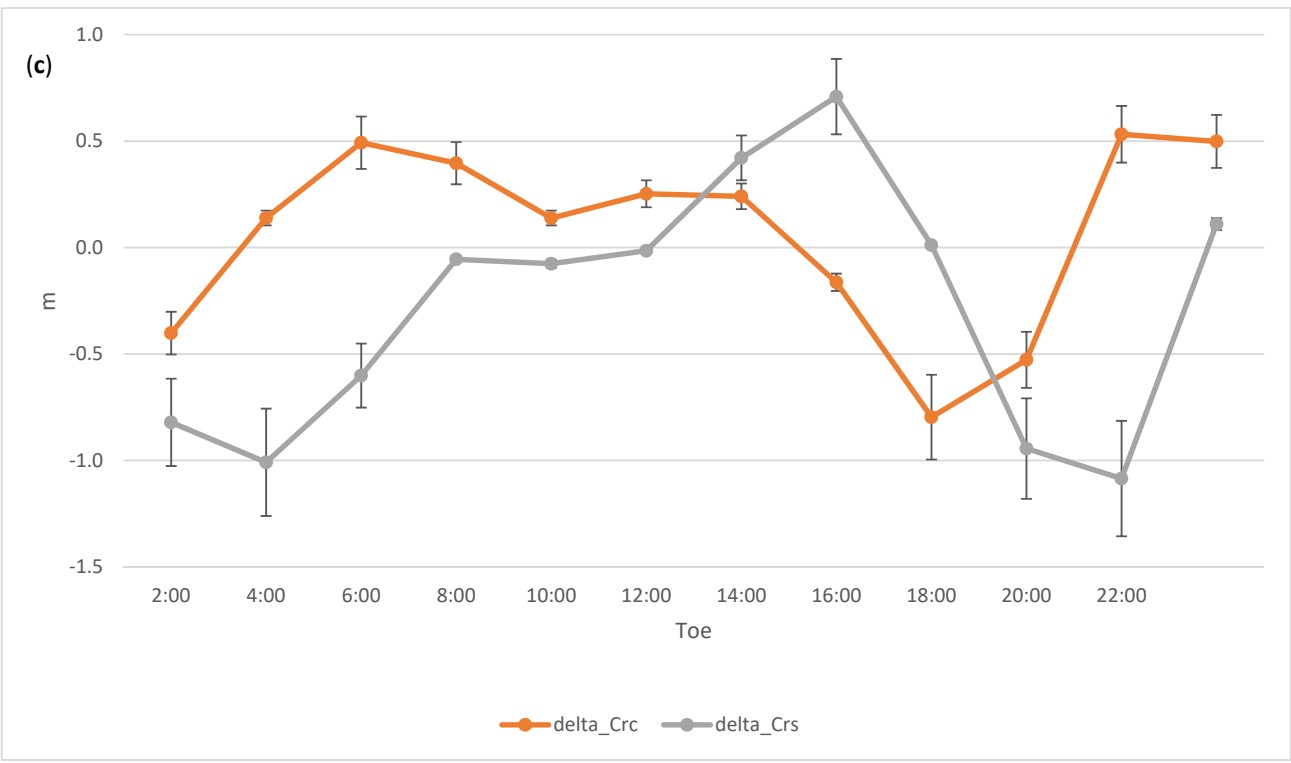

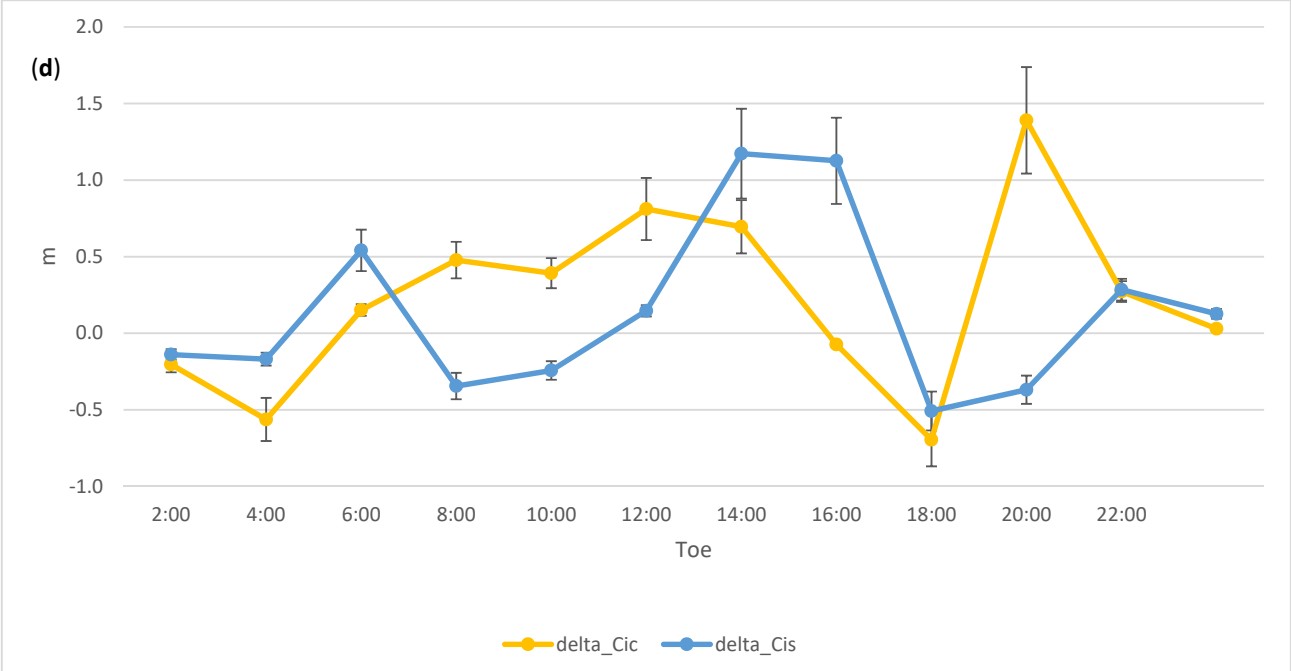

**Figure 7.** (**a**) the clock polynomial parameters, (**b**) Mean anomaly at Toe and amplitude of the cosine and sine perturbations along track due to the second zonal harmonic (Cuc, Cus), and amplitude of the cosine and sine perturbations in the (**c**) radial (Crc, Crs) and (**d**) cross track (Cic, Cis) directions. To convert to radians the scale factor $42 \times 10^6$ m should be used. Error bars are 1 sigma formal uncertainties. Satellite C07 for day 2 January 2020.

The overall improvement for Beidou's C07 in doing the fit described by Equation (1) is summarized as follows: the mean and rms of the pre-fit residuals is $-6.753 \pm 13.310$ m, and $0.000 \pm 0.029$ m for the post-fit residuals. This statistic includes the spatial and temporal coordinates. The large bias in the pre-fit residuals is primarily due to the clock offset of the broadcast message relative to the SP3 values, equivalent to nearly 30 m or $10^{-7}$ s (Figure 7a). The rms of

the post-fit residuals is dominated by the spatial component: the clock polynomials fit with a typical rms of 0.002 m, about a factor of eight smaller than the rms of the spatial components.

### 2.4. Results for Beidou C12

The SV C12 belongs to the part of the Beidou GNSS which is in Medium Earth Orbit (MEO), like GPS. Therefore, it is meaningful to use Equation (1) and model Helmert transformations on a full day arc, and arc parameters (mean anomaly, cosine and sine amplitudes of the periodic perturbations along track, cross track and radial) on two-hour arcs.

Figures 8 and 9 summarize our results for C12. The broadcast ephemeris shows, as for C07, a discrepancy relative to the precise orbit of the order of the meter, and there is a constant bias in the clock corrections (Figure 8). After the fit we observe that the post-fit residuals for both coordinates and clock agree very closely with the corresponding precise ephemeris, suggesting that the correction to the 12 sets of seven parameters each (mean anomaly at epoch Toe; amplitudes and phases of the second zonal components) plus the seven Helmert parameters have successfully accounted for the modeling imperfections of the broadcast message.

The overall improvement for Beidou's C12 in doing the fit described by Equation (1) is summarized as follows: the mean and rms of the pre-fit residuals is $-5.263 \pm 9.351$ m, and $0.000 \pm 0.046$ m for the post-fit residuals. This statistic includes the spatial and temporal coordinates. The large bias in the pre-fit residuals is, similarly to C07, primarily due to the clock offset of the broadcast message relative to the SP3 values.

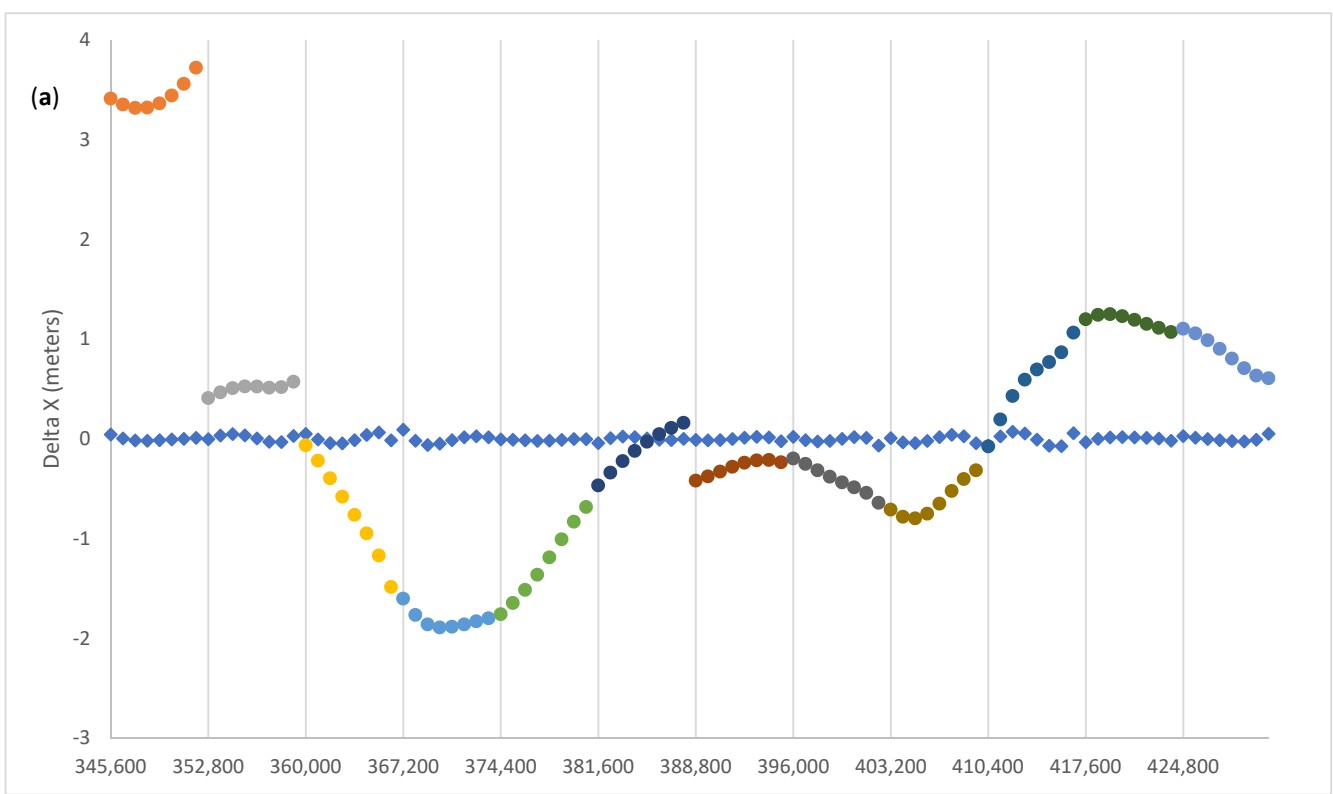

**Figure 8.** *Cont.*

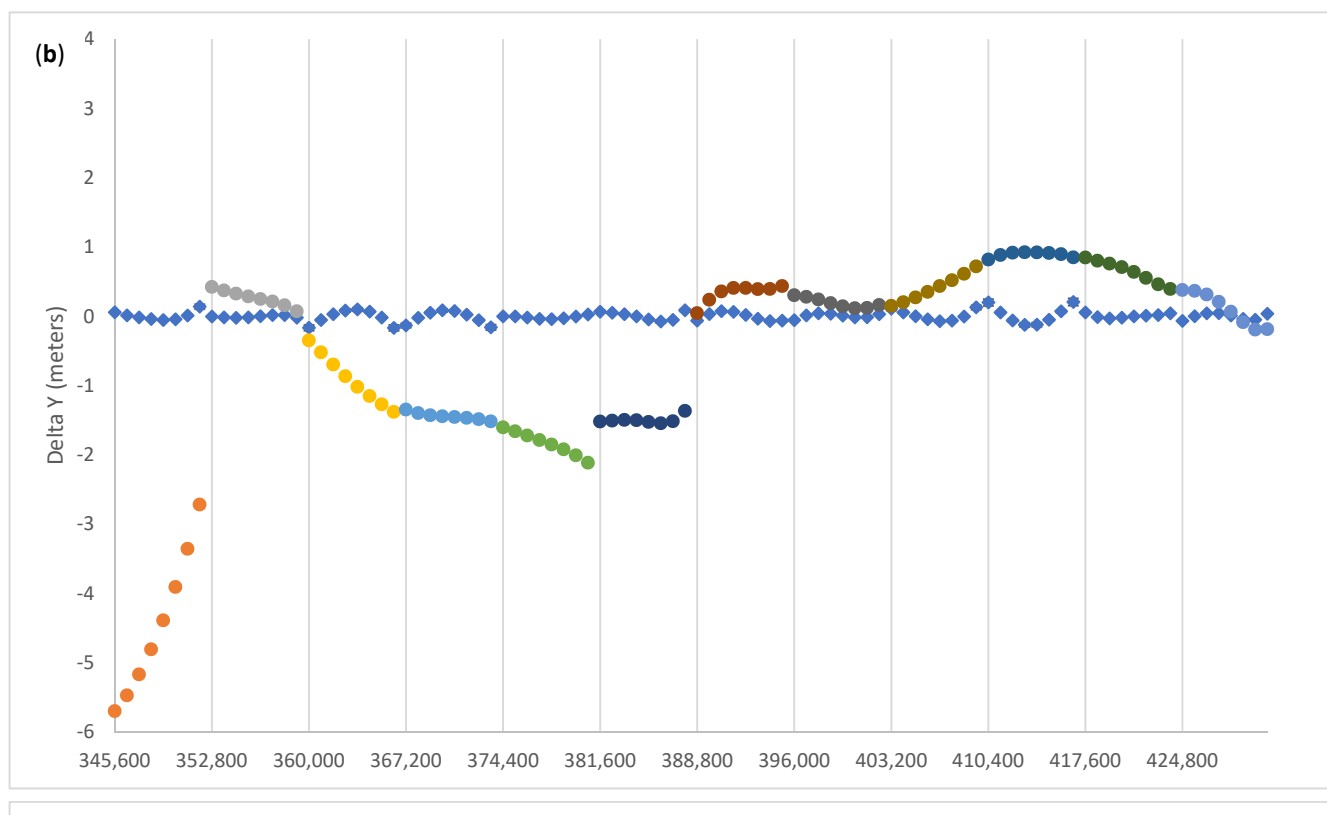

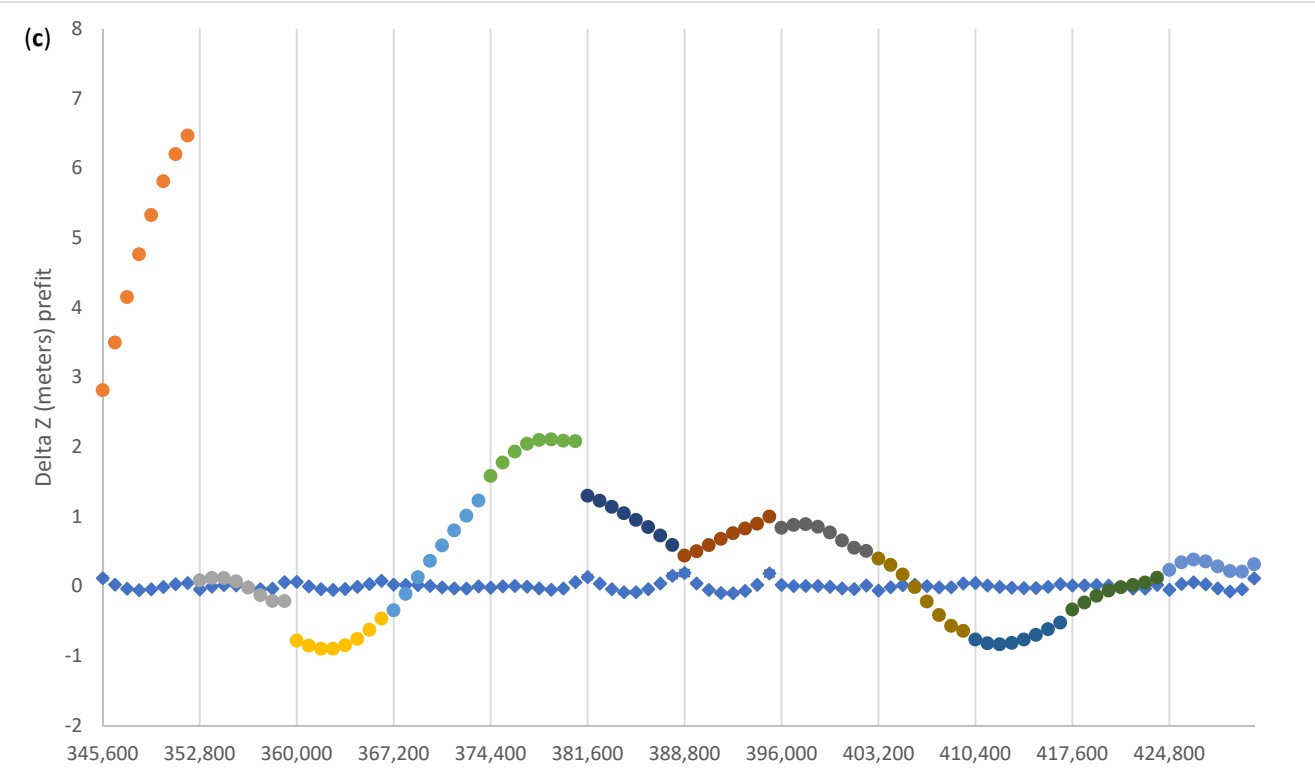

**Figure 8.** *Cont.*

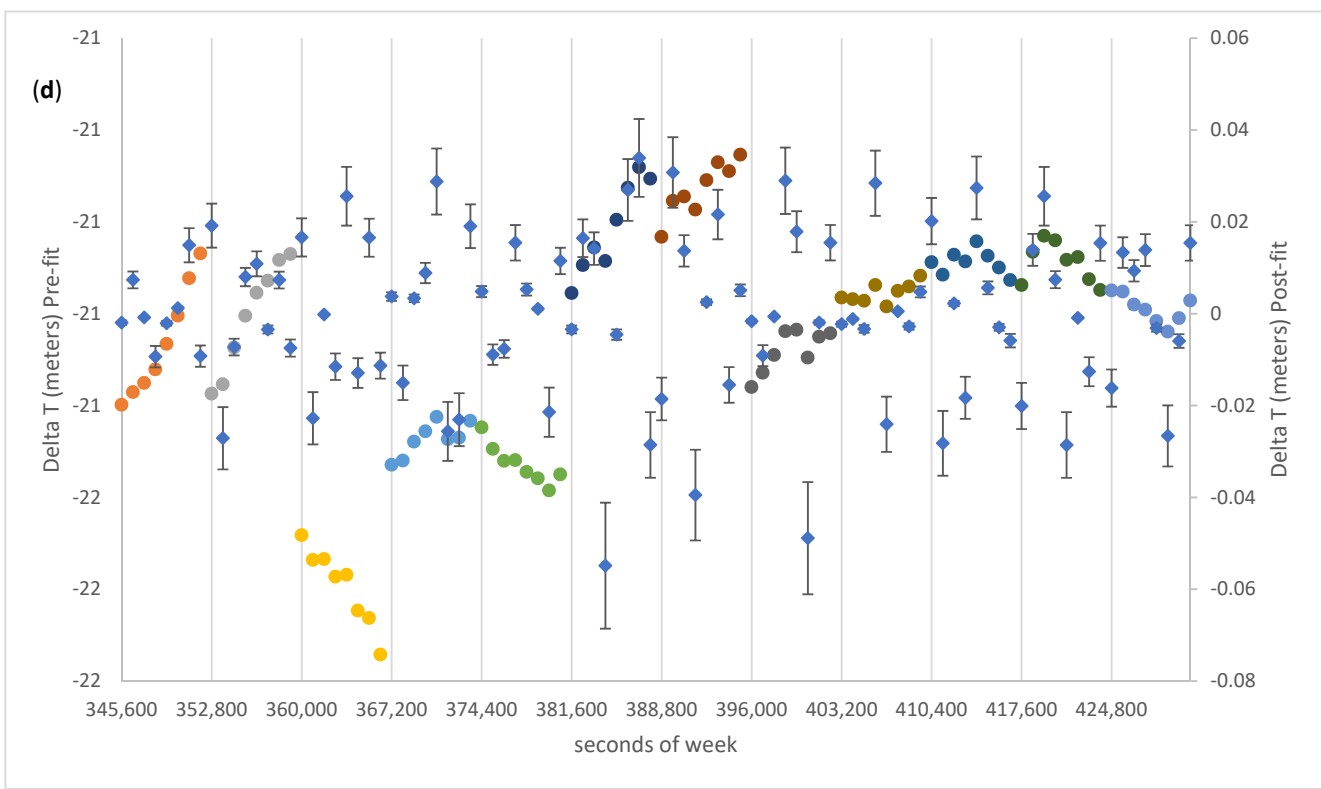

**Figure 8.** Pre-fit (dots, different colours for different ephemeris blocks) and post fit (diamonds) residuals of (**a**) X, (**b**) Y, (**c**) Z, and (**d**) T for C12. Post-fits of the time offset T (diamonds) are plotted on the right y-axis.

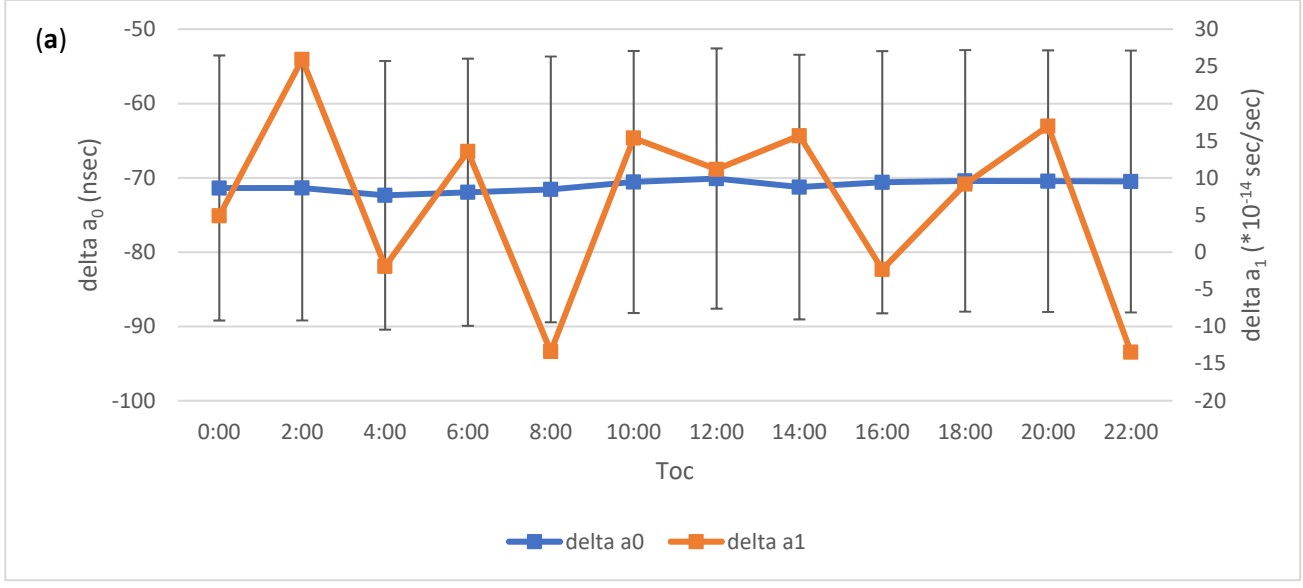

**Figure 9.** *Cont.*

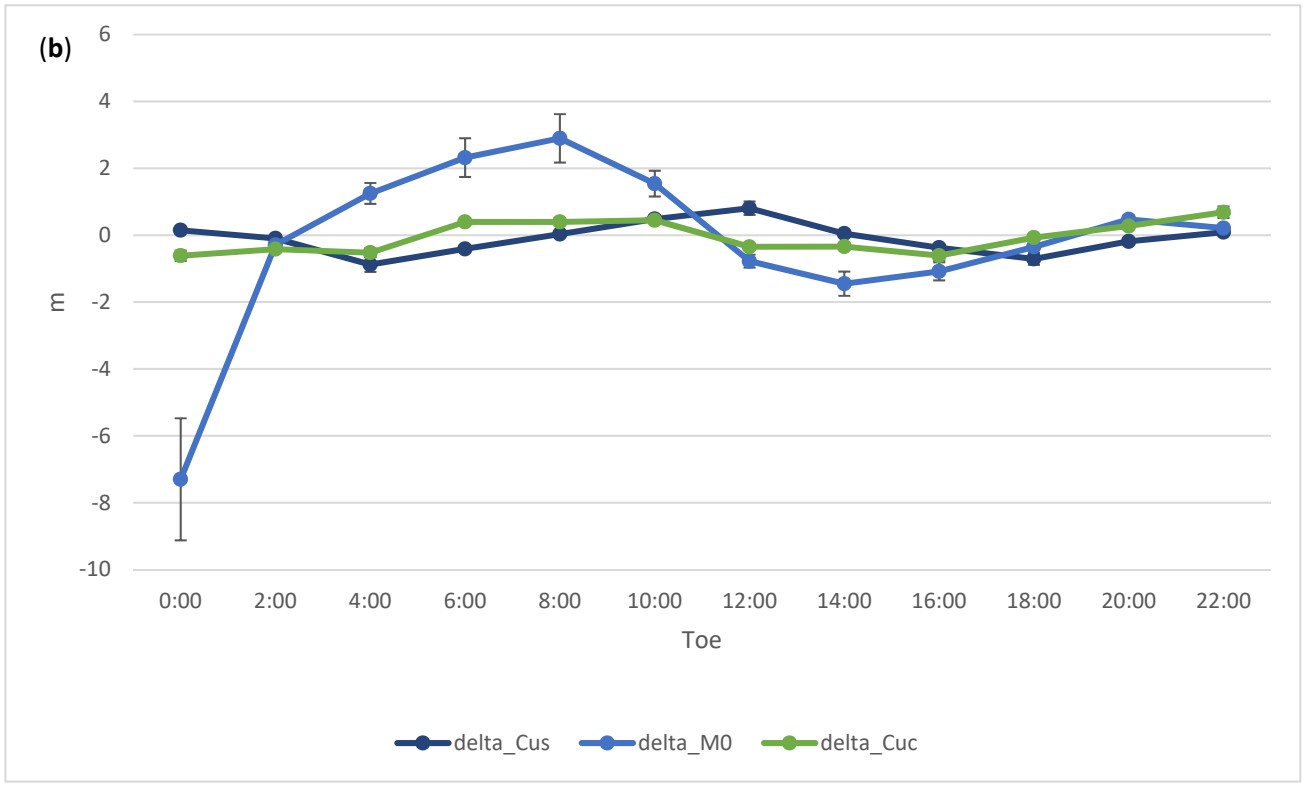

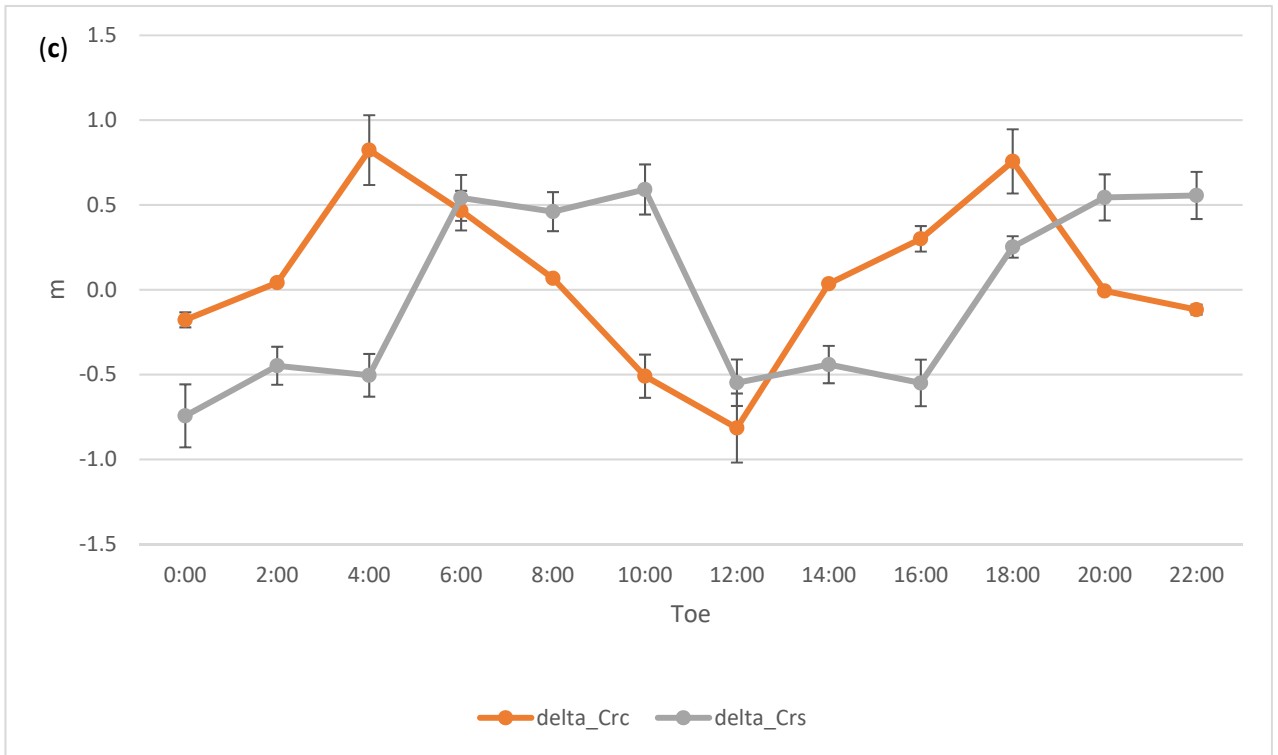

**Figure 9.** *Cont.*

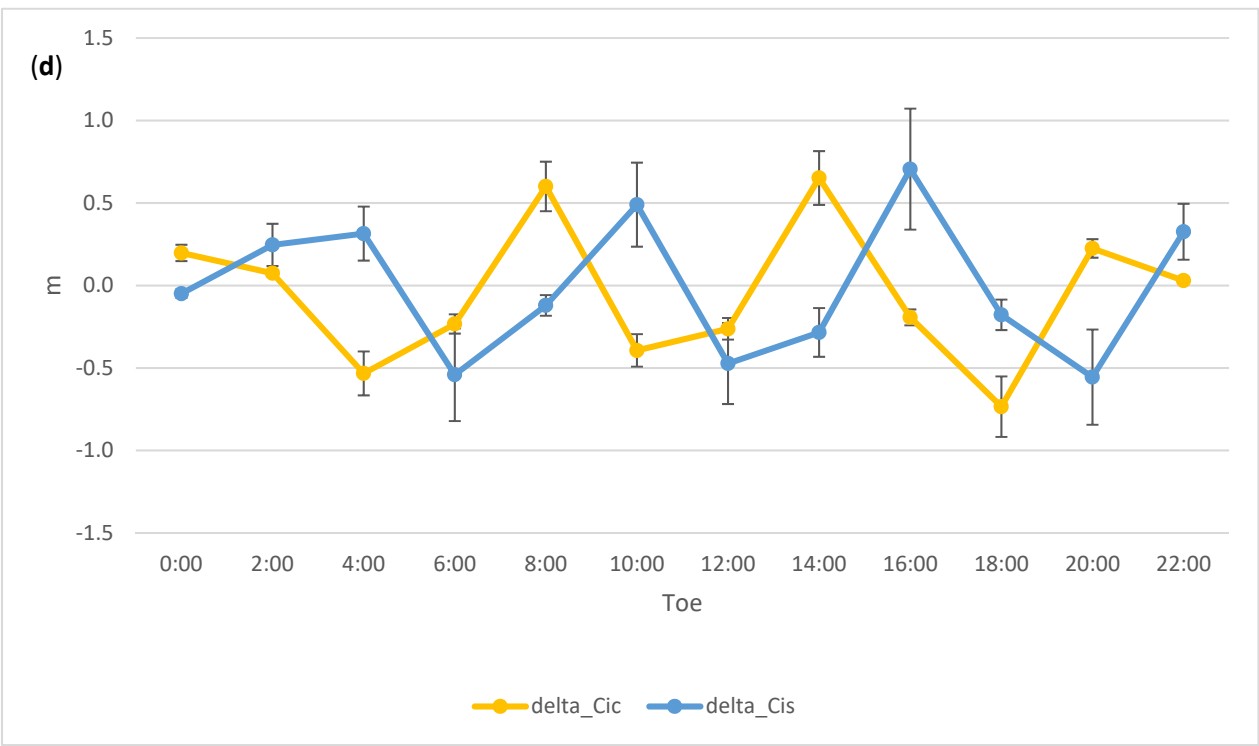

**Figure 9.** Best fitting corrections to (**a**) the clock polynomial parameters, (**b**) Mean anomaly at Toe and amplitude of the cosine and sine perturbations along track due to the second zonal harmonic (Cuc, Cus), and amplitude of the cosine and sine perturbations in the (**c**) radial (Crc, Crs) and (**d**) cross track (Cic, Cis) directions. To convert to radians the scale factor $26 \times 10^6$ m should be used. Error bars are 1 sigma formal uncertainties. Satellite C12 for day 2 January 2020.

Table 4 summarizes the seven Helmert parameters estimated for C12 on a daily basis. There is a clear shift of the origin in the z direction of ca. 0.7 m, and a smaller one in the x direction. Again, this can be interpreted as a good indication that our optimization process does require an adjustment of the reference frame parameters, if centimetric rms of the post-fit residuals relative to a precise ephemeris is the goal.

**Table 4.** Helmert parameters relating the origin, orientation and scale of the Broadcast reference frame of C12 relative to the IGS14 frame of the SP3 precise ephemeris for SV C12, 2 January 2020. The scale factor accounts for the Center of Mass–Antenna Phase Center correction in the radial direction.

|  | Tx (m) | Ty (m) | Tz (m) | Rx (mas) | Ry (mas) | Rz (mas) | Scale |
|---|---|---|---|---|---|---|---|
| Estimated | 0.26 | 0.02 | −0.70 | −0.6 | 1.9 | −2.8 | $1.25 \times 10^{-8}$ |
| 1 sigma formal error | 0.06 | 0.05 | 0.02 | 0.53 | 0.69 | 0.12 | $1.17 \times 10^{-9}$ |

## 3. Mathematical Model and Results for Glonass

In the Glonass navigation message, the clock corrections of the satellite time to the Glonass time is, as is for GPS, expressed in terms of a second order polynomial. The instantaneous ECEF position and velocity of a Glonass satellite is instead computed by numerically integrating the equations of motion. These are conveniently formulated in terms of six ordinary differential equations of first order. The vector of state of the satellite (position and velocity) is therefore broadcast at a reference epoch Toe. The numerical integrator, normally a 4th order Runge–Kutta, maps this vector of state from time Toe to any other epoch within the validity interval of the message. The force field consists of the gradient of the Earth's gravitational potential truncated to the second zonal harmonic $J_2$ plus the centrifugal and Coriolis terms, since the equations of motion are integrated in a rotating frame. Glonass broadcasts three additional terms, the Lunisolar accelerations, which

are constant accelerations during the validity time of the broadcast message, normally 30 min.

Tuning the broadcast parameters on a precise orbit requires therefore the adjustment of the clock and vector of state in arcs of at least 30 min. In the sample broadcast ephemeris data set analyzed here (https://igs.bkg.bund.de/root_ftp/MGEX/BRDC_v3/2020/002/brdm0020.20p.Z, accessed on 10 October 2021), the clock drift and clock drift rate (*a*1, *a*2) were zero, and the lunisolar accelerations were allowed to change in fixed increments. It is also important to note that the Glonass message constrains the terrestrial reference frame by providing ECEF initial coordinates and velocities. Therefore, for Glonass it is not necessary to estimate Reference Frame parameters. For the CoM to APC correction the z-bias of 2.450 m appropriate for Glonass M was used (https://files.igs.org/pub/station/general/pcv_archive/igs14_2056.atx, accessed on 10 October 2021).

To tailor the broadcast parameters on a precise ephemeris and clock, we formulate the minimum variance algorithm as follows:

$$\sum_{i=1}^{96}[\Delta(XYZT)]^2 = f(a0, a1, a2, X_0, \dot{X}_0, \ddot{X}_{LS}, Y_0, \dot{Y}_0, \ddot{Y}_{LS}, Z_0, \dot{Z}_0, \ddot{Z}_{LS}) = min \qquad (2)$$

We minimize the sum of the $96 \times 4 = 384$ square discrepancies $\Delta(XYZT)$ between the SP3 precise values and those computed with the broadcast message, using as adjustable parameters the three clock terms *a*0, *a*1 and *a*2, and a 9D vector of state containing position and velocities at epoch Toe, and three constant accelerations. If the 24-h interval is broken into 24 consecutive arcs each of one hour, then we have to adjust 288 parameters. These become 144 if we test arcs of a 2-h duration. The Time of clock in the clock polynomial and the Time of ephemeris are taken coincident.

Contrary to the GPS-like approach, where the arc terms were coupled by the global terms, i.e., the Helmert parameters (Figure 1), the partial derivative matrix set up to linearize Equation (2) is strictly block diagonal, with no correlation between arc parameters of different arcs (Figure 10).

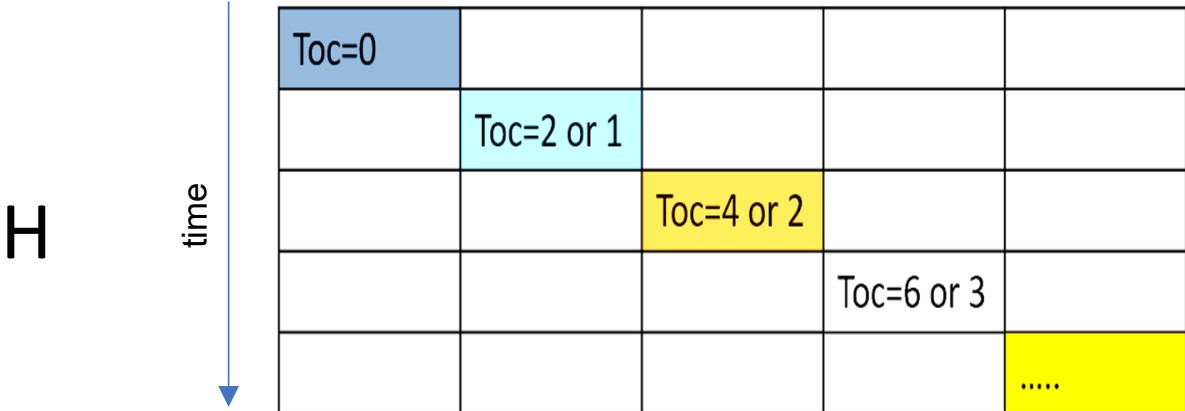

**Figure 10.** Structure of the partial derivative matrix H of the pre-fit discrepancies relative to the arc parameters. The arc parameters are indexed with the time Toc (Toe is equivalent) and the nominal values are dependent on the individual arcs being 2 h or 1 h long.

We consider on 2 January 2020, satellite R01 and precise ephemeris and clock computed by CNES (https://cddis.nasa.gov/archive/gnss/products/mgex/2086/GRG0MGXFIN_20200020000_01D_15M_ORB.SP3.gz, accessed on 10 October 2021). The results of the parameter optimization for this test case are shown in Figures 11 and 12. Figure 11 suggests that also in the example of Glonass it is possible to fine tune the broadcast model parameters so that the positions and clock corrections computed with the adjusted parameters very closely fit the reference CNES final orbit of R01. The corrections to the broadcast parameters are shown

graphically in Figure 12. Besides the rms of 0.024 m as an indication of the final accuracy, it should be mentioned as an additional benefit that the interval of validity of the corrected broadcast ephemeris has been doubled to 1 h. Tests with 2-h arcs resulted in rms spreads considerably larger. We propose 1 h as a reasonable compromise between accuracy and refresh rate.

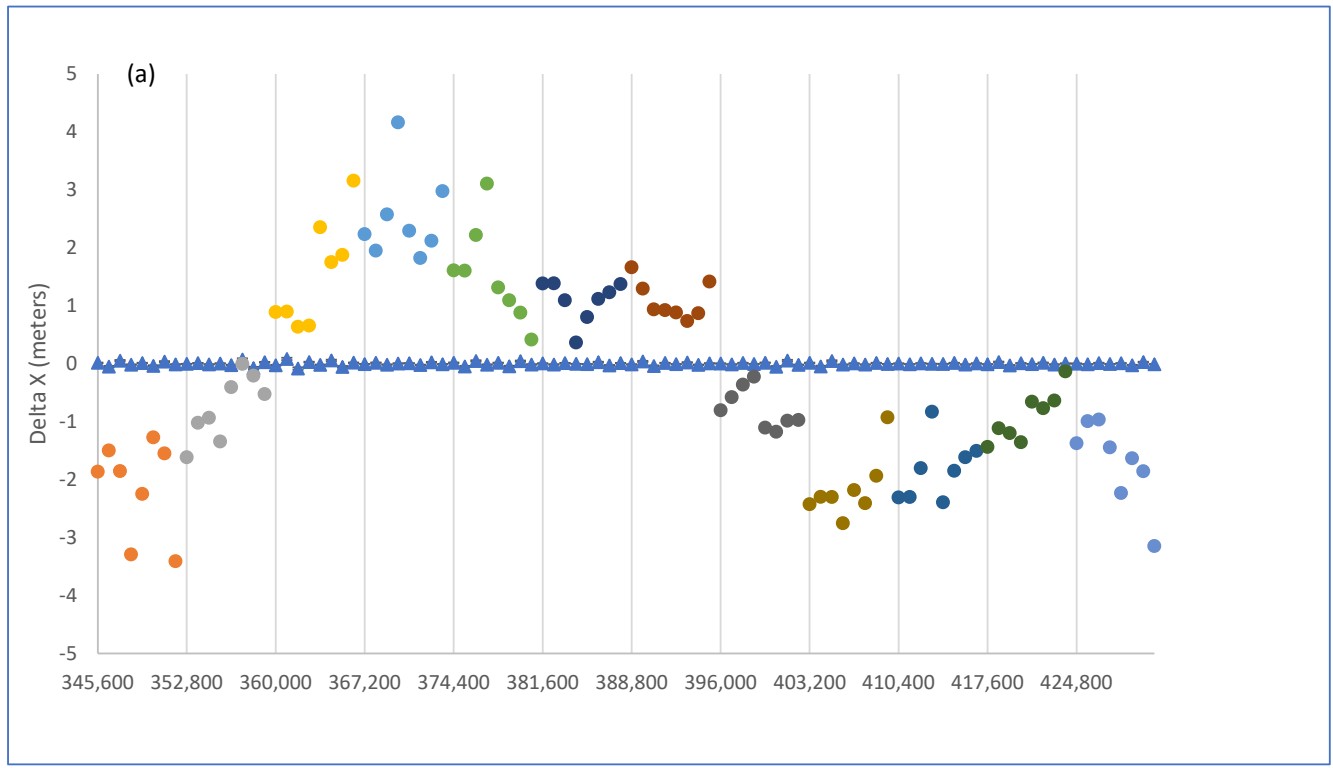

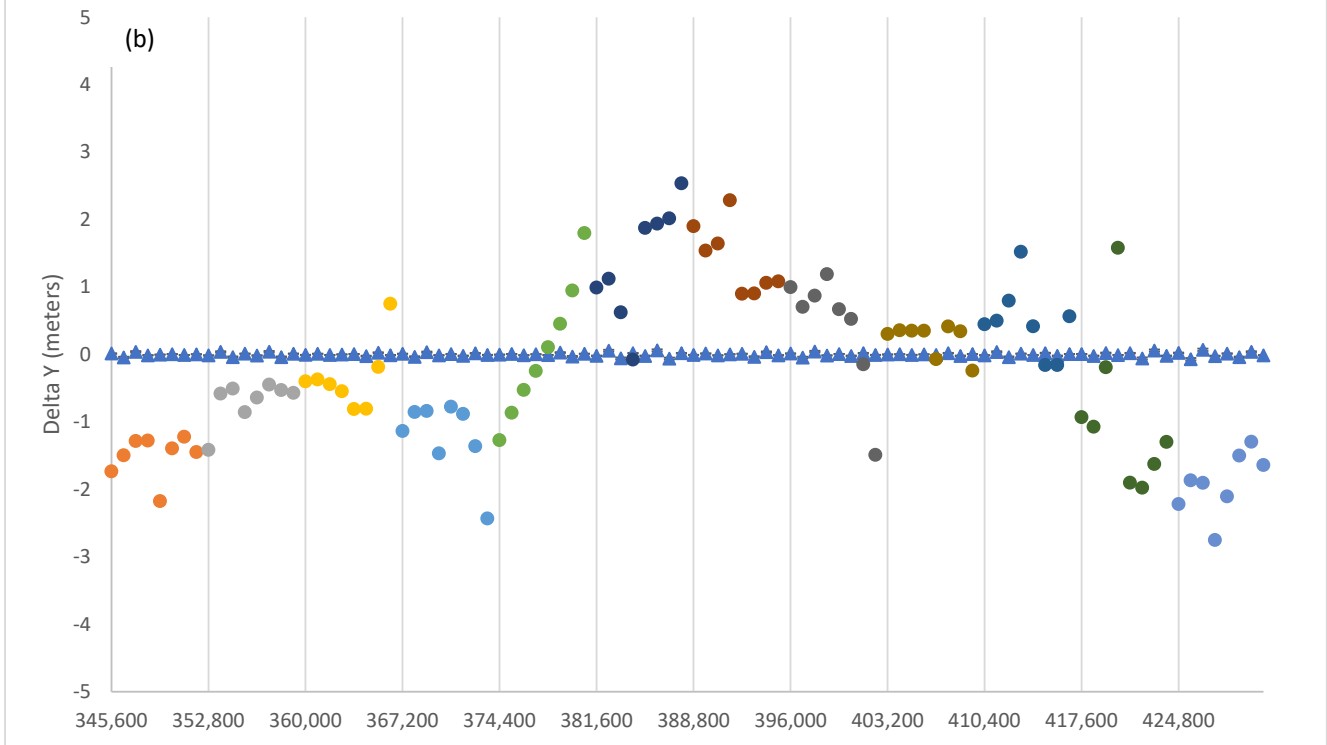

**Figure 11.** *Cont.*

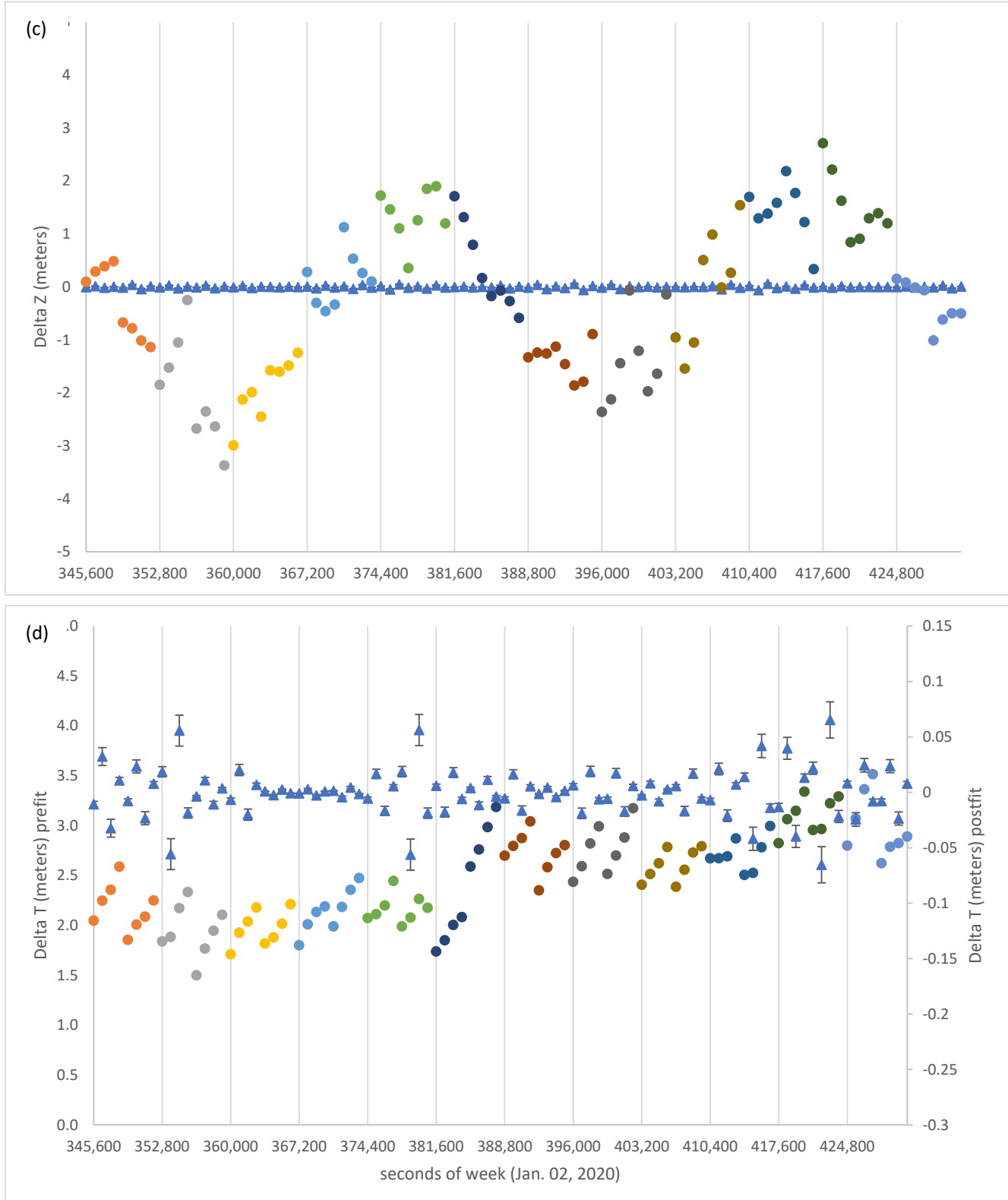

**Figure 11.** Pre-fit (dots) and post-fit (diamonds) residuals of (**a**) X, (**b**) Y, (**c**) Z, and (**d**) T for R01. Length of arc for fit is 1 h. Different colours denote different ephemeris blocks.

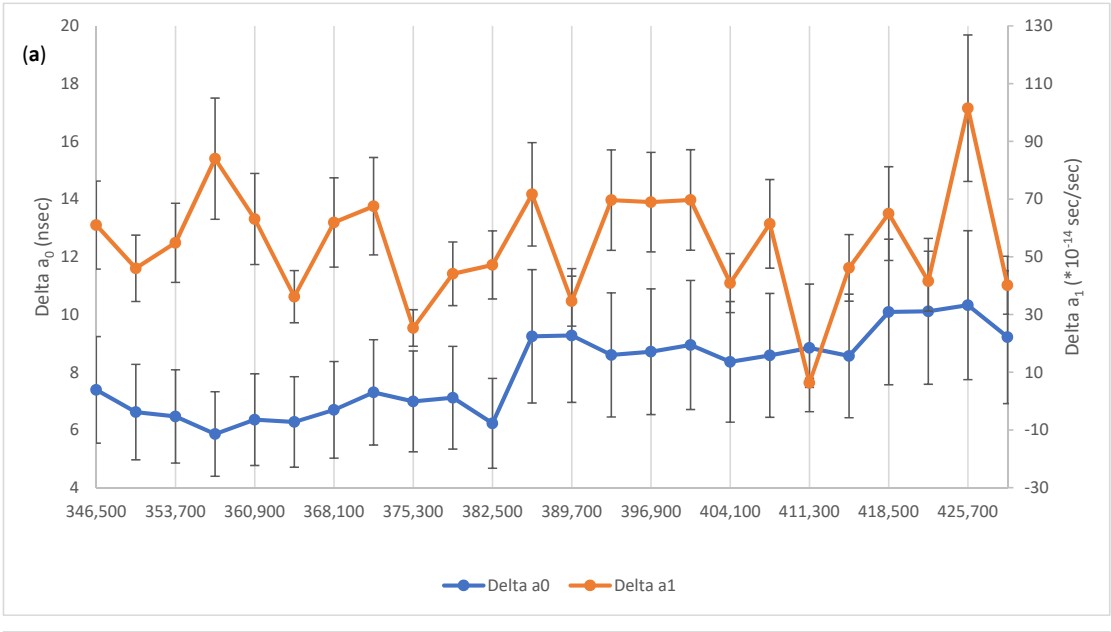

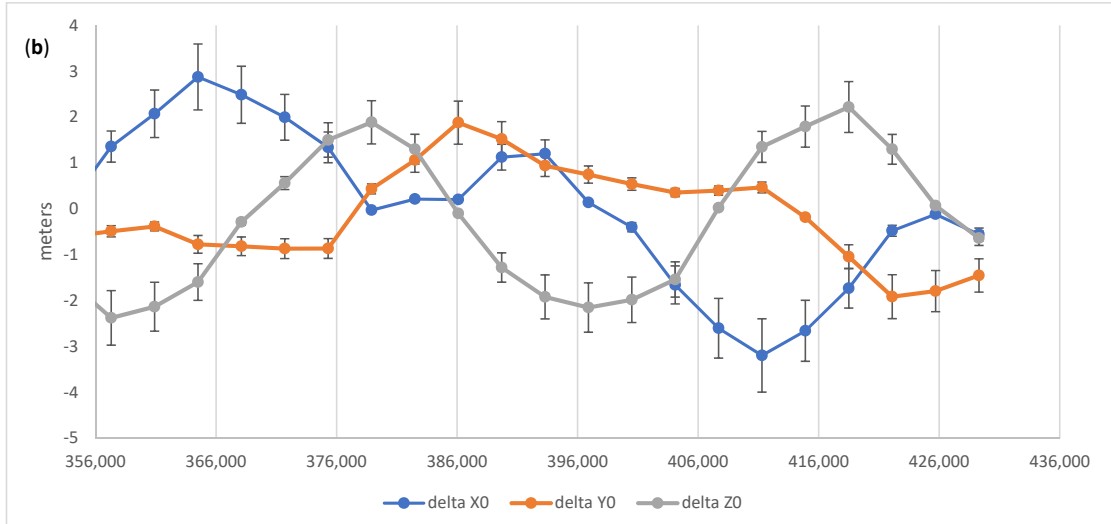

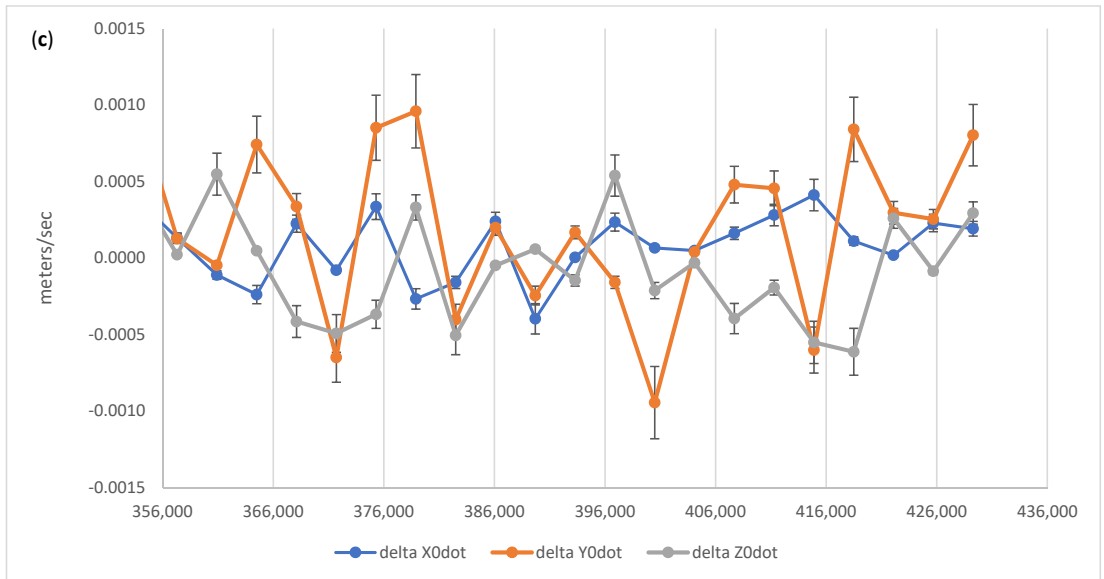

**Figure 12.** *Cont.*

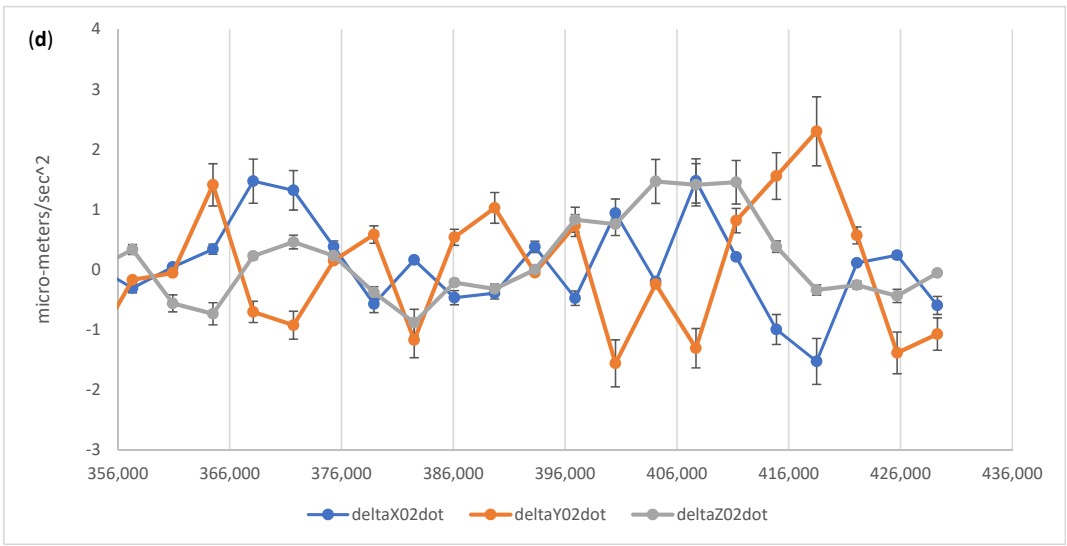

**Figure 12.** Corrections to the broadcast (**a**) clock, (**b**) positions, (**c**) velocities, and (**d**) lunisolar accelerations for R01 using CNES precise orbits as reference. The vector of state (3 clock parameters + 9 orbit parameters) is estimated at intervals of 1 h. Rate of clock drift ($a_2$) was computed but is not shown.

## 4. Polynomial Clock Model vs. IGS/MGEX High Rate Clocks

The clock polynomials we have computed for GPS, Galileo, Beidou and Glonass fit with centimetric rms the clock values tabulated at 15 min intervals in the SP3 files. For most applications, the polynomial clock corrections are used to interpolate the values at rates of the order of 1 Hertz or higher. In [2], it is pointed out that noise in rubidium, cesium and hydrogen maser clocks is characterized by a random walk phase modulation. For lag times of up to 10 s, the Allan variance is between $10^{-11}$ and $10^{-12}$ for the Cesium or Rubidium clocks. For Galileo's Passive Hydrogen Maser, the frequency stability is somewhat smaller than $10^{-12}$ [18,19,24–26]. This means that two consecutive, non-overlapping time segments of 100 s nominal length will have a 1 sigma difference of 1 ns to 0.1 ns for a two-sample Allan stability of $10^{-11}$ and $10^{-12}$ respectively. It follows that our clock polynomials need to be tested against high rate clock estimates. The IGS/MGEX makes available such files with sampling rate of 30 s. In the rest of this section, we will therefore compute differences in the clock corrections, between our polynomial values based upon 15 min sampling and high rate IGS clocks based on 30 s sampling.

Figure 13 gives a comparative example of IGS/MGEX high rate clocks (30 s sampling) and the predictions of our polynomials based on best fit to 15 min data (http://ftp.aiub.unibe.ch/CODE_MGEX/CODE/2020/COM20864.CLK.Z for Beidou and https://cddis.nasa.gov/archive/gnss/products/mgex/2086/GRG0MGXFIN_20200020000_01D_30S_CLK.CLK.gz for GPS, Glonass and Galileo, accessed on 10 October 2021). Figure 13 indicates that the rms discrepancy is smaller than 0.1 ns for GPS, Galileo and Beidou, while for Glonass we have a factor 10 worse rms. Several departures from random noise are well visible, suggesting that our clock polynomials smooth the high frequency part of the noise. The differences are on average of up to few equivalent cm or less. This implies that an overall figure of merit for our improved broadcast model of a few cm is the accuracy level we can expect, even at high sampling rates. These rms estimates are within the consistencies among the various MGEX solutions for clocks and orbits, as discussed in [16].

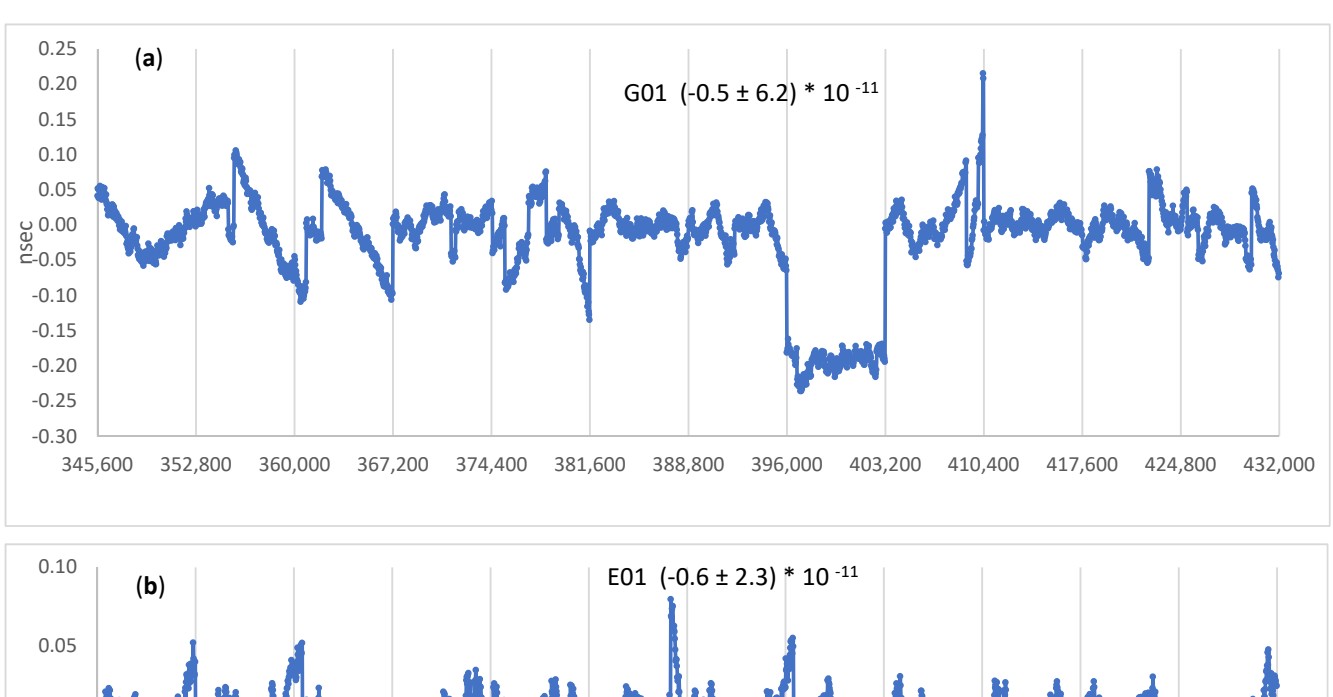

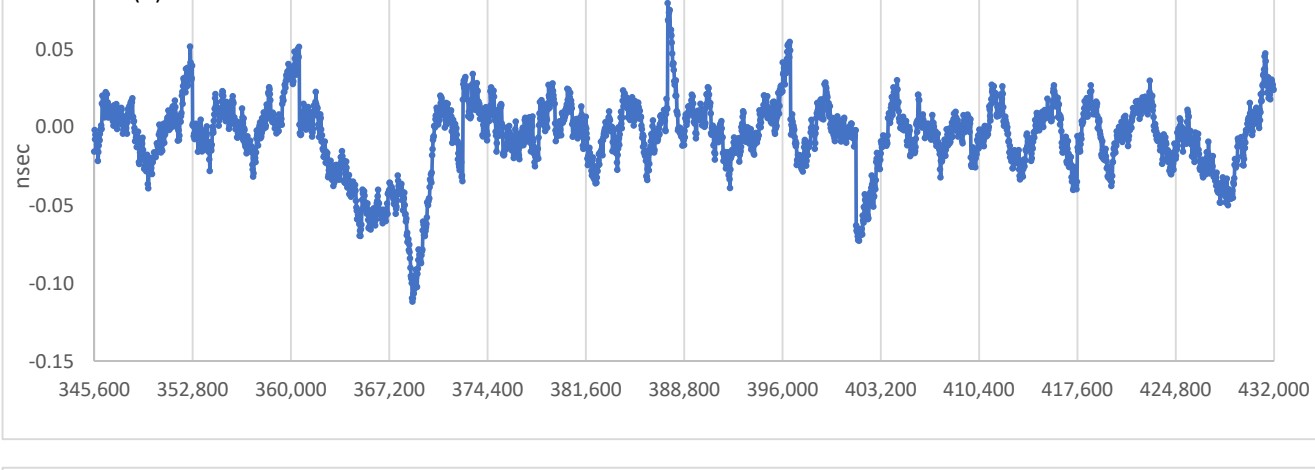

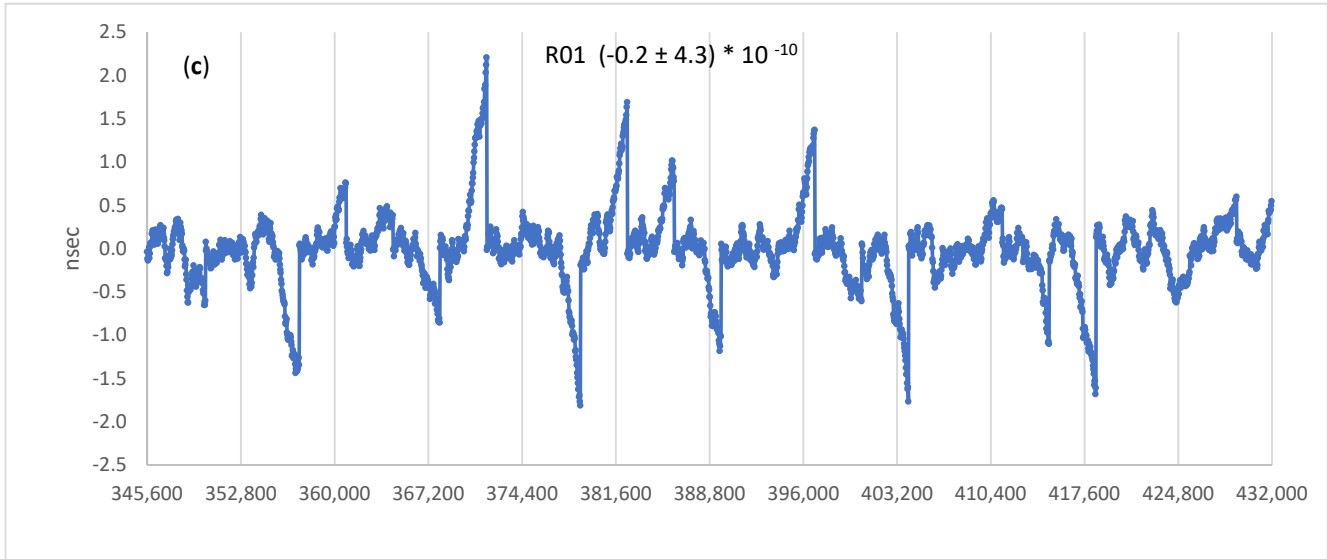

**Figure 13.** *Cont.*

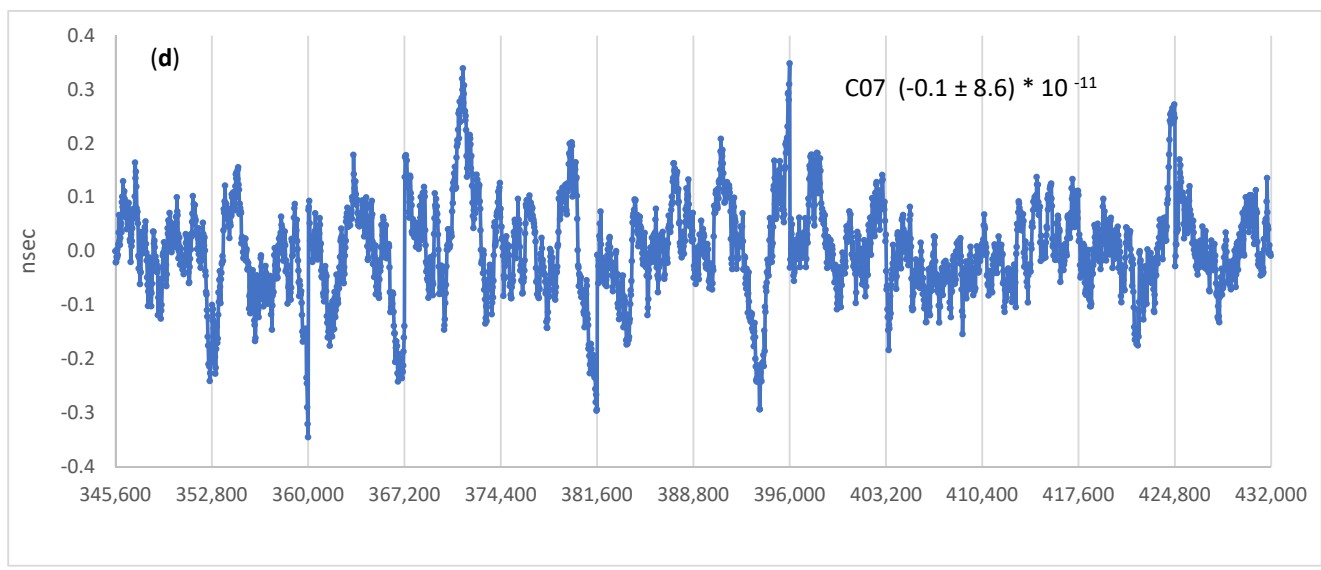

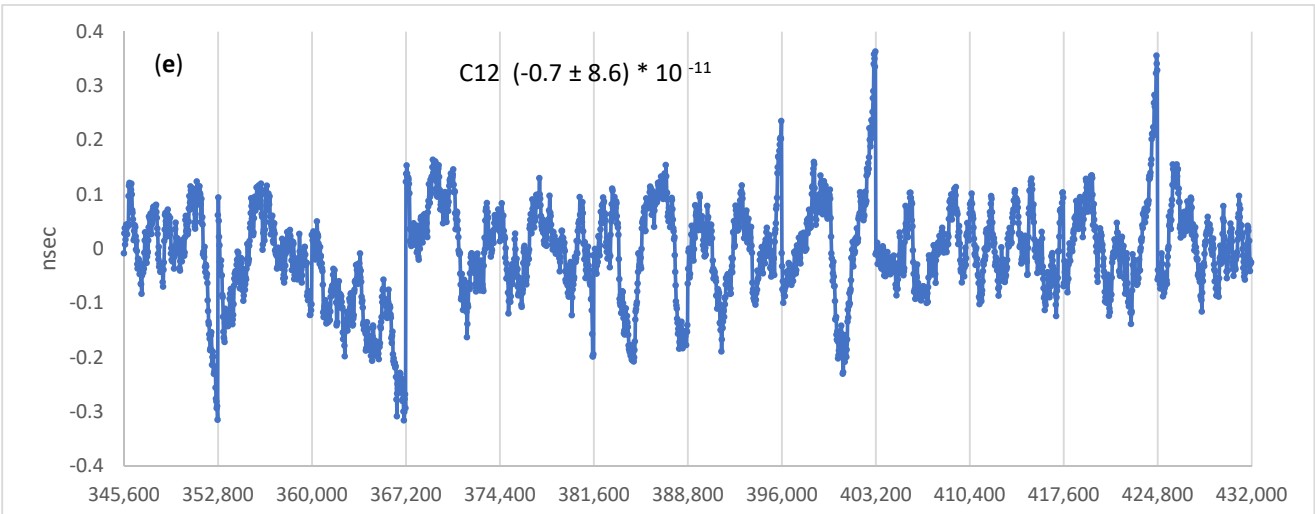

**Figure 13.** Plots of the differences between IGS high rate clocks and the prediction of our clock polynomials based on 15 min sampling. Update rate of the clock polynomials are 2 h for (**a**) G01, (**b**) E01, (**c**) 1 h for R01, (**d**) C07, and (**e**) C12. The mean and rms of the residuals are provided for each satellite.

## 5. Improving the Broadcast Model for GPS-like and Glonass-like Messages

We have seen indications that both the GPS-like and Glonass-like navigation messages have a potential for high accuracy, in the sense that the model coefficients can be tuned to a precise ephemeris with centimetric rms spread relative to a precise SP3 ephemeris. At these levels of accuracy, it is meaningful to ask if the model implemented in both types of Broadcast Ephemeris is adequate to such accuracies.

For GPS-like messages, we have secular perturbations in inclination and node, and periodic perturbations in mean anomaly, in the radial direction and inclination. The period is one half the Keplerian period, roughly 6 h, implying that the periodic perturbations caused by the $J_2$ part of the Earth's gravity field are modeled.

For Glonass-like messages, the Glonass ICD (Interface Control Document) prescribes a force field which includes, for the Earth's gravity, the monopole and quadrupole ($J_2$) components.

To investigate the effect of higher order terms of the gravitational potential on the orbital elements, we refer to [27]. The differential equations relating a disturbing potential $U$ to the temporal changes of the orbital elements $\varepsilon^k$ ($k$ = 1:6) have the general form:

$$\frac{d\varepsilon^k}{dt} = \mathcal{L}^k(a,e,I)U \tag{3}$$

where $\mathcal{L}^k(a,e,I)$ is a linear differential operator dependent on the semimajor axis $a$ eccentricity $e$ and inclination $I$. If we assume that interaction of perturbations can be ignored (this is appropriate for all the harmonics except $J_2$), then we can write:

$$\varepsilon^k = \varepsilon^k_{\,0} + \sum_{l=2}^{\infty} \sum_{m=0}^{l} \delta\varepsilon^k_{\,lm} \tag{4}$$

where $\delta\varepsilon^k_{\,lm}$ is the perturbation of the k-th orbital element due to the harmonic $C_{lm}$ and $\varepsilon^k_{\,0}$ is the unperturbed orbital element. The general form of the perturbed orbital elements is

$$\delta\varepsilon^k_{\,lm} = Re\left[ \mathcal{L}^k(a,e,I) \sum_{p=0}^{l} \sum_{q=-\infty}^{\infty} C_{lm}A_{lmpq}(a_0,e_0,I_0)\frac{e^{i[\psi_0 - \pi/2]}}{\dot{\psi}_0} \right] \tag{5}$$

$$\psi_0 = (l-2p)(\omega_0 + \dot{\omega}t) + (l-2p+q)M + m(\Omega_0 + \dot{\Omega}t - \dot{\theta}) \tag{6}$$

$$\dot{\psi}_0 = (l-2p)\dot{\omega} + (l-2p+q)n + m(\dot{\Omega} - \dot{\theta}) \tag{7}$$

The coefficients $A_{lmpq}$ scale as $(a_e/a)^l$, $e^q$ and $sin^p(I)$, $cos^p(I)$, and abs($q$) < 5 typically. $\psi_0$, $\dot{\psi}_0$ are the phase and frequency of the harmonic perturbation, and $M, \dot{\omega}, n, \dot{\Omega}, \dot{\theta}$ are respectively the mean anomaly, the rate of the perigee, the mean motion, the rate of the ascending node and the Earth rotation rate. Secular terms arise when $m = 0$ and $(l-2p) = q = 0$ that is for even zonal harmonics. Because the rate of perigee and node are of the order of $10^{-3}n$ and the period is $n \cong 2\dot{\theta}$, the period P of a perturbation is primarily determined by:

$$\frac{2\pi}{P} \approx [2(l-2p+q) - m]\dot{\theta} \tag{8}$$

It is observed that the amplitude of the harmonics $C_{lm}$ decrease with the degree $l$ according to the Kaula's rule:

$$|C_{lm}| \approx \frac{10^{-5}}{l^2} \tag{9}$$

Equation (9) implies that, except for resonant terms, we have to consider small degree terms. For example, considering the $J_3$ zonal ($l = 3$, $m = 0$), for $p = q = 0$ we have perturbations with period close to 4 h. The $J_3$ term, accounts for the pear-shaped figure of equilibrium of the Earth, that is, the lack of symmetry between north and south hemispheres. The conventional values from the JGM-3 gravity model [28] are:

$$JGM - 3 \text{ zonals}: J_2 = -0.10826 * 10^{-2}; J_3 = 0.25324 * 10^{-5} \tag{10}$$

We can estimate a maximum value of the perturbation caused by $J_3$ for an orbit of roughly 26,000 km by scaling the maximum periodic perturbation due to $J_2$, which along track has a maximum amplitude of ca. 200 m:

$$\frac{J_3}{J_2}200 \approx 0.47 \; m \tag{11}$$

The period of the associated perturbation is 1/3 of the orbital frequency, which is close to 4 h. We suggest that both GPS-like and Glonass-like messages should account for such term. The changes in the respective messages which could accommodate the improved

gravity model are described by the following equations for a GPS-like ephemeris and a Glonass-like model.

For GPS, denoting by uk the true anomaly from the node, we need to introduce six additional coefficients $c3_{uc}$, $c3_{us}$, $c3_{ic}$, $c3_{is}$, $c3_{rc}$, $c3_{rs}$ which define the amplitude of the cosine and sine component of the third harmonic periodic perturbation in the along track, cross track and radial directions, respectively. This notation is similar to that adopted ($c_{uc}$, ... , $c_{is}$) to represent the amplitudes of the perturbations caused by the second harmonic. This approach was first proposed by Zhou et al. [29]. The total periodic perturbations due to $J_2$ and $J_3$ are then given by:

$$\text{duk} = c_{uc} * \cos(2\text{uk}) + c_{us} * \sin(2\text{uk}) + c3_{uc} * \cos(3\text{uk}) + c3_{us} * \sin(3\text{uk}) \tag{12}$$

$$\text{drk} = c_{rc} * \cos(2\text{uk}) + c_{rs} * \sin(2\text{uk}) + c3_{rc} * \cos(3\text{uk}) + c3_{rs} * \sin(3\text{uk}) \tag{13}$$

$$\text{dik} = c_{ic} * \cos(2\text{uk}) + c_{is} * \sin(2\text{uk}) + c3_{ic} * \cos(3\text{uk}) + c3_{is} * \sin(3\text{uk}) \tag{14}$$

For Glonass, we simply need to modify the force function used in the numerical integration of the equations of motion in a co-rotating frame. The additional terms are marked in red.

$$F_x = -\frac{\mu x}{r^3} + J_2 \frac{\mu a_e^2}{r^7} x \left[ 6z^2 - \frac{3}{2}(x^2 + y^2) \right] + \omega_e^2 x + 2\omega_e y + \ddot{x}_{LS} + J_3 \frac{\mu a_e^3}{r^9} xz \left[ 10z^2 - \frac{15}{2}(x^2 + y^2) \right] \tag{15}$$

$$F_y = -\frac{\mu y}{r^3} + J_2 \frac{\mu a_e^2}{r^7} y \left[ 6z^2 - \frac{3}{2}(x^2 + y^2) \right] + \omega_e^2 y - 2\omega_e x + \ddot{y}_{LS} + J_3 \frac{\mu a_e^3}{r^9} yz \left[ 10z^2 - \frac{15}{2}(x^2 + y^2) \right] \tag{16}$$

$$F_z = -\frac{\mu z}{r^3} + J_2 \frac{\mu a_e^2}{r^7} z \left[ 3z^2 - \frac{9}{2}(x^2 + y^2) \right] + \ddot{z}_{LS} + J_3 \frac{\mu a_e^3}{r^9} \left[ \left\{ 4z^2 \left[ z^2 - 3\left(x^2 + y^2\right) \right] + \frac{3}{2}\left(x^2 + y^2\right)^2 \right\} \right] \tag{17}$$

For the GPS-like approach, the amplitude of the 3rd harmonic perturbations need to be estimated by including them in the set of solve for parameters (Equation (1)) and the relevant partials in the partial derivative matrix. These six additional parameters would justify an increase in length of the validity interval to 4 h or perhaps more, so that in a daily fit the total number of parameters to be estimated with 4-h arcs is 7 (Helmert) + 6 (arcs) * [3 (clocks) + 6 (orbit)] = 103 parameters, smaller than the 127 parameters of Section 2. Appropriate words would have to be defined in the Navigation message to allocate space to the six additional terms. For Glonass, there would be no change in the vector of state, nor in the navigation message. Only the function which describes the force needs to be modified.

We have seen in Figure 2 that the fit interval of the first arc of G01 was extended to 4 h to test the possibility of extending to 4 h the nominal validity time of 2 h. Figure 14 shows the detail of the post-fit residuals in this 4-h arc. The blue dots represent the best fitting signal we computed by modifying the GPS navigation message according to Equations (12)–(14). The similarity is evident, but for this short arc it is difficult to draw a sufficiently motivated conclusion.

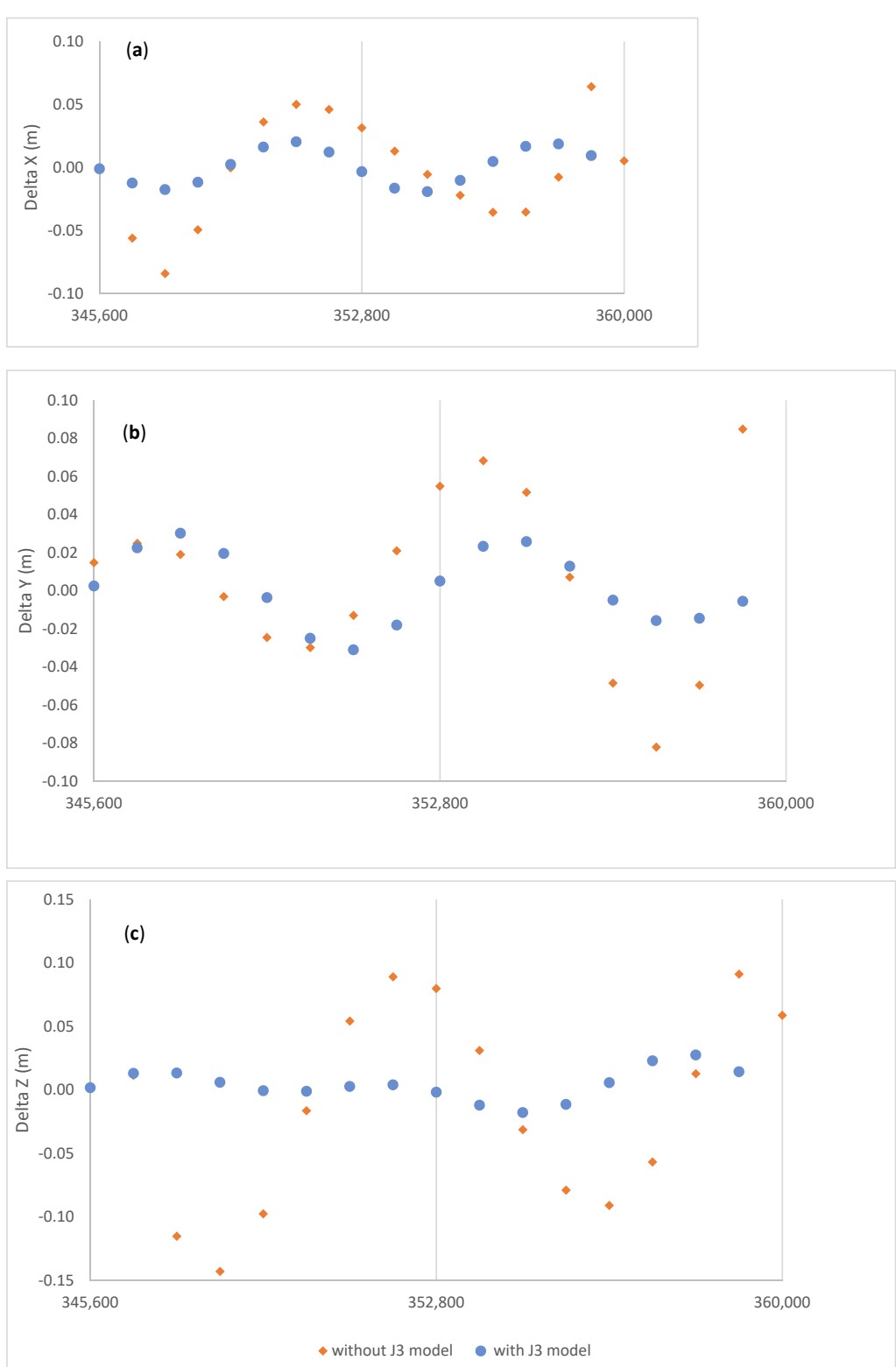

**Figure 14.** Zoom of the first 4 h of the post-fit residuals of G01 shown in Figure 2 (red diamonds) showing an oscillatory pattern ((**a**) X, (**b**) Y, and (**c**) Z). The blue dots represent an attempt to model the oscillation with a signal driven by the third zonal harmonics of the Earth's gravity field $J_3$.

## 6. Results for Volume Calculations

In the previous sections, we presented results based on limited data, in order to privilege the detail of the analysis more than the volume calculations on a larger number of days and satellites. In this section, we apply the same approach systematically to a larger body of data. We concentrate on the full month of January 2020.

In the Appendix A, Tables A1–A4 contain the Helmert parameters and their standard deviations for GPS, Glonass, Galileo and Beidou, respectively, averaged over one month. The Helmert parameters were estimated by modeling the differences between the ECEF coordinates computed with the broadcast ephemeris and those available in the SP3 file of CNES, on a satellite-by-satellite basis. For BeiDou, the CODE SP3 ephemeris were used. Then the 31 daily values were averaged. For GPS, BeiDou and Glonass, the mean of each of the seven parameters is zero within one standard deviation, implying that on average the broadcast and SP3 reference frames are aligned. For Galileo, the only parameters which are nonzero within one standard deviation are Rx and Rz. In general, the broadcast and SP3 frames are aligned. However, individual values (e.g., Rz of G01 or G11) can be large enough to generate significant departures of the broadcast predictions relative to the reference SP3 positions. Consequently, it is advisable, for maximum accuracy, to make available the Helmert parameters in conjunction with the corrections to the broadcast ephemeris, for each satellite and day.

We now turn to evaluate the effectiveness of our proposed approach to account for unmodeled reference frame adjustments and orbit effects in the broadcast ephemeris on all the satellites and days of January 2020. To this purpose, we have the average spectra of the pre-fit and post-fit residuals for each constellation. Figures 15–18 show the results for GPS, Glonass, Galileo and Beidou (IGSO and MEO), respectively. The spectra are computed in the Radial, Tangential and Across track triad. The upper plot shows the spectrum of the pre-fit residuals, i.e., the differences between broadcast and SP3 coordinates prior to the adjustment. The lower plot shows the spectra of the post-fit residuals, i.e., after the fit. The plots clearly show that the pre-fit spectra are dominated by low frequency signals (periods larger than 6 h, or frequency less than 0.05 mHz) with amplitudes of the order of 1 m. The lower plots show that the parameter adjustment proposed in this paper effectively removes the low frequency signals, so that the spectra of the differences between the positions computed with the corrected broadcast model and the SP3 positions are very nearly flat. The scales of the y axis are different. In particular for Galileo, it is worth noting that the proposed algorithm removes from the original broadcast ephemeris most of the unmodeled perturbations, whereas for Glonass we observe a lower efficiency.

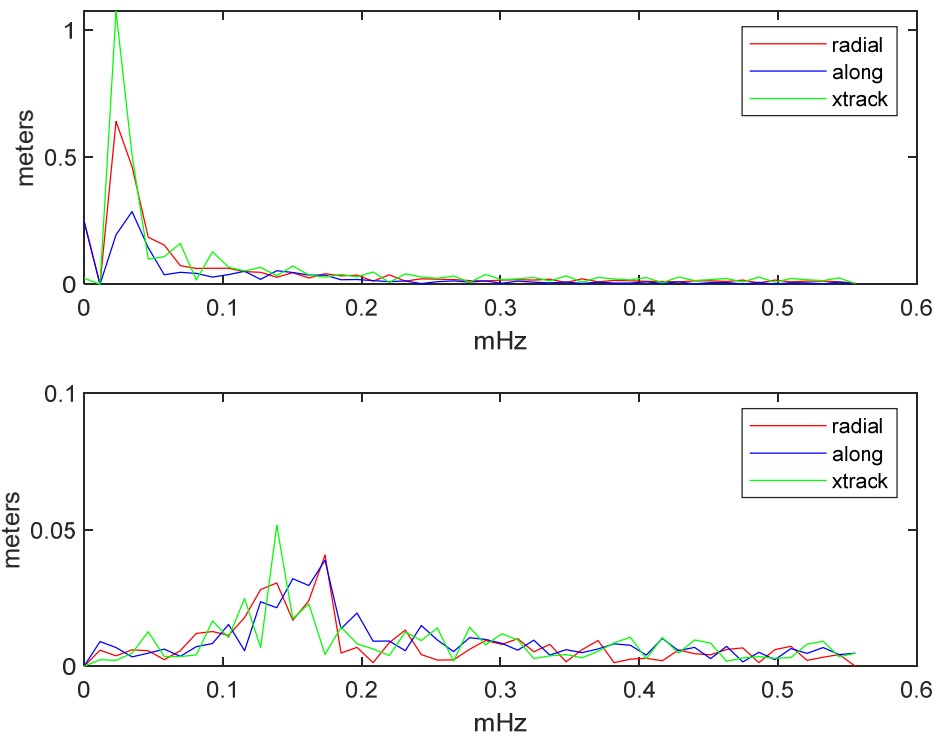

**Figure 15.** Spectrum of the pre-fit (top) and post-fit (bottom) residuals of the coordinates of the GPS satellites relative to the CNES precise ephemeris, projected on the radial, tangential and across track basis. Average values for January 2020.

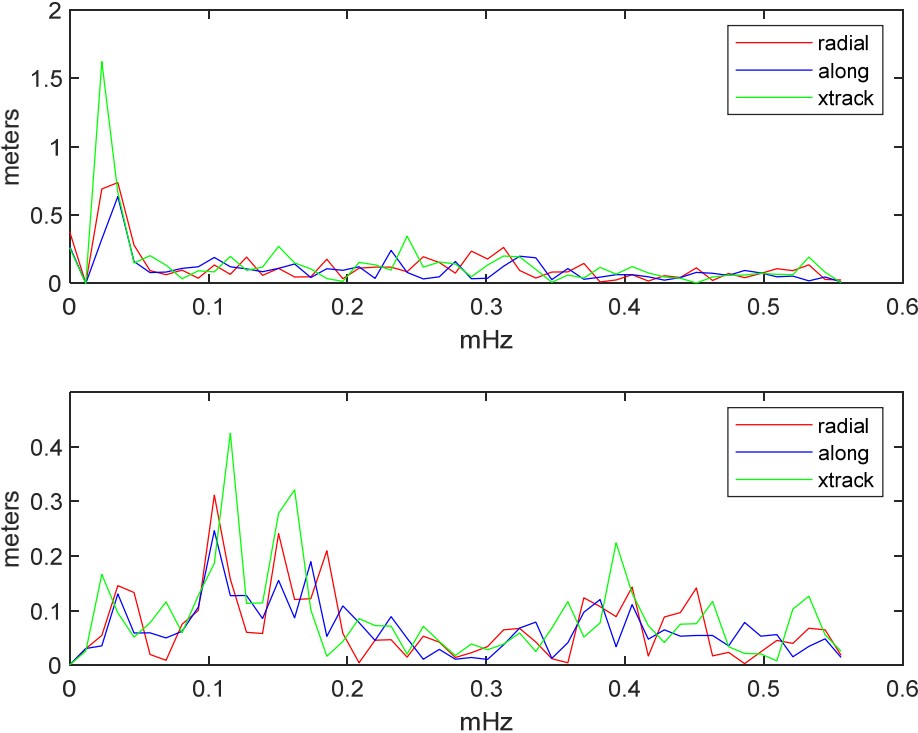

**Figure 16.** Spectrum of the pre-fit (top) and post-fit (bottom) residuals of the coordinates of the Glonass satellites relative to the CNES precise ephemeris, projected on the radial, tangential and across track basis. Average values for January 2020.

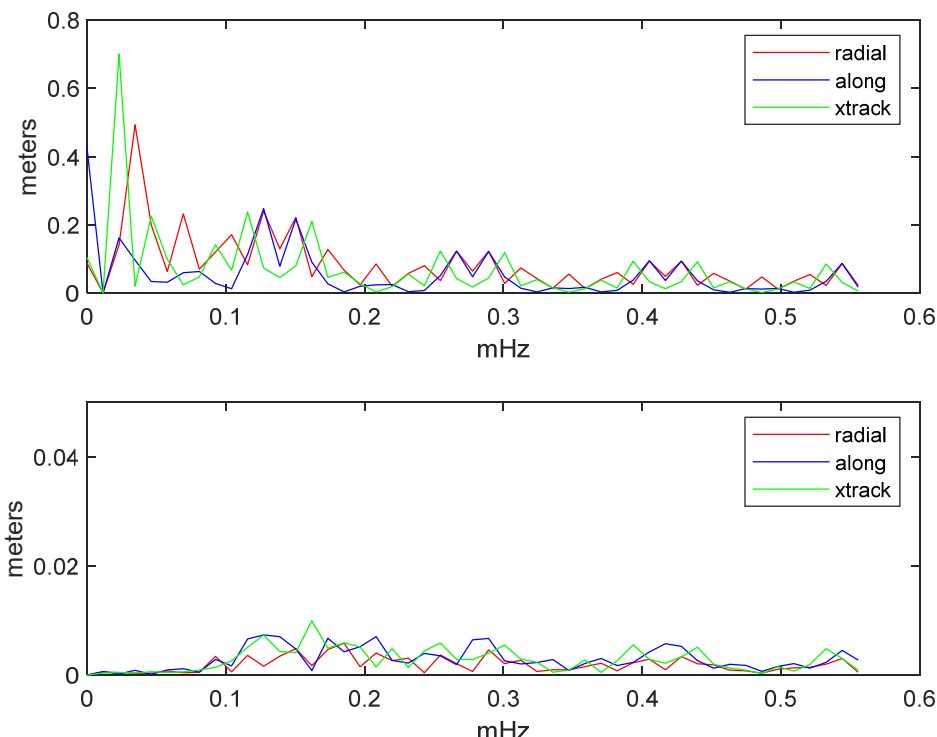

**Figure 17.** Spectrum of the pre-fit (top) and post-fit (bottom) residuals of the coordinates of the Galileo satellites relative to the CNES precise ephemeris, projected on the radial, tangential and across track basis. Average values for January 2020.

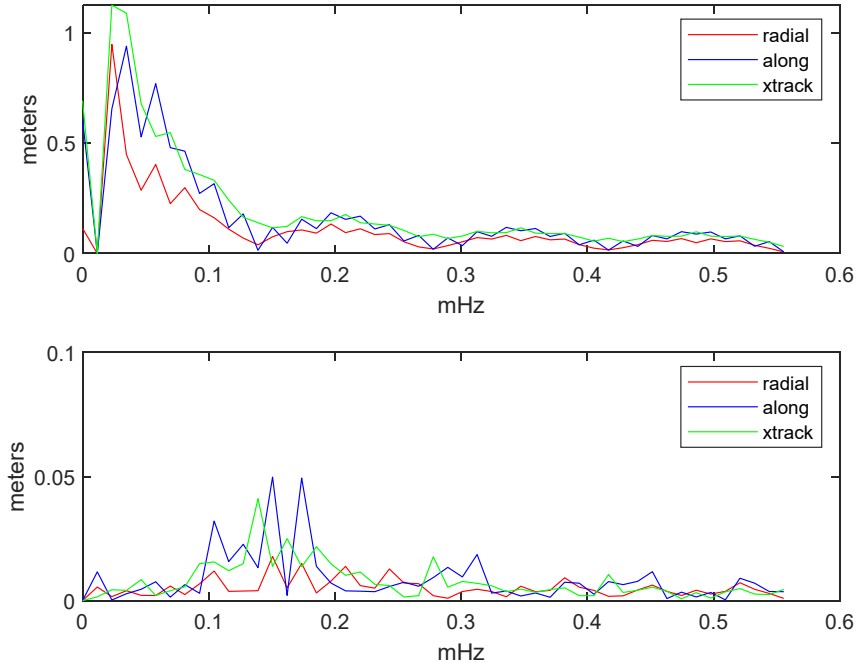

**Figure 18.** Spectrum of the pre-fit (top) and post-fit (bottom) residuals of the coordinates of the Beidou satellites (IGSO and MEO) relative to the CODE precise ephemeris, projected on the radial, tangential and across track basis. Average values for January 2020.

## 7. Discussion

The sample of data we have tested suggests that the GPS-like (Keplerian orbit plus perturbations) navigation message can be tuned to a precise ephemeris and clock with an rms spread ranging from 1.4 cm (Galileo) to 5.2 cm (GPS), provided that the corrections to

the broadcast parameters are complemented by daily adjustments in origin and orientation of the reference frame parameters. These adjustments are negligibly small for Galileo but can be of several tens of cm for GPS and Beidou. For an IGSO orbit like Beidou C07, the translation parameters are undefined due to lack of geometry. It may be expected that similar results can be obtained for Japan's QZSS which are also in a IGSO orbit, and for India's NAVIC/IRNSS.

The validity time of the ephemeris is two hours. For GPS, we tested a fit interval of 4 h. In such a case, the post-fit residuals indicate the presence of an oscillation of an approximately 4-h period, which is the signal expected to be caused primarily by the third zonal ($J_3$) of the Earth's gravity field. Including this perturbation in the navigation message would help in increasing the validity time and have a more random spread of the post-fit residuals.

The inference is that the format of the broadcast ephemeris adopted by GPS, Galileo, Beidou, and possibly NAVIC/IRNSS and QZSS, can represent the SV position to very high accuracy, comparable with that of the precise ephemeris. Comparison of power spectra of the Broadcast–SP3 residuals before and after the adjustment computed for all the satellites and one month support this hypothesis. Satellites in the GEO orbit have not been tested because a reference precise ephemeris is at this time unavailable.

For the clock corrections, a similar reasoning applies. The three-parameter model can likewise be tuned on precise clock corrections at 15 min sampling. It is to be noted, however, that flicker noise at sampling rates of up to 100 s can increase considerably the Allan variance for Cesium or Rubidium clocks. For the Galileo's Passive Hydrogen Maser this is also true but not so marked as for Cesium or Rubidium clocks. Therefore, the effective validity of the polynomial clock model at sampling rates of the order of 1 Hz needs to be tested with IGS's high rate clocks.

One of the most important challenges in satellite navigation precise positioning is to be able to broadcast corrections to the navigation message so that the user has positions and clocks of the used GNSS's of sufficient accuracy to apply for example a PPP (Precision Point Positioning) algorithm.

Currently, the International GNSS Service (IGS) agency and various analysis centers (ACs) provide users with ultra-rapid precise satellite orbit and clock products for GPS and Glonass with 6-h updates and a 3-h latency (igs.org/acc/gps-only/#ultra-rapid, accessed on 10 October 2021). Corresponding products for Galileo and Beidou are discussed by [30]. The accuracy of the predicted orbits is not good enough for high-precision PPP [25]. A widely used approach to generate high accuracy orbits and clocks usable in Real Time PPP is the SSR (State Space Representation). In the SSR concept (https://www.igscb.org/wp-content/uploads/2020/10/igs_ssr_-v1_00_20201005.pdf, accessed on 10 October 2021), the broadcast ephemeris data are complemented with a set of corrections along track, across track and radial, which are described by a piecewise linear function of time. Likewise, the clock corrections to the broadcast values are described by a piecewise quadratic function of time. Tests done on several Real Time and Orbit and Clock products indicate an update rate of typically 5 s, with a comparable latency. Specific RTCM messages 1060 and 1066 have been defined for GPS and Glonass SSR, respectively.

We suggest that if sufficiently accurate orbit and clock predictions are available, the navigation message could be broadcast in a corrected form as a streamed RTCM (Radio Technical Commission for Maritime Services, www.rtcm.org, accessed on 10 October 2021) message as described in this paper, with additional reference frame information for GPS-like messages. Given the validity time of two hours, perhaps extendable to four hours with a refined gravity model, for GPS-like messages, and one hour for Glonass, the refresh rate could be considerably longer than SSR. If the SSR corrections are based on the improved navigation message, there would be a twofold advantage: (a) the SSR corrections would address the finest details of the orbit and clocks, for improved overall accuracy; (b) the user would have a redundancy in case of unavailable SSR data, as he would still be able to compute its position with a high accuracy ephemeris in a broadcast rather than SP3 format.

Applicable RTCM messages could be 1019, 1020, 1042, 1045/6 for GPS, Glonass, Beidou and Galileo I/NAV and F/NAV, respectively (the two Galileo navigation messages refer to the two different ionospheric free linear combinations and corresponding clocks), and 1021 for the Helmert parameters [31].

In conclusion, we suggest that a corrected broadcasted message as it has been presented here, rather than the message broadcast by the various GNSSs, could be the basis for an even more accurate set of SSR corrections for both position and clocks.

## 8. Conclusions

The limited accuracy of the broadcast data has been discussed in detail in several publications. Possible actions towards an improvement of the broadcast message are the focus of this paper. We have investigated the potential of the GPS and Glonass navigation messages to represent satellite position and clock with an accuracy comparable with that of the precise ephemeris delivered by the IGS within the MGEX project. The reference precise products (ephemeris and clocks) were generated by CNES for GPS, Glonass and Galileo, and by CODE for Beidou (MEO and IGSO). We have examined one satellite per constellation in one specific day, and obtained indication that the broadcast message can reproduce the precise ephemeris with a centimetric rms, provided that: (a) for GPS-like messages (GPS, Galileo and Beidou), arcs of two hours are used for the fit, complemented by one set of Helmert parameters for the day; and (b) for Glonass-like messages arcs of one hour are used for the fit. For Glonass the Helmert parameters are unnecessary because the Glonass message is based directly on Cartesian ECEF coordinates and velocities. Volume calculations on all satellites and for one month indicate that the proposed approach is applicable in general.

Corrections to the clock parameters are also computed, and similar accuracy has been demonstrated. High frequency clock files from IGS have also been tested against the estimated clock polynomials. The results suggest that the rms spread is limited to fractions of nanoseconds for GPS, Galileo and Beidou, and a factor of 10 higher for Glonass. Systematic drifts in the clock differences are clearly visible, implying that our clock polynomial is likely to smooth the high frequency noise of the on board clocks, particularly for the Rubidium and Cesium clocks. Order of magnitude estimates of the perturbations caused by the higher order terms of the gravity field suggest the opportunity to modify the message to include the effect of the $J_3$ component of the Earth's gravity field.

Implications of our results for real time applications very much depend on the availability of predicted precise orbits and clocks which can be represented in a broadcast form. In such case, appropriate RTCM messages are available for the examined constellations, both for the broadcast message and the complementary Helmert parameters.

**Author Contributions:** A.C. conceptualization, paper writing, analysis; J.Z. software coding and implementation. All authors have read and agreed to the published version of the manuscript.

**Funding:** This research is supported in part by a contract with Regione del Veneto titled 'Use of precision GNSS for applications to land monitoring of the Regione of Veneto', and by the contracts with the GSA (Galileo Supervising Authority) GISCAD-OV and GRC-MS.

**Institutional Review Board Statement:** Not applicable.

**Informed Consent Statement:** Not applicable.

**Data Availability Statement:** Not applicable.

**Conflicts of Interest:** The authors declare no conflict of interest.

## Appendix A

**Table A1.** Helmert parameters relating the GPS broadcast reference frame to the ITRF2014 Reference frame of the precise ephemeris of CNES. Average values for January 2020.

| | TX (m) | Std_TX (m) | TY (m) | Std_TY (m) | TZ (m) | Std_TZ (m) | RX (mas) | Std_RX (mas) | RY (mas) | Std_RY (mas) | RZ (mas) | Std_RZ (mas) | Sc (ppm) | Std_Sc (ppm) |
|---|---|---|---|---|---|---|---|---|---|---|---|---|---|---|
| G01 | 0.030 | 0.150 | −0.154 | 0.196 | −0.015 | 0.056 | −1.531 | 1.569 | 1.101 | 2.508 | 10.197 | 3.345 | 0.018 | 0.000 |
| G02 | −0.008 | 0.167 | −0.020 | 0.187 | −0.057 | 0.112 | −0.001 | 1.318 | 0.105 | 2.334 | 0.215 | 7.045 | 0.033 | 0.001 |
| G03 | 0.161 | 0.216 | 0.040 | 0.259 | −0.248 | 0.184 | −0.696 | 2.532 | 1.599 | 3.782 | −7.023 | 7.854 | 0.015 | 0.001 |
| G04 | −0.070 | 0.126 | 0.315 | 0.203 | 0.005 | 0.110 | 1.027 | 1.430 | 3.241 | 2.680 | 5.033 | 2.941 | −0.039 | 0.000 |
| G05 | −0.010 | 0.096 | −0.008 | 0.084 | 0.147 | 0.077 | −0.059 | 1.380 | −0.034 | 1.098 | −0.707 | 3.871 | 0.031 | 0.000 |
| G06 | −0.059 | 0.075 | 0.083 | 0.125 | −0.097 | 0.078 | −1.028 | 1.324 | −0.469 | 1.713 | −2.255 | 4.417 | 0.014 | 0.000 |
| G07 | −0.019 | 0.098 | 0.069 | 0.174 | −0.167 | 0.094 | 0.001 | 1.142 | 0.702 | 2.166 | 2.844 | 4.397 | 0.031 | 0.000 |
| G08 | −0.071 | 0.116 | 0.066 | 0.175 | 0.064 | 0.176 | −0.472 | 0.934 | −0.781 | 2.385 | 6.746 | 8.891 | 0.017 | 0.001 |
| G09 | 0.070 | 0.165 | −0.083 | 0.225 | −0.237 | 0.093 | −0.998 | 2.112 | −0.606 | 2.164 | 3.085 | 7.022 | 0.012 | 0.001 |
| G10 | −0.007 | 0.174 | −0.050 | 0.124 | −0.104 | 0.089 | 0.335 | 2.486 | 0.061 | 1.337 | 0.730 | 3.393 | 0.015 | 0.001 |
| G11 | −0.103 | 0.135 | −0.027 | 0.112 | 0.168 | 0.066 | −1.161 | 1.320 | −0.688 | 1.557 | 7.600 | 3.838 | −0.017 | 0.001 |
| G12 | 0.033 | 0.149 | −0.020 | 0.080 | 0.020 | 0.103 | −0.447 | 1.858 | 0.084 | 1.292 | −4.620 | 3.282 | 0.033 | 0.001 |
| G13 | 0.034 | 0.156 | 0.031 | 0.084 | 0.007 | 0.088 | 0.387 | 1.643 | 0.196 | 1.451 | −0.749 | 3.911 | −0.010 | 0.001 |
| G14 | 0.050 | 0.225 | 0.003 | 0.073 | −0.117 | 0.066 | −0.191 | 0.858 | −0.650 | 3.300 | 0.078 | 4.042 | −0.022 | 0.000 |
| G15 | −0.059 | 0.092 | −0.006 | 0.158 | −0.054 | 0.107 | −0.748 | 1.312 | −0.352 | 2.224 | 1.194 | 4.491 | 0.024 | 0.001 |
| G16 | 0.015 | 0.202 | 0.072 | 0.099 | −0.006 | 0.090 | −0.101 | 2.528 | −0.859 | 1.336 | −4.817 | 5.834 | −0.007 | 0.001 |
| G17 | 0.148 | 0.199 | 0.139 | 0.277 | −0.036 | 0.062 | −0.697 | 2.086 | −2.526 | 3.561 | −9.596 | 6.593 | 0.030 | 0.001 |
| G19 | 0.028 | 0.173 | 0.013 | 0.080 | 0.027 | 0.053 | −0.090 | 1.000 | −0.386 | 2.211 | −1.517 | 3.482 | 0.031 | 0.001 |
| G19 | 0.028 | 0.173 | 0.013 | 0.080 | 0.027 | 0.053 | −0.090 | 1.000 | −0.386 | 2.211 | −1.517 | 3.482 | 0.031 | 0.001 |
| G20 | −0.053 | 0.155 | 0.015 | 0.155 | 0.066 | 0.092 | −0.075 | 2.165 | 0.720 | 1.443 | −5.388 | 2.972 | −0.012 | 0.000 |
| G21 | −0.042 | 0.101 | −0.062 | 0.187 | −0.021 | 0.077 | 0.615 | 1.368 | −0.542 | 2.141 | 4.565 | 3.996 | −0.006 | 0.001 |
| G22 | 0.021 | 0.154 | −0.028 | 0.087 | 0.061 | 0.080 | −0.325 | 1.688 | 0.047 | 1.138 | −0.465 | 4.762 | 0.031 | 0.000 |
| G23 | 0.042 | 0.070 | 0.155 | 0.148 | −0.149 | 0.066 | −0.004 | 0.798 | 1.510 | 1.865 | 6.234 | 4.165 | 0.050 | 0.000 |
| G24 | −0.021 | 0.471 | −0.044 | 0.271 | −0.144 | 0.214 | −0.529 | 1.672 | −0.305 | 6.123 | −0.451 | 8.189 | 0.012 | 0.002 |
| G25 | 0.029 | 0.054 | 0.134 | 0.134 | −0.087 | 0.097 | 1.275 | 1.557 | 0.411 | 0.911 | −3.036 | 4.794 | 0.015 | 0.000 |
| G26 | 0.112 | 0.127 | 0.096 | 0.093 | 0.031 | 0.128 | 1.320 | 1.493 | −0.649 | 1.055 | −2.084 | 5.556 | 0.014 | 0.001 |
| G27 | −0.032 | 0.143 | −0.028 | 0.080 | 0.178 | 0.091 | −0.560 | 1.797 | −0.252 | 1.278 | 0.115 | 3.791 | 0.016 | 0.001 |
| G28 | 0.077 | 0.304 | −0.014 | 0.194 | −0.042 | 0.145 | −0.472 | 3.445 | −1.127 | 1.850 | −2.172 | 4.414 | −0.022 | 0.001 |
| G29 | 0.097 | 0.203 | −0.048 | 0.129 | 0.061 | 0.080 | −1.340 | 2.466 | −0.135 | 1.404 | −5.017 | 9.345 | 0.030 | 0.001 |
| G30 | 0.024 | 0.092 | 0.072 | 0.093 | −0.086 | 0.087 | −0.187 | 1.315 | 0.769 | 1.232 | 3.065 | 4.180 | 0.017 | 0.000 |
| G31 | −0.072 | 0.164 | 0.025 | 0.092 | −0.172 | 0.075 | −1.005 | 1.904 | 0.195 | 1.537 | −4.561 | 3.012 | 0.035 | 0.001 |
| G32 | 0.030 | 0.257 | 0.051 | 0.254 | −0.181 | 0.052 | −0.598 | 1.934 | −0.712 | 3.800 | 4.074 | 4.608 | 0.017 | 0.000 |
| Average | 0.013 | | 0.025 | | −0.036 | | −0.264 | | −0.023 | | −0.006 | | 0.014 | |
| Std | 0.064 | | 0.086 | | 0.109 | | 0.689 | | 0.995 | | 4.502 | | 0.021 | |

**Table A2.** Helmert parameters relating the Glonass reference frame to the ITRF2014 reference frame of the precise ephemeris of CNES. Average values for January 2020.

| | TX (m) | Std_TX (m) | TY (m) | Std_TY (m) | TZ (m) | Std_TZ (m) | RX (mas) | Std_RX (mas) | RY (mas) | Std_RY (mas) | RZ (mas) | Std_RZ (mas) | Sc (ppm) | Std_Sc (ppm) |
|---|---|---|---|---|---|---|---|---|---|---|---|---|---|---|
| **R01** | 0.030 | 0.134 | −0.013 | 0.140 | 0.231 | 0.186 | 0.225 | 2.535 | −0.115 | 1.508 | 2.958 | 6.688 | −0.081 | 0.001 |
| **R02** | −0.008 | 0.218 | −0.048 | 0.190 | 0.045 | 0.272 | 0.183 | 3.020 | −0.156 | 3.289 | −11.250 | 8.906 | −0.080 | 0.002 |
| **R03** | −0.050 | 0.155 | −0.019 | 0.145 | 0.028 | 0.159 | −0.386 | 2.257 | −0.768 | 2.295 | 0.885 | 5.835 | −0.080 | 0.001 |
| **R04** | 0.055 | 0.130 | −0.017 | 0.111 | 0.407 | 0.210 | 0.210 | 1.709 | −0.039 | 1.687 | −9.333 | 7.088 | −0.085 | 0.002 |
| **R05** | −0.003 | 0.103 | −0.015 | 0.117 | 0.136 | 0.180 | 0.176 | 1.755 | −0.404 | 1.969 | −6.234 | 7.836 | −0.082 | 0.001 |
| **R07** | −0.022 | 0.107 | −0.059 | 0.135 | 0.035 | 0.258 | 0.061 | 2.456 | −0.865 | 2.348 | 1.454 | 12.849 | −0.080 | 0.002 |
| **R08** | −0.088 | 0.188 | −0.070 | 0.238 | 0.158 | 0.238 | −0.079 | 3.245 | −0.785 | 2.524 | 0.275 | 9.565 | −0.080 | 0.001 |
| **R09** | 0.015 | 0.157 | 0.057 | 0.125 | −0.008 | 0.272 | −0.243 | 2.630 | 0.720 | 1.653 | 16.964 | 5.741 | −0.057 | 0.002 |
| **R11** | 0.021 | 0.274 | 0.083 | 0.187 | −0.050 | 0.247 | 0.318 | 3.861 | −0.392 | 3.380 | 8.649 | 10.854 | −0.082 | 0.002 |
| **R11** | 0.021 | 0.274 | 0.083 | 0.187 | −0.050 | 0.247 | 0.318 | 3.861 | −0.392 | 3.380 | 8.649 | 10.854 | −0.082 | 0.002 |
| **R12** | −0.008 | 0.127 | 0.023 | 0.123 | 0.076 | 0.214 | −0.424 | 2.142 | −0.585 | 1.808 | 7.451 | 10.115 | −0.079 | 0.002 |
| **R13** | 0.020 | 0.273 | −0.009 | 0.279 | 0.161 | 0.437 | 0.028 | 4.544 | −0.090 | 4.261 | 10.536 | 17.612 | −0.081 | 0.003 |
| **R14** | 0.029 | 0.137 | 0.021 | 0.168 | 0.031 | 0.241 | 0.494 | 2.411 | −0.998 | 1.869 | −3.003 | 9.143 | −0.083 | 0.001 |
| **R15** | −0.002 | 0.124 | 0.020 | 0.131 | 0.093 | 0.211 | 0.068 | 1.951 | −0.713 | 1.509 | 4.912 | 8.398 | −0.081 | 0.002 |
| **R16** | 0.040 | 0.138 | 0.027 | 0.141 | 0.145 | 0.240 | 0.459 | 2.449 | −0.687 | 1.717 | −4.259 | 9.195 | −0.083 | 0.001 |
| **R17** | −0.028 | 0.161 | −0.041 | 0.161 | −0.522 | 0.145 | 0.041 | 1.854 | −0.919 | 3.291 | 7.842 | 8.461 | −0.079 | 0.002 |
| **R18** | −0.023 | 0.183 | −0.056 | 0.159 | −0.330 | 0.123 | −0.107 | 1.529 | −0.978 | 3.257 | 2.069 | 8.527 | −0.081 | 0.001 |
| **R19** | 0.031 | 0.184 | −0.052 | 0.145 | −0.411 | 0.184 | 0.088 | 1.791 | −0.441 | 3.590 | −2.504 | 11.598 | −0.079 | 0.003 |
| **R20** | 0.029 | 0.203 | −0.043 | 0.187 | −0.301 | 0.150 | 0.496 | 1.863 | −0.525 | 3.990 | 2.336 | 9.321 | −0.095 | 0.001 |
| **R21** | −0.014 | 0.167 | −0.027 | 0.143 | −0.306 | 0.118 | 0.089 | 1.646 | −0.915 | 2.784 | −5.814 | 6.011 | −0.080 | 0.001 |
| **R22** | 0.184 | 0.273 | −0.040 | 0.230 | 0.219 | 0.177 | 0.782 | 2.639 | 1.377 | 5.168 | −4.685 | 12.145 | −0.080 | 0.002 |
| **R23** | −0.029 | 0.240 | −0.023 | 0.210 | −0.313 | 0.207 | −0.289 | 1.875 | −0.343 | 4.151 | 8.951 | 11.344 | −0.080 | 0.002 |
| **R24** | | | | | | | | | | | | | | |
| **Average** | 0.009 | | −0.010 | | −0.024 | | 0.114 | | −0.410 | | 1.675 | | −0.080 | |
| **std** | 0.051 | | 0.044 | | 0.239 | | 0.301 | | 0.567 | | 7.142 | | 0.006 | |

**Table A3.** Helmert parameters relating the Galileo reference frame to the ITRF2014 reference frame of the precise ephemeris of CNES. Average values for January 2020.

| | TX (m) | Std_TX (m) | TY (m) | Std_TY (m) | TZ (m) | Std_TZ (m) | RX (mas) | Std_RX (mas) | RY (mas) | Std_RY (mas) | RZ (mas) | Std_RZ (mas) | Sc (ppm) | Std_Sc (ppm) |
|---|---|---|---|---|---|---|---|---|---|---|---|---|---|---|
| **E01** | −0.021 | 0.073 | −0.024 | 0.054 | 0.038 | 0.063 | −0.161 | 0.537 | 0.153 | 0.625 | 0.684 | 0.708 | −0.003 | 0.000 |
| **E02** | −0.004 | 0.051 | −0.019 | 0.057 | 0.009 | 0.052 | −0.267 | 0.430 | 0.287 | 0.777 | 0.250 | 0.695 | −0.004 | 0.000 |
| **E03** | −0.002 | 0.043 | 0.012 | 0.070 | 0.020 | 0.066 | −0.291 | 0.484 | −0.092 | 0.611 | 0.668 | 0.556 | −0.005 | 0.000 |

**Table A3.** *Cont.*

| | TX (m) | Std_TX (m) | TY (m) | Std_TY (m) | TZ (m) | Std_TZ (m) | RX (mas) | Std_RX (mas) | RY (mas) | Std_RY (mas) | RZ (mas) | Std_RZ (mas) | Sc (ppm) | Std_Sc (ppm) |
|---|---|---|---|---|---|---|---|---|---|---|---|---|---|---|
| E04 | −0.013 | 0.065 | −0.021 | 0.072 | 0.054 | 0.103 | −0.188 | 0.624 | −0.289 | 0.596 | 0.531 | 1.188 | −0.005 | 0.000 |
| E05 | 0.002 | 0.045 | −0.003 | 0.061 | 0.093 | 0.061 | −0.134 | 0.621 | 0.054 | 0.644 | 0.528 | 0.665 | −0.007 | 0.000 |
| E07 | −0.023 | 0.063 | −0.019 | 0.052 | 0.051 | 0.066 | 0.062 | 0.546 | −0.160 | 0.432 | 0.800 | 0.711 | −0.002 | 0.000 |
| E08 | −0.006 | 0.047 | −0.016 | 0.069 | 0.020 | 0.076 | −0.108 | 0.514 | 0.132 | 0.488 | 0.441 | 0.553 | −0.005 | 0.000 |
| E09 | −0.023 | 0.051 | −0.018 | 0.054 | 0.018 | 0.066 | −0.088 | 0.624 | 0.008 | 0.530 | 0.216 | 0.634 | −0.004 | 0.000 |
| E11 | −0.001 | 0.071 | −0.018 | 0.110 | 0.035 | 0.118 | −0.337 | 1.068 | 0.030 | 1.009 | 0.660 | 1.419 | −0.001 | 0.000 |
| E12 | −0.025 | 0.070 | 0.017 | 0.061 | 0.056 | 0.061 | −0.235 | 0.613 | 0.144 | 0.772 | 1.593 | 0.994 | −0.002 | 0.000 |
| E13 | −0.003 | 0.057 | 0.010 | 0.064 | −0.003 | 0.050 | −0.393 | 0.537 | −0.080 | 0.620 | 0.963 | 1.066 | −0.008 | 0.000 |
| E14 | | | | | | | | | | | | | | |
| E15 | 0.008 | 0.047 | 0.009 | 0.070 | −0.029 | 0.051 | −0.230 | 0.579 | 0.224 | 0.607 | 0.854 | 0.927 | −0.005 | 0.000 |
| E18 | | | | | | | | | | | | | | |
| E19 | −0.027 | 0.055 | 0.022 | 0.066 | 0.274 | 0.073 | −0.316 | 0.574 | 0.097 | 0.896 | 0.085 | 1.018 | −0.001 | 0.000 |
| E21 | −0.051 | 0.072 | −0.036 | 0.062 | 0.006 | 0.074 | −0.136 | 0.496 | −0.099 | 0.842 | 0.370 | 0.580 | −0.009 | 0.001 |
| E24 | −0.016 | 0.076 | −0.002 | 0.083 | 0.019 | 0.107 | −0.137 | 0.710 | −0.050 | 0.685 | 0.181 | 0.769 | −0.003 | 0.001 |
| E25 | −0.020 | 0.066 | 0.008 | 0.058 | 0.002 | 0.067 | −0.213 | 0.483 | −0.109 | 0.596 | 0.042 | 0.435 | −0.006 | 0.000 |
| E26 | 0.008 | 0.068 | −0.033 | 0.062 | 0.026 | 0.057 | −0.431 | 0.643 | 0.036 | 0.590 | 1.011 | 0.841 | −0.006 | 0.000 |
| E27 | −0.030 | 0.063 | 0.011 | 0.081 | −0.007 | 0.075 | −0.277 | 0.612 | −0.037 | 0.520 | 0.441 | 0.941 | −0.005 | 0.000 |
| E30 | −0.036 | 0.054 | −0.013 | 0.057 | 0.005 | 0.065 | −0.149 | 0.511 | 0.138 | 0.718 | 0.271 | 0.869 | −0.003 | 0.000 |
| E31 | −0.050 | 0.070 | −0.025 | 0.063 | 0.053 | 0.073 | −0.235 | 0.540 | −0.066 | 0.550 | 0.574 | 0.724 | −0.005 | 0.000 |
| E33 | −0.008 | 0.069 | −0.005 | 0.073 | 0.023 | 0.066 | −0.434 | 0.848 | 0.042 | 0.633 | 1.062 | 0.680 | −0.005 | 0.000 |
| E36 | −0.001 | 0.058 | 0.006 | 0.065 | 0.014 | 0.089 | −0.329 | 0.671 | 0.101 | 0.480 | 0.474 | 0.496 | −0.005 | 0.000 |
| Average | −0.015 | | −0.007 | | 0.035 | | −0.229 | | 0.021 | | 0.577 | | −0.005 | |
| std | 0.017 | | 0.017 | | 0.059 | | 0.120 | | 0.135 | | 0.370 | | 0.002 | |

**Table A4.** Helmert parameters relating the Beidou reference frame to the ITRF2014 reference frame of the precise ephemeris of CODE. Average values for January 2020.

| Orbit Type | | TX (m) | Std_TX (m) | TY (m) | Std_TY (m) | TZ (m) | Std_TZ (m) | RX (mas) | Std_RX (mas) | RY (mas) | Std_RY (mas) | RZ (mas) | Std_RZ (mas) | Sc (ppm) | Std_Sc (ppm) |
|---|---|---|---|---|---|---|---|---|---|---|---|---|---|---|---|
| IGSO | C06 | −0.010 | 0.591 | −0.303 | 0.338 | 0.362 | 0.209 | 2.230 | 1.760 | 3.522 | 1.523 | −5.739 | 5.415 | −0.027 | 0.000 |
| IGSO | C07 | −0.636 | 0.573 | −1.247 | 0.454 | 0.115 | 0.335 | 2.232 | 3.277 | 2.154 | 3.446 | 0.176 | 8.256 | −0.002 | 0.002 |
| IGSO | C08 | −0.998 | 0.927 | −1.903 | 1.667 | 0.078 | 0.478 | −2.057 | 3.929 | 4.229 | 3.180 | −5.988 | 7.477 | −0.003 | 0.001 |
| IGSO | C09 | 0.339 | 0.730 | −0.066 | 0.273 | 0.135 | 0.289 | 1.982 | 1.276 | 0.476 | 2.170 | −5.809 | 5.322 | −0.030 | 0.000 |
| IGSO | C10 | −1.417 | 0.821 | −0.835 | 0.213 | 0.152 | 0.309 | 2.311 | 1.303 | −2.750 | 3.054 | 6.918 | 5.200 | −0.019 | 0.001 |
| MEO | C11 | −0.126 | 0.563 | 0.124 | 0.503 | −0.402 | 0.393 | 1.675 | 4.108 | −1.418 | 8.841 | −1.229 | 9.970 | −0.028 | 0.002 |
| MEO | C12 | −0.145 | 0.650 | 0.226 | 0.507 | −0.467 | 0.465 | 3.745 | 4.368 | −1.500 | 8.715 | −0.165 | 9.271 | −0.020 | 0.002 |
| IGSO | C13 | −1.244 | 0.525 | −0.751 | 0.257 | 0.114 | 0.203 | −0.420 | 1.298 | 2.481 | 1.600 | −1.932 | 3.658 | −0.036 | 0.000 |
| MEO | C14 | −0.055 | 0.499 | 0.286 | 0.560 | −0.709 | 0.487 | 2.284 | 4.577 | 0.454 | 6.516 | 3.984 | 10.634 | −0.028 | 0.005 |
| IGSO | C16 | 0.122 | 0.598 | 0.049 | 0.346 | 0.254 | 0.180 | 2.229 | 1.698 | 3.084 | 1.443 | −5.497 | 5.389 | −0.021 | 0.000 |
| | Average | −0.417 | | −0.442 | | −0.037 | | 1.621 | | 1.073 | | −1.528 | | −0.021 | |
| | std | 0.613 | | 0.725 | | 0.356 | | 1.645 | | 2.388 | | 4.453 | | 0.011 | |

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
