# Peer review of "Broadcast Ephemeris with Centimetric Accuracy: Test Results for GPS, Galileo, Beidou and Glonass"

_remotesensing, doi:10.3390/rs13204185_

Round 1
Reviewer 1 Report
Review of article remotesensing-1401325 „Broadcast Ephemeris with centimetric accuracy for GPS, Galileo, Beidou and Glonass.“ by Alessandro Caporali and Joaquin Zurutuza.
This paper contains the research results of the two pre-submitted papers remotesensing-1349279 and remotesensing-1349334, which I have already reviewed. This new revised essay has satisfactorily incorporated the comments I have already made in the two early essays. Therefore, I have hardly anything to complain about and find this new version very good.
I have noted further comments in a pdf document and attach it to this report. I would like to highlight three points:
a) The text in lines 376 to 382 is set in the wrong font (size).
b) The numbering of the formulas needs to be checked again. Formulas 12 (22), 13 (33), 14 (44), and 16(55) are numbered incorrectly in the text.
c) In the “Discussion” section, I would delete lines 588-591, as the information is already sufficiently presented in lines 610-615.
Language:
Linguistically, I can hardly find a significant number of errors in the paper. I found small errors here and there, which I have noted in the attached PDF document.

Author Response
Reviewer 1
I have noted further comments in a pdf document and attach it to this report. I would like to highlight three points:
- a) The text in lines 376 to 382 is set in the wrong font (size).
- b) The numbering of the formulas needs to be checked again. Formulas 12 (22), 13 (33), 14 (44), and 16(55) are numbered incorrectly in the text.
- c) In the “Discussion” section, I would delete lines 588-591, as the information is already sufficiently presented in lines 610-615.
Language:
Linguistically, I can hardly find a significant number of errors in the paper. I found small errors here and there, which I have noted in the attached PDF document.
Thank you, we have modified the text as indicated. The small corrections suggested in the annotated manuscript have been done.
Reviewer 2 Report
The paper has been improved and now motivates the study a bit better. However, I still think it could be motivated even better. Especially, it should be clear what scientific question(s) are answered in the paper.
It is nice to see that GLONASS is now included in the study.
Still, only one day and one satellite per system is studied. The authors stated in their cover letter that volume calculations of more days/satellites would follow in a separate paper. However, I am not convinced that the topic is scientifically interesting enough to motivate the study to be divided into two (or more) papers, hence I think it would be better to finish the study and summarize everything in a single paper.
I can understand the need for also including the Helmert transformation parameters, at least for translation and scale. For rotation, however, I would expect that this would be something that the orbit parameters could describe. Hence, are the Helmert rotation parameters really needed?
Line 107-111: explain the abbreviations MEO and IGSO at their first occurrence. Now they are explained first in the sentence after they first occur.
Author Response
Reviewer 2
Still, only one day and one satellite per system is studied. The authors stated in their cover letter that volume calculations of more days/satellites would follow in a separate paper. However, I am not convinced that the topic is scientifically interesting enough to motivate the study to be divided into two (or more) papers, hence I think it would be better to finish the study and summarize everything in a single paper.
Thank you, we have added Section 6 on volume calculations. Here we start from the algorithm described in the previous sections and tested for one satellite and one day for each constellation. We apply the algorithm to a full month of data (January 2020) for all the GPS, Glonass, Galileo and Beidou satellites. We compute average Helmert parameters (an Appendix has been added with detailed tables). We compare the average power spectra of the Broadcast minus SP3 differences before and after the least squares fit and show that the spectral features present in the pre-fit spectra at low frequencies have been filtered out by the proposed algorithm. We believe that this is the most effective way to prove that the proposed method works for all satellites and at any time.
I can understand the need for also including the Helmert transformation parameters, at least for translation and scale. For rotation, however, I would expect that this would be something that the orbit parameters could describe. Hence, are the Helmert rotation parameters really needed?
Agree, the rotations of the frame are equivalent to changes in the orbit angular parameters. This is emphasized in lines 366 – 374.
Line 107-111: explain the abbreviations MEO and IGSO at their first occurrence. Now they are explained first in the sentence after they first occur.
Done
Reviewer 3 Report
This paper presents a method to generate broadcast ephemeris with cm-level accuracy by fitting the parameters to post-processed IGS ephemeris. It could be interesting to the potential readers as it extends previous work on improved broadcast ephemeris of GPS to all the other GNSS constellations. However, I think the paper can be further improved by adding more test results and analysis of a longer period and all satellites, instead of presenting the results of several satellites on a single day as examples. A comprehensive literature review on existing work could also improve the paper, e.g., https://doi.org/10.1007/s10291-018-0820-0, as it already shows similar approaches and tested with large datasets. Therefore, I think major revision might be needed to make this paper acceptable for publication at Remote Sensing.
Specific comments:
- Line 36: I think the approaches should be tested with large datasets and the results and statistical analysis should be presented in this paper. The approach has already shown to be feasible in previous work (about GPS), so here we need more results to show it works in the context of GNSS, rather than a few numerical examples.
- Line 143-144: the same as above.
- Line 67: As far as I know, the GLONASS clock model is linear with parameters -TauN (clock bias) and GammaN (relative frequency bias).
- Line 195-200: why the 8 parameters are selected to be fixed while other parameters are estimated in Equation 1? Is it better to estimate the Helmert parameters first and then estimate the broadcast orbit parameters?
- Line 402: what does it mean that the scale factor accounts for the CoM to APC correction in the radial direction? does this correction also applies to GPS and Galileo (don’t see it in Table 1 and Table 2)? The nominal PCOs are released by IGS and can be used to mitigate this APC-CoM inconsistency…
- Line 438:why no Helmert parameters are considered for GLONASS?
- Line 467: it is unfair to compare a0 and a1 considering that the estimated GLONASS broadcast ephemeris has one additional term a2 compared to the original.
- Line 553: review of similar research on improve the GPS-like broadcast navigation ephemeris should be included, such as https://doi.org/10.1007/s10291-011-0243-7. The third set of cosine and sine parameters have been discussed so this section might be simplified.
- It might help to transform the fit residuals on X/Y/Z to the orbital radial, along-track, and cross-track system – this could be more intuitive to users
- The quality of the plots might be improved, e.g., the x-tick label for time- from units of seconds to hours; the y-tick range of dx dy dz to be consistent
- The results presented in Section 2 can be significantly simplified so that the readers can compare the results more intuitively – the current version is tedious – the satellites from different constellations can be included in the same type of plots; Table 1-4 can be listed in the same table; the discussions are quite similar for different satellites, etc.,
Author Response
Reviewer 3
- Line 36: I think the approaches should be tested with large datasets and the results and statistical analysis should be presented in this paper. The approach has already shown to be feasible in previous work (about GPS), so here we need more results to show it works in the context of GNSS, rather than a few numerical examples.
Thank you, we have added Section 6 on volume calculations. Here we start from the algorithm described in the previous sections and tested for one satellite and one day for each constellation. We apply the algorithm to a full month of data (January 2020) for all the GPS, Glonass, Galileo and Beidou satellites. We compute average Helmert parameters (an Appendix has been added with detailed tables). We compare the average power spectra of the Broadcast minus SP3 differences before and after the least squares fit and show that the spectral features present in the pre-fit spectra at low frequencies have been filtered out by the proposed algorithm. We believe that this is the most effective way to prove that the proposed method works for all satellites and at any time.
- Line 143-144: the same as above.
- Line 67: As far as I know, the GLONASS clock model is linear with parameters -TauN (clock bias) and GammaN (relative frequency bias).
We considered a quadratic for generality and in analogy with the other constellations. The quadratic term in any case is zero within one standard deviation.
- Line 195-200: why the 8 parameters are selected to be fixed while other parameters are estimated in Equation 1? Is it better to estimate the Helmert parameters first and then estimate the broadcast orbit parameters?
An important result in this paper is that a very low rms can be achieved estimating only a subset of the parameters of the broadcast model. In other words it is unnecessary to solve for the full set of 15 parameters of the broadcast model, for centimetric accuracy. We think that the block adjustment as described by the partial derivative matrix in Fig.1 is the best approach as it treats consistently the correlations which may result between the global parameters (Helmert) and the arc parameters (corrections to a subset of orbital model). The way you suggest would –in our view- force to zero these correlations.
- Line 402: what does it mean that the scale factor accounts for the CoM to APC correction in the radial direction? does this correction also applies to GPS and Galileo (don’t see it in Table 1 and Table 2)? The nominal PCOs are released by IGS and can be used to mitigate this APC- CoM inconsistency…
There is literature describing the equivalence between the CoM to APC correction and the scale factor in the Helmert transformation. In fact we have tested both ways and verified that the final results are identical. So, we see no real inconsistency, but rather a plurality of approaches leading to the same result.
- Line 438:why no Helmert parameters are considered for GLONASS?
This is explained in lines 666 and ff. : the Glonass broadcast model constrains the terrestrial reference frame by providing ECEF initial coordinates and velocities. Therefore, for Glonass it is not necessary to estimate Reference Frame parameters.
- Line 467: it is unfair to compare a0 and a1 considering that the estimated GLONASS broadcast ephemeris has one additional term a2 compared to the original.
As explained under item 3, the estimated term a2 of Glonass is always zero within one standard deviations even doubling the validity time of the Glonass message. It may be added that the Glonass message often has the a1 set to zero, but for 30 minutes validity time this is normally good enough. We considered appropriate to introduce a quadratic in an attempt to increase the validity time of the message and, hence, of the clock polynomial. If the linear model is accurate enough, the data will yield a a2 term compatible with zero, which is what we normally find.
- Line 553: review of similar research on improve the GPS-like broadcast navigation ephemeris should be included, such as https://doi.org/10.1007/s10291-011-0243-7. The third set of cosine and sine parameters have been discussed so this section might be simplified.
Thank you, we were aware of a PPT presentation but not of the full paper. The reference has been added. We think that our treatment is somewhat more exhaustive, and prefer to leave as it is.
- It might help to transform the fit residuals on X/Y/Z to the orbital radial, along-track, and cross-track system – this could be more intuitive to users
Agree, the new section 6 reporting volume calculations provides spectra of the pre and post fit residuals projected on the osculating triad.
- The quality of the plots might be improved, e.g., the x-tick label for time- from units of seconds to hours; the y-tick range of dx dy dz to be consistent
We prefer the notation with the seconds as it carries the second of week information and with it the information that the data refer to one specific day. The vertical lines have a spacing of two hours exactly, which helps in understanding the length of the time intervals. For the changes in the estimated parameters we did use the HH:MM notation. We checked the y-tick range of dx, dy dz for consistency.
- The results presented in Section 2 can be significantly simplified so that the readers can compare the results more intuitively – the current version is tedious – the satellites from different constellations can be included in the same type of plots; Table 1-4 can be listed in the same table; the discussions are quite similar for different satellites, etc.,
We agree in principle. Originally we made plots and tables with all the satellites in them. The resulting plots and tables were exceedingly crowded, difficult to understand. We preferred therefore the attitude that the results should be presented as clearly as possible, which implied organizing plots and tables in a systematic way.
Round 2
Reviewer 2 Report
The paper has been improved by adding the volume calulations. I have no further comments
Reviewer 3 Report
No more comments
This manuscript is a resubmission of an earlier submission. The following is a list of the peer review reports and author responses from that submission.
Round 1
Reviewer 1 Report
Review of article remotesensing-1349279 „Precise Orbits in Broadcast Ephemeris form. Part 2: Test Results for Glonass, High Rate Clocks and Model Improvements.“ by Alessandro Caporali and Joaquin Zurutuza.
This second paper follows on from the first in this series of two papers. First, it deals with the Broadcast Ephemeris (BE) of the Glonass satellites, which are not modeled using Kepler elements, but are calculated by transmitting the positions and velocities of the respective satellite using numerical integration. Glonass BE have the same purpose as GPS, Galileo or Beidou orbits: data are transmitted with the lowest possible bandwidth and at the same time allow a fast calculation of the satellite position with sufficient accuracy for navigation purposes. The paper also examines the precision of the transmitted clock corrections via a 2nd degree polynomial and additionally compares it with higher resolution clock corrections of the IGS (International GNSS Service). In the last part of the paper, an improvement of the orbital data to extend the validity of the respective BE is presented by considering the gravity field up to term J3 in the orbit calculation. The authors were able to show that the addition of J3 to the BE model makes a lot of sense and it has been shown to be a significant improvement over the standard modeling used to date.
Again, I would recommend the paper for publication. But as already stated, this and the previous paper need to be published together.
Again, I have some remarks. Some of my comments are included in the attached PDF.
In my opinion, the advantage of the improved BE in connection with real-time use is mentioned a bit late in the text. I think it would be advisable to prepare the reader early and much more clearly for the connection with real-time use. Only at the end of the discussion, in section 5, the implications for RTCM corrections are mentioned.
In general, I noticed that in this second part, but still independent essay, the abbreviations used are not sufficiently explained. As a non-expert, you would always have to look at the previous paper. I will try to mark all these abbreviations in the attached pdf-document.
Please, explain also why you have used two different sources for precise ephemeris (GRG and COD). What is the reason for this?
Links: please check your links again and keep in mind that data from CDDIS are now provided through “https” only. The ftp access is closed.
Language:
I found only small errors here and there, which I have corrected in the attached PDF document. The text is clearly formulated and easy to understand.

Reviewer 2 Report
The article that I received for review "Precise Orbits in Broadcast ephemeris form. Part 2: Test Results for Glonass, High Rate Clocks and Model Improvements" is basically a copy of Part I. In this the authors analyzed GLONASS signals, while in part I signals of GPS, Beidou and Galileo systems. I do not know what the other reviews are or will be, but from my perspective the design and construction of the part II is basically the same as the first one (part of the figures and equations are the same!).
Moreover, the article is of a low scientific level and does not generate any new knowledge (the same as part I). This basically looks like plagiarism of Part I with other, GLONASS data.
Unfortunately, I do not see anything new in this article, therefore I do not recommend it for publication in Remote Sensing.